# Interfacial electronic structure engineering on molybdenum sulfide for robust dual-pH hydrogen evolution

Mingqiang Liu[1,2,7], Jia-Ao Wang[3,7], Wantana Klysubun[4], Gui-Gen Wang [1✉], Suchinda Sattayaporn[4], Fei Li[1], Ya-Wei Cai[1], Fuchun Zhang[5], Jie Yu [1] & Ya Yang [2,6✉]

Molybdenum disulfide, as an electronic highly-adjustable catalysts material, tuning its electronic structure is crucial to enhance its intrinsic hydrogen evolution reaction (HER) activity. Nevertheless, there are yet huge challenges to the understanding and regulation of the surface electronic structure of molybdenum disulfide-based catalysts. Here we address these challenges by tuning its electronic structure of phase modulation synergistic with interfacial chemistry and defects from phosphorus or sulfur implantation, and we then successfully design and synthesize electrocatalysts with the multi-heterojunction interfaces (e.g., $1T_{0.81}$-$MoS_2$@$Ni_2P$), demonstrating superior HER activities and good stabilities with a small overpotentials of 38.9 and 95 mV at 10 mA/cm$^2$, a low Tafel slopes of 41 and 42 mV/dec in acidic as well as alkaline surroundings, outperforming commercial Pt/C catalyst and other reported Mo-based catalysts. Theoretical calculation verified that the incorporation of metallic-phase and intrinsic HER-active Ni-based materials into molybdenum disulfide could effectively regulate its electronic structure for making the bandgap narrower. Additionally, X-ray absorption spectroscopy indicate that reduced nickel possesses empty orbitals, which is helpful for additional H binding ability. All these factors can decrease Mo-H bond strength, greatly improving the HER catalytic activity of these materials.

[1] Shenzhen Key Laboratory for Advanced Materials, School of Materials Science and Engineering, Harbin Institute of Technology, Shenzhen, People's Republic of China. [2] CAS Center for Excellence in Nanoscience, Beijing Key laboratory of Micro-nano Energy and Sensor, Beijing Institute of Nanoenergy and Nanosystems, Chinese Academy of Science, Beijing, People's Republic of China. [3] Department of Chemistry and the Oden Institute for Computational Engineering and Sciences, University of Texas at Austin, Austin, TX, USA. [4] Synchrotron Light Research Institute, Muang, Nakhon Ratchasima, Thailand. [5] School of Physics and Electronic Information, Yan'an University, Yan'an, People's Republic of China. [6] School of Nanoscience and Technology, University of Chinese Academy of Sciences, Beijing, People's Republic of China. [7]These authors contributed equally: Mingqiang Liu, Jia-Ao Wang. ✉email: wangguigen@hit.edu.cn; yayang@binn.cas.cn

Extensive use and depletion of fossil fuels resulting in serious pollution. Therefore, green and renewable fuel resources are required for continuing sustainable economic development[1–3]. Electrocatalysis acts as a vital role in the conversion of clean energy to achieve a sustainable approach to various commercial processes, including HER[4,5]. However, electrochemical water splitting is hindered by the large kinetic barrier and slow kinetics[6–9]. Pt-based electrocatalysts are recognized as highly efficient electrocatalysts due to good electrical conductivity[10], fast kinetics[11,12], and the preference to overcome the large kinetic energy barrier involved in the above-mentioned process[13]. Unfortunately, high price and not desirable stability hinder the extended Pt-based catalysts' application[14]. Thus, it is very urgent to develop cost-effective Pt-free electrocatalysts with comparable activity and better stability.

Researchers recently have designed a wide range of low-cost catalysts, including transition-metal chalcogenides (TMDCs)[15,16], metal nitrides[17,18], metal carbides[19,20], and metal phosphides[21,22]. Among these candidates, $MoS_2$, a typical layered 2D TMDCs formed by Van der Waals interaction and stacking of S–Mo–S layers, attracts extensive interests with its adjustable bandgap, unique band structure, high energy-conversion efficiency, and earth abundance[23–25]. However, the electrocatalytic activity of $MoS_2$ is closely associated with its surface electric structure[26–36], many researchers have focused on adjusting the electronic structure of the $MoS_2$ surface to promote electrocatalytic activity, such as surface engineering[26], doping[27], single-atom anchoring[28], phase structure[29–33], interface active site[34,35], and defect[36]. Interestingly, two main phases of $MoS_2$ were widely justified: 2H and 1T phases[29]. 2H phase has the most thermodynamical stability among the molybdenum sulfide family, whose HER activities are restrained by the amount and active site types as well as conductivity. Unlike the 2H phase, 1T-phase one demonstrates higher catalytic activity since it has numerous active sites on the edges and a fast transfer rate. However, it is remaining a giant challenge of directly synthesizing the high percentage 1T-phase molybdenum sulfide due to the thermodynamic instability of $1T_{phase}$-$MoS_2$[30]. To solve this problem, a feasible strategy is to efficiently realize the 2H → 1T-phase transformation to improve HER capability. Wang et al. found that a partial 2H → 1T-$MoS_2$ phase transition by facile one-pot annealing of a large amount of $2H_{phase}$-$MoS_2$ under phosphorus vapor is able to enhance HER catalytic activities[31]. A synergistic strategy of doping nitrogen and intercalating $PO_4^{3-}$ is reported, which can convert 2H- to 1T-phase with a conversion rate of up to 41%, and has excellent HER performance[32]. However, the electronic transport capacity and phase stability of the phase boundary of a single component (pure 1T-phase) are generally poor. In order to overcome the puzzles, the HER activity of the pure phase can be improved by constructing a heterogeneous boundary. Therefore, it is expected to further enhance the HER performance and its stability of traditional single 1T-phase or 2H-phase interface by constructing a composite heterojunction between 1T-phase and the other phases[33].

Interface modification could be an effective approach to construct a composite heterojunction[34,35]. Ni-based materials (such as $Ni_2P$, $NiS_2$, $Ni_2S_3$, etc) with high activity and conductivity have been considered as highly efficient electrocatalysis materials for HER[21,22,37], as another heterogeneous interface, which is also very important to control the electronic structure of the $MoS_2$ interface. Kim et al. reported that $Ni_2P$ nanoparticles were used to activate the $MoS_2$ base surface, which exhibits Pt-like HER performance in 0.5 M HCl solution[37]. Because the electronic structure of $Ni_2P$ is a $P\bar{6}2m$ space group, which could facilitate recombination at the atomic scale. Moreover, Ni has a unique α and β orbital integral asymmetric $d$ orbital, which makes it easy

for the lone pair of electrons to recombine with the $d$ orbital of the exposed Mo atom on $MoS_2$ to generate new interface electrons, thereby improving HER performance. Lin et al. reported that a defect-rich heterogeneous interfacial catalyst ($MoS_2$/$NiS_2$) could provide abundant active sites to promote electron transfer, thereby further rapidly promoting electrocatalytic hydrogen evolution[38]. More importantly, the introduction of $NiS_2$ hybridization on the surface of $MoS_2$ generates a new form of interface electrons, and $Ni^{\delta+}$ is reduced to low-valence Ni to improve the binding energy with hydrogen elements, thereby weakening the Mo–H strength. To sum up, although the heterojunction-phase catalyst synthesized by the above-mentioned approach further improves the HER activity and good stability, the understanding and regulation of the surface electronic structure on the $MoS_2$ interface are still huge challenges, and thus it is very necessary to develop an efficient synthesis approach to obtain stable multi-heterogeneous interface catalyst.

Here, we address these challenges by tuning its electronic structure through phase modulation synergistic with interfacial chemistry and defects of phosphorus or sulfur implantation, and we then successfully design and prepare a series of heterojunction-phase-interface electrocatalysts (denoted $1T_{0.81}$-$MoS_2$@$Ni_2P$ and $1T_{0.72}$-$MoS_2$@$NiS_2$) with an outstanding HER activity and are stable in dual-pH surroundings. The strategies to control the electronic characteristics of the $MoS_2$ surface include surface phase modulation, surface defects, and the construction of hetero-structure (Fig. 1a). Furthermore, we control the hydrogen and hydroxyl adsorption energy through the synergistic effect of heterojunction-phase-interface catalysts (Fig. 1b, c–f) because the energy of the hydroxyl species is very important for the hydrolysis accelerator. Starting from hydrothermally synthesized $MoS_2$ nanosheets, we develop a simple surface electronic structure modulation strategy of constructing multi-heterogeneous-phase-interface $1T_{0.81}$-$MoS_2$@$Ni_2P$ and $1T_{0.72}$-$MoS_2$@$NiS_2$ electrocatalysts (Fig. 1a) by citric acid-induced hydrothermal synthesis, electrodeposition and then phosphorus (or sulfur) vapor thermal treatment approach for the first time. Our approach can not only realize the construction of abundant catalytic reactive sites but also improve the conversion rate of 2H to 1T (81%), and it is also convenient to introduce $Ni_2P$ or $NiS_2$ heterogeneous interfaces. As to the surface electronic structure of catalysts, high-resolution transmission electron microscopy (HRTEM) images show that such phase-structures, heterojunction-phase-interface edges, and defects are derived by the featured electronic states and Ni atomic coordination. Additionally, X-ray photoelectron spectra (XPS) showed that citric acid induces hydrothermal synthesis of stable $1T_{0.41}$-$MoS_2$ (41% of 1T-phase), and the $1T_{0.81}$-$MoS_2$ or $1T_{0.72}$-$MoS_2$ (81% or 72% of 1T-phase) conversion rate is further improved after phosphorus or sulfur vapor thermal treatment. As-synthesized $1T_{0.81}$-$MoS_2$@$Ni_2P$ (or $1T_{0.72}$-$MoS_2$@$NiS_2$) multi-heterogeneous catalyst exhibits the remarkable HER catalytic activity, achieving the low overpotentials of 38.9 (or 186) and 98.5 mV (or 128 mV) for HER at a current density of 10 mA/cm². They also have Tafel slopes of 41 (or 79) and 42 (or 68) mV/dec in 0.5 M $H_2SO_4$ or 1.0 M KOH media, and good stability during testing for 16 h in both media, respectively. The $1T_{0.81}$-$MoS_2$@-$Ni_2P$ (or $1T_{0.72}$-$MoS_2$@$NiS_2$) catalysts exhibited superior activities with Tafel slope values and the overpotentials lower than the values reported for Mo-base HER catalysts in both alkaline and acidic media[30,31,33,37–40]. Moreover, as-synthesized $1T_{0.72}$-$MoS_2$@$NiS_2$ (or $1T_{0.81}$-$MoS_2$@$Ni_2P$) catalyst also exhibits excellent OER and overall-water splitting catalytic activity. DFT calculation results display that the introduction of 1T-phase $MoS_2$ and Ni-based materials can regulate $MoS_2$ electronic structure effectively for making the bandgap narrower, and decreasing H* and water adsorption energy. In situ electrochemical-Raman

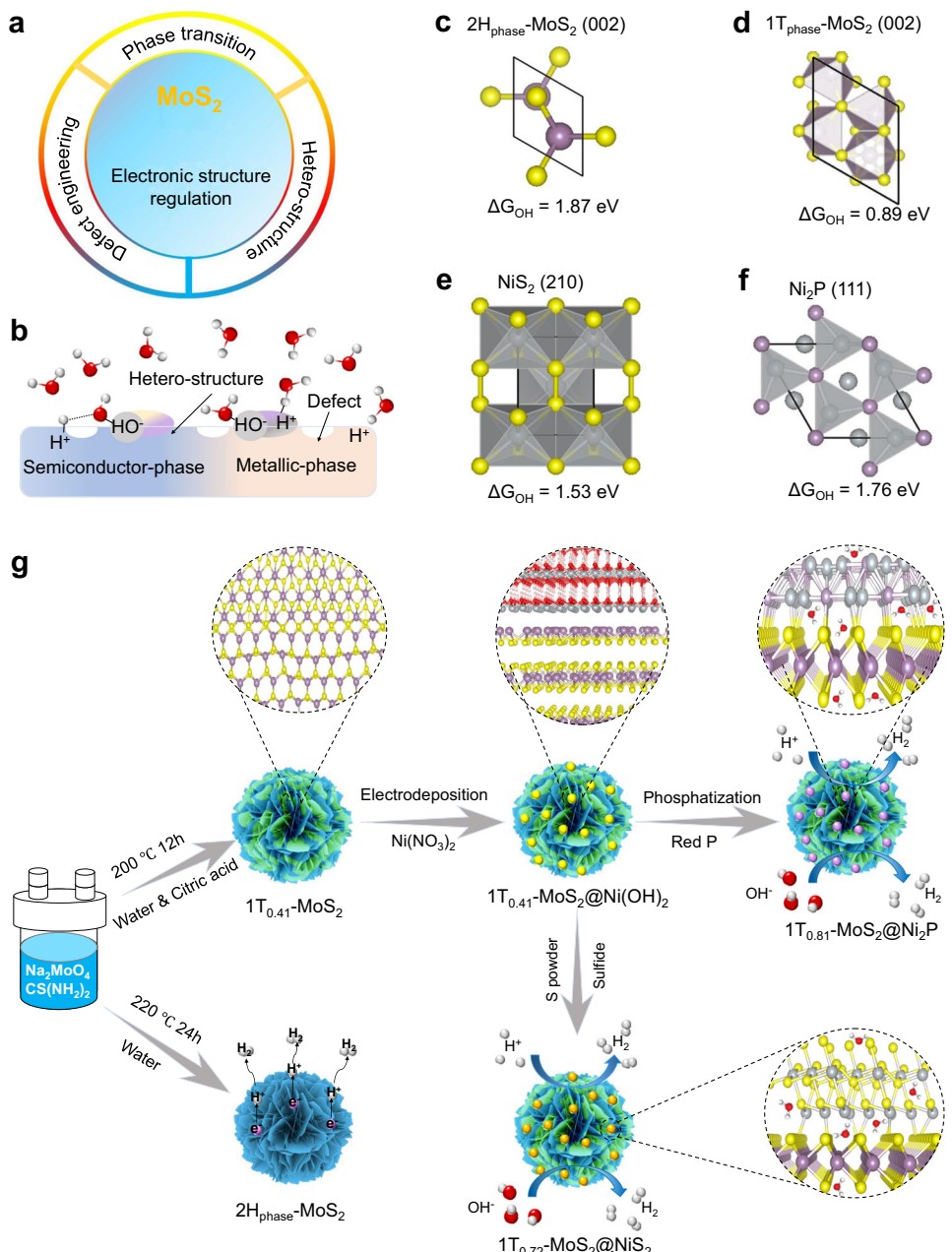

**Fig. 1 Motivation and design of electrocatalyst. a** Tuning strategy of the electronic structure of the $MoS_2$ surface. **b** Design ideas of hydrogen evolution catalyst. **c–f** the energetics of hydroxyl species on $2H_{phase}$-$MoS_2$ (002), $1T_{phase}$-$MoS_2$ (002), $NiS_2$ (210), and $Ni_2P$ (111) HER electrocatalyst surfaces. **g** Schematics of the $1T_{0.72}$-$MoS_2$@$NiS_2$ and $1T_{0.81}$-$MoS_2$@$Ni_2P$ synthesis steps.

spectra results indicate that the OH– ions are driven to be adsorbed on Mo, Ni atoms in the alkaline medium, and then there forms the OOH* intermediates. There is a strong interaction between Ni and Mo on the surface of the catalyst, thereby increasing the local electronic state of Mo atoms, reducing the hydrogen-adsorption energy for protons on Mo atoms, and thus improving its intrinsic catalytic. Moreover, X-ray absorption spectroscopy results imply that reduced Ni supply empty *d*-orbitals to facilitate H atom capture, and decrease Mo–H strength of $1T_{0.81}$-$MoS_2$@$Ni_2P$ (or $1T_{0.72}$-$MoS_2$@$NiS_2$) catalyst. This work provides useful insights for exploring the enhancement mechanisms of HER with an optimized surface electronic structure on the $MoS_2$ interface, which provides an effective insight of constructing invaluable metal electrocatalysts for HER and other fields.

## Results

**Preparation and characterizations of multi-heterojunction interface electrocatalysts**. The formation process of multi-heterojunction interface electrocatalysts is schematically illustrated in Fig. 1g. $1T_{0.81}$-$MoS_2$@$Ni_2P$ and $1T_{0.72}$-$MoS_2$@$NiS_2$ catalysts were synthesized by a three-step procedure. First, $1T_{0.41}$-$MoS_2$ nanospheres were obtained on carbon cloth (CC) by acid-induced hydrothermal approach at 200 °C for 12 h (see details in "Methods" section). The as-obtained $1T_{0.41}$-$MoS_2$ shows a large number of microspheres (Supplementary Fig. 1b–d) with a narrow diameter distribution of 2.0–4.0 μm distributed uniformly on the surface of CC substrate. Flower-shaped $MoS_2$ microspheres consist of many aligned $1T_{0.41}$-$MoS_2$ nanosheets, on which the $Ni(OH)_2$ nanoparticles were then electrodeposited (see details in "Methods" section). $1T_{0.41}$-$MoS_2$@$Ni(OH)_2$ material inherited its

morphology from spherical $MoS_2$. When being electrodeposited for 100 s, a small amount of $Ni(OH)_2$ nanoparticles can be anchored on the surface of $MoS_2$ nanospheres (Supplementary Fig. 2). As the electrodeposition time increases to 300 s, a large number of $Ni(OH)_2$ nanoparticles can be observed to adhere to the $MoS_2$ surface (Supplementary Fig. 3). Subsequently, as-prepared $1T_{0.41}$-$MoS_2$@$Ni(OH)_2$ was loaded into a quartz tube mixed with red phosphorus or sulfur powder and sealed by oxyacetylene flame. Finally, these were heated to 600 °C for the reaction with red phosphorus or sulfur to synthesize $1T_{0.81}$-$MoS_2$@$Ni_2P$ and $1T_{0.72}$-$MoS_2$@$NiS_2$ catalysts, respectively (Supplementary Figs. 4 and 5). As to $1T_{0.81}$-$MoS_2$@$Ni_2P$ catalyst, the $MoS_2$ microspheres are very rough, on which there distribute many random $Ni_2P$ nanoparticles (Supplementary Fig. 5). It is because that the 1T/2H-mixed phase and heterojunction-interface structure reduces the adhesion of the gas-solid interface and facilitates releasing hydrogen from the catalyst surface, which is essential for enhancing HER[34].

Next, the phase composition and crystal structure of $1T_{0.81}$-$MoS_2$@$Ni_2P$ and $1T_{0.72}$-$MoS_2$@$NiS_2$ were obtained by X-ray diffraction (XRD) and Raman spectroscopy. There are some obvious characteristic diffraction peaks of 14.3°, 33.4°, and 59.2° (Supplementary Fig. 6b, c), which can be ascribed to $2H_{phase}$-$MoS_2$ (JCPDS#75-1539). However, the XRD peak of $1T_{0.81}$-$MoS_2$@$Ni_2P$ and $1T_{0.72}$-$MoS_2$@$NiS_2$ located at $2\theta \approx 28.8°$ can be indexed as the (004) peak of $1T_{phase}$-$MoS_2$, which indicates that 1T- and 2H-mixed phases were successfully hydrothermally synthesized[41]. The other characteristic peaks ($2\theta \approx 31.3°$, 35.2°, 38.8°, 44.9°, and 53.3°) demonstrate that the $1T_{0.72}$-$MoS_2$@$NiS_2$ is a hybrid of $NiS_2$ (JCPDS#11-0099), which verifies the presence of $NiS_2$ nanoparticles. Similarly, as to $1T_{0.81}$-$MoS_2$@$Ni_2P$ catalyst, its XRD results also showed the presence of $Ni_2P$ nanoparticles (JCPDS#21-0590) on the $1T_{0.41}$-$MoS_2$ surface. Raman spectroscopy showed $E_{2g}^1$ and $A_{1g}$ vibrational bands at 376.2 and 402.9 $cm^{-1}$ peaks typical for $2H_{phase}$-$MoS_2$[42]. $J_1$, $J_2$, and $J_3$ vibrations at 147.3, 235.4 and 335.2 $cm^{-1}$ are characteristic for $1T_{phase}$-$MoS_2$[43] (Supplementary Fig. 6b). These results prove that the 1T-phase of $MoS_2$ is formed by the hydrothermal reaction induced by organic acids (e.g., citric acid)[41]. $1T_{0.72}$-$MoS_2$@$NiS_2$ or $1T_{0.81}$-$MoS_2$@$Ni_2P$ demonstrated three characteristic peaks of $1T_{phase}$-$MoS_2$ and the two characteristic peaks ($E_{2g}^1$ and $A_{1g}$) of $2H_{phase}$-$MoS_2$. Additionally, they showed a vibrational peak (437.3 $cm^{-1}$) of Ni–S[38] or three vibrational peaks (216.2, 249.7, and 269.5 $cm^{-1}$) of Ni–P[37]. More importantly, the $E_{2g}^1$ and $A_{1g}$ vibrations of $1T_{0.72}$-$MoS_2$@$NiS_2$ at 382.2 and 408.1 $cm^{-1}$ were red-shifted by 6.0 and 5.2 $cm^{-1}$, respectively (Supplementary Fig. 6d). This could be attributed to the exploits the S layer of $MoS_2$ as an external S source to grow $NiS_2$ in situ. Therefore, it changes the original vibration mode of the Mo–S bonds, and the out-of-plane vibration mode has a more significant change[44–46]. Similarly, the $E_{2g}^1$ and $A_{1g}$ peaks for the $1T_{0.81}$-$MoS_2$@$Ni_2P$ catalyst slightly red-shifted by 7.3 and 3.0 $cm^{-1}$, respectively, because of interfacial stress between $Ni_2P$ and $MoS_2$, indicating that the formation of $MoS_2$@$Ni_2P$ heterojunction leads to the Raman shift of $MoS_2$[44–46]. These results confirm that rich multi-heterojunction interface edges active sites catalysts were successfully synthesized.

**Electronic structure characterizations of $1T_{0.72}$-$MoS_2$@$NiS_2$ and $1T_{0.81}$-$MoS_2$@$Ni_2P$ catalysts**. To further identify the surface electronic structure of multi-heterogeneous interface catalysts, we applied the high-resolution transmission electron microscopy (HRTEM) to assess the morphology and crystal structures of $1T_{0.81}$-$MoS_2$@$Ni_2P$ and $1T_{0.72}$-$MoS_2$@$NiS_2$ catalysts. Supplementary Fig. 7a, b shows the typical low-magnification image of

the $1T_{0.72}$-$MoS_2$@$NiS_2$ on the Cu grid, which confirms the flower-like nanosphere morphologies of $1T_{0.72}$-$MoS_2$@$NiS_2$. TEM and corresponding elemental distribution map obtained for the $1T_{0.72}$-$MoS_2$@$NiS_2$ sample demonstrated uniformly distributed Mo, Ni, and S (Supplementary Fig. 7c-c4). As revealed by the HRTEM image (Fig. 2a–c and Supplementary Fig. 7e, f), $NiS_2$ nanoparticles are decorated on $MoS_2$ nanosheets edge (Supplementary Fig. 10a, b). The HRTEM image of $1T_{0.72}$-$MoS_2$@$NiS_2$clearly shows the crystal lattice of 0.25 nm, referring to the $NiS_2$ (210). Interestingly, Fig. 2a shows the HRTEM image of $1T_{0.72}$-$MoS_2$@$NiS_2$ flower-like nanosheets, which there demonstrate the lattice fringes perpendicularly to the electron beam direction circled by blood color, justifying the S defect (Fig. 2c). The trigonal lattice in the yellow circle implies the presence of 1T-phase $MoS_2$, while the hexagonal lattice in the blue circle suggests the presence of 2H phase $MoS_2$. The above-described results further confirm the successful preparation of the $1T_{0.72}$-$MoS_2$@$NiS_2$ multi-heterojunction interface catalyst. The anion is changed to be P to produce $1T_{0.81}$-$MoS_2$@$Ni_2P$ multi-heterojunction interface catalyst by phosphorus vapor thermal treatment. Supplementary Fig. 8a, b displays the morphologies of $1T_{0.81}$-$MoS_2$@$Ni_2P$ catalyst, overlapping nanosheets with many embedded particles can be clearly identified. There is an obvious alternation of 1T and 2H phases, and a large number of defects or disorder (Fig. 2f and Supplementary Fig. 9). As shown in Supplementary Fig. 8c, there are the distributions of Mo, Ni, S, and P over the whole $1T_{0.81}$-$MoS_2$@$Ni_2P$, verifying that $Ni_2P$ nanoparticles are encapsulated by $MoS_2$ edges (Supplementary Fig. 10c, d). The interplanar spacings of 0.62 and 0.22 nm are assigned to (002) and (111) interplanar distances of $MoS_2$ and $Ni_2P$, respectively (Fig. 2d, e). Similarly, Fig. 2e, f displays two amplified HRTEM images truncated from Fig. 2d, in which Fig. 2f demonstrates some hexagonal and trigonal lattice areas of semiconductor $2H_{phase}$- and metallic $1T_{phase}$-$MoS_2$, respectively. The HRTEM results further confirm the successful preparation of the $1T_{0.81}$-$MoS_2$@$Ni_2P$ multi-heterojunction interface catalyst.

Next, we performed XPS measurement to assess the elemental valence states of all the as-synthesized samples (Fig. 2g–i and Supplementary Fig. 13a). Full XPS spectrum for $1T_{0.72}$-$MoS_2$@-$NiS_2$ (Supplementary Fig. 13a) showed that atomic ratios of Mo, S, and Ni were equal to 13.96%, 36.96%, and 4.39%, respectively, and close to that measured by HRTEM elemental mapping (~14.30%, 35.87%, and 4.76%). Mo $3d$ spectra obtained for the $1T_{0.41}$-$MoS_2$ sample shows $Mo^{4+}$ $3d_{3/2}$ and $Mo^{4+}$ $3d_{5/2}$ peaks at 232.68 and 229.43 eV (Fig. 2g), respectively, confirming the existence of $Mo^{4+}$ for the $1T_{0.41}$-$MoS_2$. As to the $1T_{0.72}$-$MoS_2$@$NiS_2$, or $1T_{0.81}$-$MoS_2$@$Ni_2P$ heterostructures catalyst, the high-solution Mo $3d$ XPS spectrum shows that both $Mo^{4+}$ $3d_{3/2}$ and $Mo^{4+}$ $3d_{5/2}$ peaks for mixed-phase $MoS_2$ has a shift of 0.23 eV and 0.15 eV to lower binding energy compared with $1T_{0.41}$-$MoS_2$ (Supplementary Fig. 11a), which is attributed to the existence of $1T_{phase}$-$MoS_2$[47]. In addition, two peaks of 163.41 and 162.22 eV are observed in the $1T_{0.41}$-$MoS_2$, corresponding to $S^{2-}$ $2p_{1/2}$ and $S^{2-}$ $2p_{3/2}$, respectively (Fig. 2h). However, the binding energies of $S^{2-}$ $2p_{1/2}$ and $S^{2-}$ $2p_{3/2}$ in $1T_{0.72}$-$MoS_2$@$NiS_2$, or $1T_{0.81}$-$MoS_2$@$Ni_2P$ heterostructures catalyst shift to 163.28 and 162.10 eV, respectively (Supplementary Fig. 11b). This negative-shift (0.13 eV) suggests little electron transfer between $NiS_2$ (or $Ni_2P$) and $MoS_2$, also suggesting the reconfiguration of the electronic structure during the transferring of electron from $Mo^{4+}$ to the surrounding Ni sites[48]. Interestingly, the 1T-phase contents in the $1T_{0.81}$-$MoS_2$@$Ni_2P$ and $1T_{0.72}$-$MoS_2$@$NiS_2$ samples (81% and 72%, respectively) were higher than the 41% value observed for the $1T_{0.41}$-$MoS_2$. Thus, phosphorus or sulfur implantation further facilitates the phase transformation of $1T_{phase}$-$MoS_2$[24,32]. The reason may be that phosphorus can be

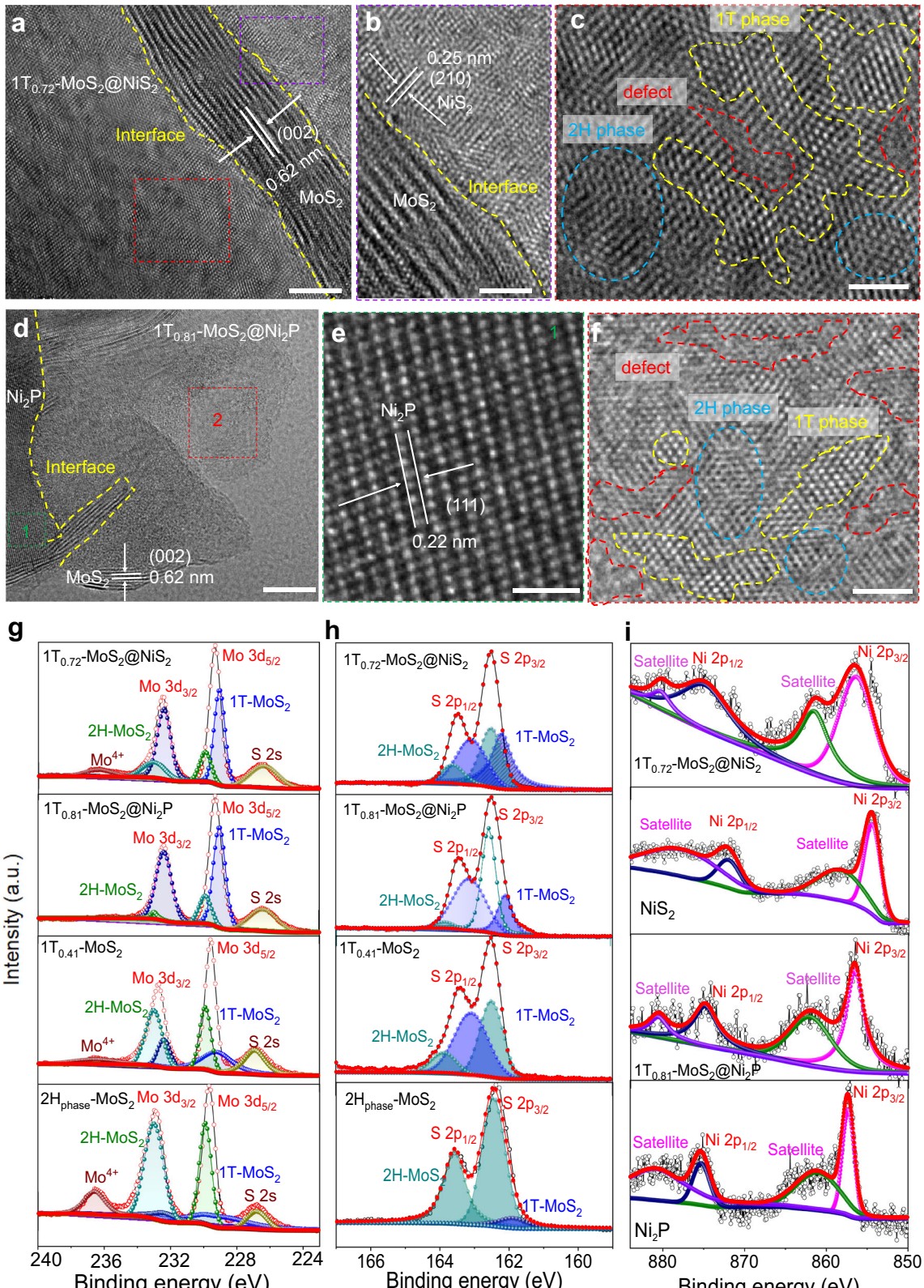

**Fig. 2 Electronic structure characterizations of $1T_{0.72}$-MoS$_2$@NiS$_2$ and $1T_{0.81}$-MoS$_2$@Ni$_2$P catalysts. a–c** HRTEM image of $1T_{0.72}$-MoS$_2$@NiS$_2$. **a** MoS$_2$ lattice, scale bars are 5 nm. **b** NiS$_2$ lattice and heterojunction interface, scale bars are 1 nm. **c** shows 2H and 1T lattices, scale bars are 2 nm. **d–f** Typical HRTEM image of $1T_{0.81}$-MoS$_2$@Ni$_2$P. **d** shows MoS$_2$ lattice and heterojunction interface, scale bars are 5 nm. **e** shows Ni$_2$P lattice fringes, scale bars are 1 nm. **f** shows 2H and 1T lattices, scale bars are 2 nm. **g** HR Mo 3$d$ core-level XPS spectra of $1T_{0.72}$-MoS$_2$@NiS$_2$, $1T_{0.81}$-MoS$_2$@Ni$_2$P, $1T_{0.41}$-MoS$_2$, and $2H_{phase}$-MoS$_2$. **h** S 2$p$ core-level XPS spectra of $1T_{0.72}$-MoS$_2$@NiS$_2$, $1T_{0.81}$-MoS$_2$@Ni$_2$P, $1T_{0.41}$-MoS$_2$ and $2H_{phase}$-MoS$_2$, respectively. **i** Ni 2$p$ XPS spectrum for $1T_{0.72}$-MoS$_2$@NiS$_2$, $1T_{0.81}$-MoS$_2$@Ni$_2$P, pure Ni$_2$P, and Ni$_2$S.

simultaneously inserted into S–Mo–S atomic planes, inducing the glide of S atomic planes, affording in-plane heterostructures between 1T and 2H MoS$_2$ domains (Supplementary Fig. 14), which is consistent with previous reports[31,32]. As shown in Fig. 2i, the Ni 2$p$ spectrum of 1T$_{0.81}$-MoS$_2$@Ni$_2$P shows two spin–orbit doublets at 856.6 and 874.9 eV corresponding to Ni$^{2+}$ 2$p_{3/2}$ and Ni$^{2+}$ 2$p_{1/2}$ oxidation states in Ni$_2$P, respectively, and two satellite peaks (identified as "Satellite.")[49]. Notably, compared with the binding energies of Ni 2$p_{3/2}$ (857.4 eV) and Ni 2$p_{1/2}$ (875.4 eV) of pure Ni$_2$P, the two binding energies of Ni 2$p_{3/2}$ and Ni 2$p_{1/2}$ have a significant negative-shift of approximately 0.8 and 0.6 eV in 1T$_{0.81}$-MoS$_2$@Ni$_2$P (Fig. 2i), respectively. This result implies the transfer of electrons from Mo$^{4+}$ to Ni$^{2+}$ sites in the 1T$_{0.81}$-MoS$_2$@Ni$_2$P sample, resulting in a low-valence state and electron-rich structure of Ni$^{2+}$ sites[50]. For pure NiS$_2$ sample, the peaks of Ni 2$p_{3/2}$ and Ni 2$p_{1/2}$ located at 854.7 and 872.2 eV, and the corresponding satellites appear at 858.7 and 878.6 eV, respectively. However, the binding energies of Ni 2$p_{3/2}$, Ni 2$p_{1/2}$ and satellite in 1T$_{0.72}$-MoS$_2$@NiS$_2$ sample (Fig. 2i) are positive-shifted to 856.6 (by 1.9 eV), 875.3 (by 3.1 eV), 861.8 (by 3.1 eV) and 862.2 (by 1.9 eV), respectively. The positive-shift of the Ni 2$p$ binding energies peaks manifest a higher valence state, which are ascribed to Ni bonded to S and O atoms, such as sulfides or surface oxides/hydroxides[51]. For the 1T$_{0.81}$-MoS$_2$@Ni$_2$P, the P 2$p$ spectrum shows two peaks at 130.4 and 129.5 eV corresponding to P 2$p_{1/2}$ and P 2$p_{3/2}$, respectively, suggesting the existence of Ni$_2$P. In addition, it also can be observed another peak at 134.7 eV of oxidized phosphate (P–O) species (Supplementary Fig. 13b), which is due to the partial oxidation of Ni$_2$P in air. Notably, the binding energies of Ni 2$p_{3/2}$ (852.7 eV) and P 2$p_{3/2}$ (129.5 eV) are both shifted, indicating that charge transfer occurs from Ni to P, which can greatly promote the catalytic activity of 1T$_{0.81}$-MoS$_2$@Ni$_2$P.

**Electrocatalytic HER performances in alkaline and acidic media.** 1T$_{0.72}$-MoS$_2$@NiS$_2$ and 1T$_{0.81}$-MoS$_2$@Ni$_2$P electrodes exhibited attractive multi-heterogeneous interface edges, plentiful active sites, and abundant mass transfer and gas release channels and are expected to be used as very effective and stable catalysts for H$_2$ production. First, we analyzed HER activities (in 1.0 M KOH) of the electrodes containing these electrodes. The 1T$_{0.72}$-MoS$_2$@NiS$_2$ and 1T$_{0.81}$-MoS$_2$@Ni$_2$P electrodes exhibit small overpotentials of 95 and 170 mV at 10 mA/cm$^2$, respectively (see linear sweep voltammetry (LSV) results in Fig. 3a), which are better than the commercial Pt/C electrode (127 mV). To in-depth understand the HER kinetic mechanism, we calculated the Tafel slopes of these electrodes using the Tafel equation[52] and obtained the smallest slopes equal to 68 and 79 mV/dec for the 1T$_{0.72}$-MoS$_2$@NiS$_2$ and 1T$_{0.81}$-MoS$_2$@Ni$_2$P electrodes, respectively (Supplementary Fig. 15a). These values are even closer to the Tafel slope of the Pt/C electrode (56 mV/dec). Thus, the 1T$_{0.72}$-MoS$_2$@NiS$_2$ and 1T$_{0.81}$-MoS$_2$@Ni$_2$P electrodes as active electrocatalysts exhibit the fastest HER processes and better reactivity, which is attributed to the multi-heterogeneous interface effect, a large number of defects, and a higher proportion of 1T$_{phase}$-MoS$_2$. Next, we evaluated the long-term cycling stability of the as-prepared electrodes using the chronopotentiometry technique at 10 and 30 mA/cm$^2$, respectively. The 1T$_{0.72}$-MoS$_2$@NiS$_2$ and 1T$_{0.81}$-MoS$_2$@Ni$_2$P electrodes were very robust and exhibited negligible damping after 16 h measurement (Supplementary Fig. 15b), and the LSV curves measured before and after the long-term tests are almost the same (Supplementary Fig. 15c), demonstrating excellent long-term stability. Supplementary Fig. 15d lists the overpotential values for the 20.0 wt % Pt/C, 1T$_{0.72}$-MoS$_2$@NiS$_2$, and 1T$_{0.81}$-MoS$_2$@Ni$_2$P electrodes in 1.0 M

KOH at various current densities. 1T$_{0.72}$-MoS$_2$@NiS$_2$ electrodes exhibited lower overpotential. Generally, low overpotential and Tafel slope values demonstrated the superior HER catalytic activities, which was the case for our 1T$_{0.72}$-MoS$_2$@NiS$_2$ and 1T$_{0.81}$-MoS$_2$@Ni$_2$P electrodes. Moreover, 1T$_{0.72}$-MoS$_2$@NiS$_2$ electrode has such excellent HER activity comparable to those of as-reported Mo-based materials (Fig. 3b) and composites and various representative catalysts[30,31,33,37–40] (Supplementary Table 3). Thus, 1T$_{0.72}$-MoS$_2$@NiS$_2$ electrode is a catalyst with the best HER activity in alkaline solutions.

To obtain the electrochemically active area (ECSA) of the 1T$_{0.72}$-MoS$_2$@NiS$_2$ and 1T$_{0.81}$-MoS$_2$@Ni$_2$P electrodes, the double-layer capacitance ($C_{dl}$) was calculated because the two values are proportional to each other. Therefore, we tested their cyclic voltammetry (CV) by continuously increasing scanning speed (Supplementary Fig. 16a–c) in order to obtain the CV curve of the electrode materials in the non-Faraday region (−0.2 to 0.4 V). Then, as shown in Supplementary Fig. 16d, the $C_{dl}$ was calculated from the plot slope (slope = $2C_{dl}$) between current-density difference ($\Delta j$) (0.15 V vs. RHE) and scan rate. The 1T$_{0.72}$-MoS$_2$@NiS$_2$ electrodes possessed the highest $C_{dl}$ value ($C_{dl}$ = 359.7 mF/cm$^2$), suggesting a multi-heterogeneous interface could be effectively enhanced conductivity and exposed more active sites of as-prepared electrodes. We recorded the electrochemical impedance spectra (EIS). The corresponding Nyquist (Supplementary Fig. 17) of the 1T$_{0.72}$-MoS$_2$@NiS$_2$ electrode showed the lowest value for the charge transfer resistance ($R_{ct}$). Thus, it possessed very favorable charge transfer kinetics. To further reveal the intrinsic catalytic activity of each active sites, the turnover frequency (TOF) is also calculated[53]. Based on the above-mentioned analysis, CV approach is regarded as the promising way to determine reasonable results (Supplementary Fig. 18). The TOF value of 1T$_{0.81}$-MoS$_2$@Ni$_2$P (3.56 S$^{-1}$) and 1T$_{0.72}$-MoS$_2$@NiS$_2$ (2.26 S$^{-1}$) heterojunction catalyst at the overpotential of 200 mV is 18.7 and 11.9 times higher than of 2H$_{phase}$-MoS$_2$ catalyst (0.19 S$^{-1}$) for HER, respectively (Supplementary Table 1). Typically, the amount of hydrogen evolution was measured of 1T$_{0.72}$-MoS$_2$@NiS$_2$ catalyst in 1.0 M KOH solution (Supplementary Fig. 19), presenting HER Faraday efficiency of 97.6 ± 0.6%, owing to the synergistic effect of the phase, defect and interface engineering of electrocatalyst.

Next, we also studied the HER performance of all the as-prepared electrodes in 0.5 M H$_2$SO$_4$ (Fig. 3c). The HER catalytic performance of the 1T$_{0.81}$-MoS$_2$@Ni$_2$P and 1T$_{0.72}$-MoS$_2$@NiS$_2$ electrodes was significantly improved their HER activities according to the LSV data: their overpotential values at 10 mA/cm$^2$ were as low as 38.5 and 152 mV, respectively, which is lower than the values for the electrodes containing 1T$_{0.41}$-MoS$_2$@Ni(OH)$_2$ (236 mV), 1T$_{0.41}$-MoS$_2$ (389 mV), 1T$_{phase}$-MoS$_2$ (392 mV), and 2H$_{phase}$-MoS$_2$ (354 mV). The Tafel slopes for the 1T$_{0.81}$-MoS$_2$@Ni$_2$P and 1T$_{0.72}$-MoS$_2$@NiS$_2$ electrodes were 41 and 42 mV/dec (Supplementary Fig. 20a). These values were lower than the values obtained for 1T$_{0.41}$-MoS$_2$ (169 mV/dec), 1T$_{phase}$-MoS$_2$ (163 mV/dec), and 2H$_{phase}$-MoS$_2$ (189 mV/dec) electrodes and were better than the electrode based on 20 wt% Pt/C (86 mV/dec). It is probably because, in the acidic environment, the H$_2$ desorption is the limiting step because H$^+$ are abundant. The 1T$_{0.81}$-MoS$_2$@Ni$_2$P electrode had a weaker adsorption capacity toward H$_{ads}$ so it exhibits a better catalytic effect than 2H$_{phase}$-MoS$_2$[54]. Meanwhile, compared to the other electrodes, 1T$_{0.81}$-MoS$_2$@Ni$_2$P also has a higher ECSA because it has a larger $C_{dl}$ ($C_{dl}$ = 106.15 mF/cm$^2$, Supplementary Fig. 21) and, as a result, more catalytical sites, which significantly contributed to the overall activity. Furthermore, 1T$_{0.81}$-MoS$_2$@Ni$_2$P also possesses a much smaller $R_{ct}$, in contrast to other electrodes at 300 mV overpotential vs. RHE (Supplementary Fig. 22), revealing

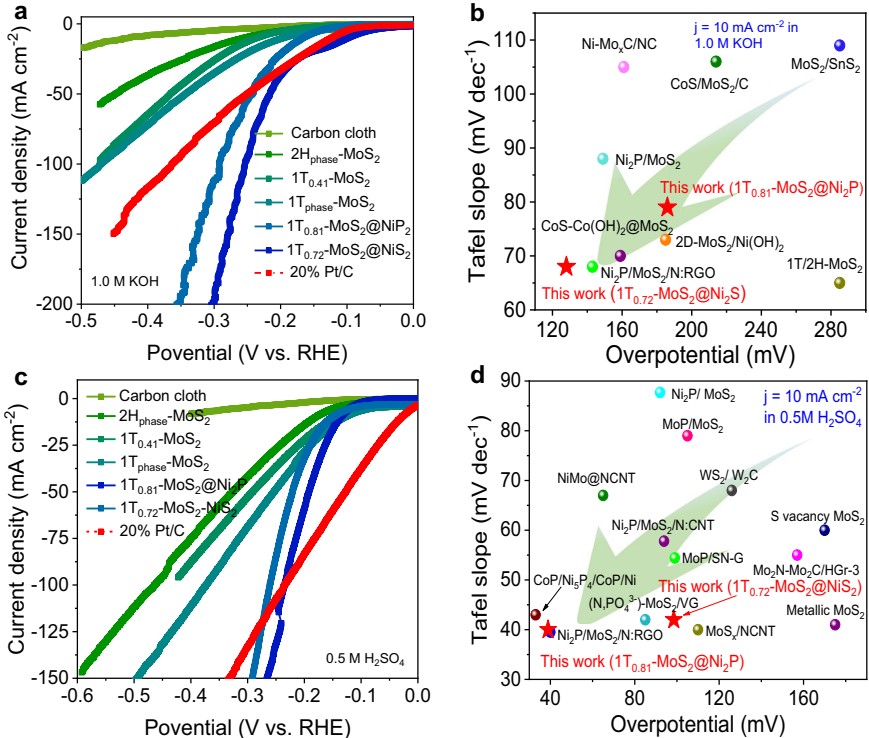

**Fig. 3 HER was performed in alkaline and acidic electrolytes. a** LSV curves in 1 M KOH. **b** $\eta_{10}$ and Tafel slopes for various Mo-based HER electrocatalysts in 1.0 M KOH. **c** LSV curves in 0.5 M $H_2SO_4$. **d** $\eta_{10}$ and Tafel slopes for various Mo-based HER catalysts in 0.5 M $H_2SO_4$. (All LSV curves were corrected without iR-compensation).

satisfied electron transport and good catalytic kinetics, which leads to high activity and low Tafel slope. Supplementary Fig. 20b shows that at 10 and 45 mA/cm², $1T_{0.81}$-MoS$_2$@Ni$_2$P and $1T_{0.72}$-MoS$_2$@NiS$_2$ electrodes were very durable and possesses negligible damping after 16 h measurement, which displays excellent long-term stability. In addition, even after 16 h of a chronoamperometric stability test of the electrodes, the current density remains above 95% (Supplementary Fig. 20c), and there is only a slight deviation for the LSV recorded after the stability test, indicating that as-prepared electrodes have very good stability in an acidic environment. As to 20.0 wt% Pt/C, $1T_{0.72}$-MoS$_2$@NiS$_2$, and $1T_{0.81}$-MoS$_2$@Ni$_2$P electrodes in 0.5 M H$_2$SO$_4$, Supplementary Fig. 20d shows overpotentials vs. various current densities. $1T_{0.81}$-MoS$_2$@Ni$_2$P exhibits lower overpotential. We also compared the overpotentials (at 10 mA/cm² in acidic medium) and Tafel slopes with previously excellent Mo-based electrocatalysts[8,32,37,55–57] (Fig. 3d and Supplementary Table 4). Catalytic HER performance of 1 $T_{0.81}$-MoS$_2$@Ni$_2$P is also superior. Afterward, the amount of hydrogen evolution of $1T_{0.81}$-MoS$_2$@Ni$_2$P catalyst was given in Supplementary Fig. 23, demonstrating a promising Faraday efficiency of 98.7 ± 0.5% towards real water splitting into hydrogen. Based on the above-mentioned results, $1T_{0.81}$-MoS$_2$@Ni$_2$P multi-heterogeneous interface catalyst shows the remarkable intrinsic HER activities in acidic medium mainly attributed to multi-heterointerface interface edges active sites. In addition, as-synthesized $1T_{0.72}$-MoS$_2$@NiS$_2$ (or $1T_{0.81}$-MoS$_2$@Ni$_2$P) catalyst also exhibits excellent OER and overall-water splitting catalytic activity (Please see Supplementary Information for details, Supplementary Figs. 25–27).

**Theoretical calculation and mechanisms analysis of the surface electronic structure and HER activation energy for the as-prepared electrocatalysts**. To explain the distinguished

synergistic effect of $1T_{0.72}$-MoS$_2$@NiS$_2$ (or $1T_{0.81}$-MoS$_2$@Ni$_2$P) multi-heterogeneous interface catalysts, Density functional theory (DFT) calculations were also performed. Model building and computational parameters can be seen in the "Methods" section. Firstly, the interfacial electron interaction was investigated. The charge difference images (Fig. 4a, b and Supplementary Fig. 37) reveal the charge transfer from $1T_{0.41}$-MoS$_2$ to the Ni$_2$S or/and Ni$_2$P interface, and the introduction of 1T-phase is more conducive to charge transfer from MoS$_2$ to NiS$_2$ or Ni$_2$P interface, which significantly increases the interface electron concentration and thus improves its activity. To better understand the surface electronic structure reconfiguration of MoS$_2$ through a coordinated phase transition and interface regulation in theory, the band structure and density of states (DOS) of bare NiS$_2$, Ni$_2$P, 2H$_{phase}$-MoS$_2$, 1T$_{phase}$-MoS$_2$, 2H$_{phase}$-MoS$_2$@NiS$_2$, 2H$_{phase}$-MoS$_2$@Ni$_2$P, 1T$_{phase}$-MoS$_2$@NiS$_2$, and 1T$_{phase}$-MoS$_2$@Ni$_2$P (Fig. 4c–e and Supplementary Figs. 38–40) obtained using the hybrid DFT-HSE06 exchange–correlation functional, which is presented in the Supplementary Information. The calculation results show that the bare NiS$_2$ exhibits typical semiconductor characteristics (Fig. 4c), with a narrow bandgap equal to 0.68 eV (Supplementary Figs. 38 and 39a). The band structure of 1T$_{phase}$-MoS$_2$ (Fig. 4d) and 1T$_{phase}$-MoS$_2$@NiS$_2$ (Fig. 4e) exhibited a certain zero bandgap, indicating a complete transition from the semiconductor phase (0.91 eV) to the metallic phase (0 eV) with improved conductivities[27]. Notably, the intensity of PDOS of 1T$_{phase}$-MoS$_2$@NiS$_2$ was higher than that of 1T$_{phase}$-MoS$_2$ and NiS$_2$ at the Fermi level (Supplementary Figs. 38 and 39). Thus, the electron mobility of the 1T$_{phase}$-MoS$_2$@NiS$_2$ catalysts was more favorable for the efficient charge transfer, which agrees consistent with the EIS test results[58]. Moreover, the PDOS results imply that the NiS$_2$ interface hybrid generates some new interface electronic states in 1T$_{phase}$-MoS$_2$ (Supplementary Fig. 39c), which was very likely because of the hybridization of the *d*-orbital of Mo

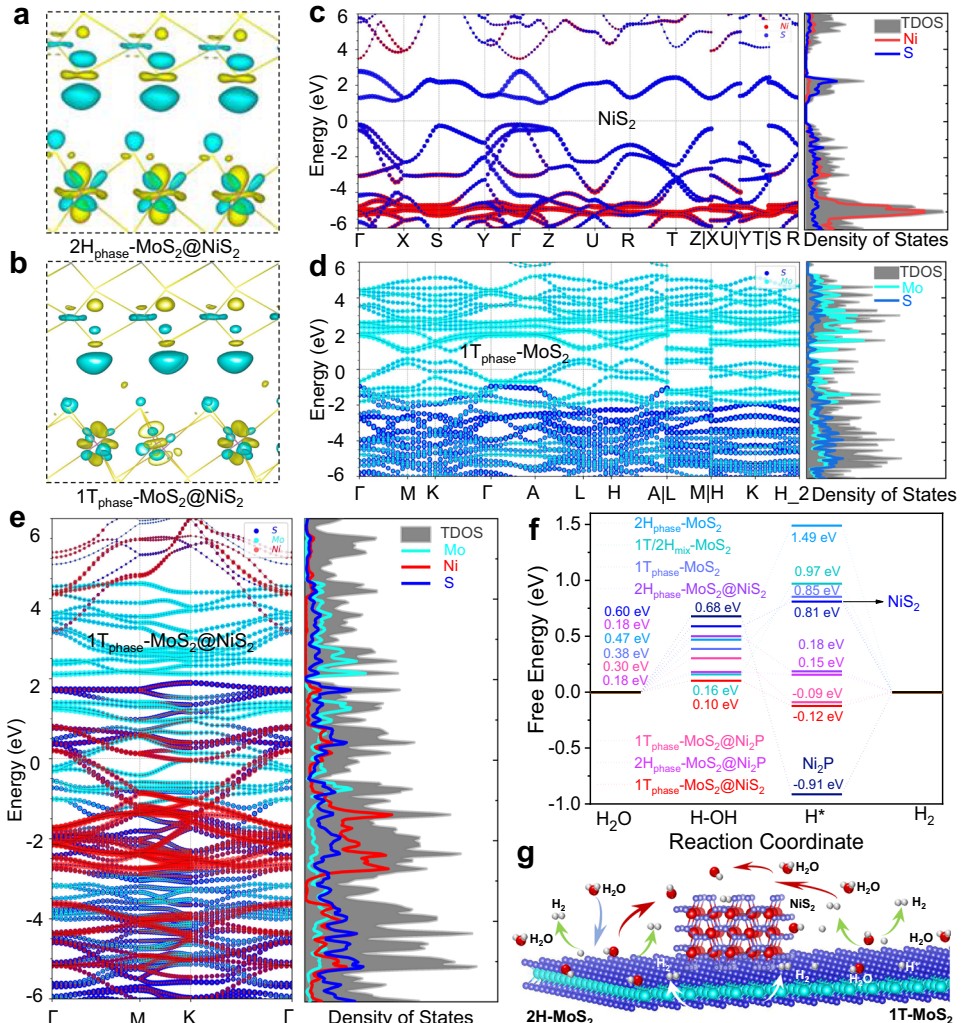

**Fig. 4 Theoretical calculation and mechanisms analysis of the surface structure and HER activation energy of the as-prepared electrocatalysts.** the deformation of the electronic density of **a** $2H_{phase}$-$MoS_2$@$NiS_2$ and **b** $1T_{phase}$-$MoS_2$@$NiS_2$ interface, in which yellow/green isosurfaces correspond to positive/negative spin densities (0.00295308 e/Å³). Band structure and density of states (DOS) for **c** $NiS_2$, **d** $1T_{phase}$-$MoS_2$ and **e** $1T_{phase}$-$MoS_2$@$NiS_2$. **f** Free-energy diagrams for HER on the $2H_{phase}$-$MoS_2$, $1T_{phase}$-$MoS_2$, pure $Ni_2P$, pure $NiS_2$, $1T/2H_{mix}$-$MoS_2$, $2H_{phase}$-$MoS_2$@$Ni_2P$, $2H_{phase}$-$MoS_2$@$NiS_2$, $1T_{phase}$-$MoS_2$@$NiS_2$ and $1T_{phase}$-$MoS_2$@$Ni_2P$ interface edge. **g** Schematics showing water activation, *H intermediate formation and hydrogen generation on multi-heterojunction electrocatalysts.

and an empty *d*-orbital of Ni. Thus, higher HER activity of $1T_{phase}$-$MoS_2$@$NiS_2$ in comparison to $1T_{phase}$-$MoS_2$ agrees with the Fermi level DOS (Fig. 4d, e). Thus, the actual electrochemical performance would show even faster conductivity and charge transfer kinetics.

To reveal further the relationship of HER activity of catalysts with phase structure and heterojunction-interface, we used DFT to calculate the optimized structures and free-energy diagrams for HER on $2H_{phase}$-$MoS_2$, $1T_{phase}$-$MoS_2$, $1T/2H_{mix}$-$MoS_2$, pure $Ni_2P$, pure $NiS_2$, $2H_{phase}$-$MoS_2$@$NiS_2$, $2H_{phase}$-$MoS_2$@$Ni_2P$, $1T_{phase}$-$MoS_2$@$NiS_2$, and $1T_{phase}$-$MoS_2$@$Ni_2P$ catalysts with partially multi-heterojunction interface modification. As shown in Fig. 4f and Supplementary Fig. 42, the reaction pathway for alkaline HER is constructed[59,60], including prior $H_2O$ dissociation to form H* intermediates (Volmer step) and hydrogen generation (Tafel step or Heyrovsky step). However, the energy of the intermediate state H*($\Delta G_{H*}$) is a critical indicator of the ability of hydrogen evolution (Tafel step or Heyrovsky step)[35,59]. Figure 4f displays the calculated free-energy diagram on the most stable energy of the $2H_{phase}$-$MoS_2$, $1T_{phase}$-$MoS_2$, $Ni_2P$, $NiS_2$, $2H_{phase}$-$MoS_2$@$Ni_2P$, $2H_{phase}$-$MoS_2$@$NiS_2$, $1T_{phase}$-$MoS_2$@$NiS_2$,

and $1T_{phase}$-$MoS_2$@$Ni_2P$ catalysts (Supplementary Fig. 41). For $2H_{phase}$-$MoS_2$, the $\Delta G_{H*}$ is very positive (1.49 eV), indicating that there is a strong interaction between H* and $2H_{phase}$-$MoS_2$, showing poor HER reaction kinetics. More importantly, $MoS_2$ shows unfavorable catalyst-$OH_{ad}$ energetics ($\Delta G_{H2O} = 0.82$ eV), suggesting that the relatively high activated $H_2O$-adsorption energy will hinder the decomposition of $H_2O$ into H* intermediates and results in slow HER kinetics. The introduction of the $1T/2H_{mix}$-phase into $MoS_2$ can obviously decrease the value of $\Delta G_{H*}$ to 0.97 eV and $\Delta G_{H2O}$ to 0.16 eV, implying promoted HER activity compared to $2H_{phase}$-$MoS_2$. Notably, constructing multi-heterointerface interface edges active sites with $NiS_2$ can provide the active sites for –OH adsorption, and the followed $\Delta G_{H2O}$ and $\Delta G_{H*}$ are decreased to 0.10 and −0.12 eV on the $1T_{phase}$-$MoS_2$@$NiS_2$ interface, indicating the 1T/$2H_{mix}$-phase and $NiS_2$ nanoparticles are effective for cleaving HO–H bonds and weaker interaction between H*. Also, the charge transfer from $Ni_2P$ to the $MoS_2$ is verified by the DFT calculations, and hence there is a more optimal $\Delta G_{H*}$ value of about −0.09 eV. Hence, the $NiS_2$ (or $Ni_2P$) can act as a promoter of $H_2O$ dissociation and form hydrogen intermediates which then

adsorb on nearby $MoS_2$ catalyst sites. In this way, the multi-heterointerface can also accelerate the subsequent generation of $H_2$. The reaction pathways on the single side (such as $Ni_2P$, $NiS_2$, and $MoS_2$) of the interface have also been shown in Fig. 4f and Supplementary Fig. 42. These both show there is more unfavorable energetics than that of the synergetic pathway on $MoS_2@NiS_2$ or $MoS_2@Ni_2P$ interface. The reason is that H* adsorbed on the surface of $2H_{phase}$-$MoS_2$ bounds to Mo atoms, and strong Mo–H strength and poor conductivity. However, H* can be absorbed not only by the $1T_{phase}$-$MoS_2@NiS_2$ surface. Ni atoms possess empty $d$ orbitals capable of binding H atoms, thereby weakening the Mo–H strength. More importantly, the introduction of the 1T-phase not only increases its electrical conductivity but also creates abundant active sites at the multi-heterojunction interface edges, which synergistically promote HER activity (Fig. 4g). Thus, our work demonstrates a novel and efficient design to create multi-heterogeneous interfacial electrocatalysts without noble metal materials and with excellent HER activity.

**In situ electrochemical-Raman spectroscopy.** To better understand the active sites of the $1T_{0.81}$-$MoS_2@Ni_2P$ and $1T_{0.72}$-$MoS_2@NiS_2$ electrodes during the HER process, we used in situ Raman spectroscopy to study it (Supplementary Fig. 46). The in situ Raman test system is shown in Supplementary Fig. 46a. The as-prepared sample, Ag/AgCl, and Pt wires were used as working electrode, reference electrode, and counter electrode, respectively. In addition, the electrolyte was 1.0 M KOH. As shown in Supplementary Fig. 46c, d, the Raman spectra of $1T_{0.72}$-$MoS_2@NiS_2$ electrode collected under potentiostatic conditions at stepped potential values from 0 V to −1.5 V. The results show that three characteristic peaks (147.3, 235.4, and 335.2 $cm^{-1}$ are attributed to $J_1$, $J_2$, and $J_3$ vibrations) of 1T-$MoS_2$, two characteristic peaks (382 and 407 $cm^{-1}$ are attributed to the $E_{2g}^1$ and $A_{1g}$ vibrational bands) of 2H-$MoS_2$, and a vibrational peak (437.3 $cm^{-1}$) of Ni–S. However, when the $1T_{0.72}$-$MoS_2@NiS_2$ sample was put in the electrolyte solution (1.0 M KOH solution), at different applied voltages from the −0.4 to −1.5 V during electrocatalytic HER, these Raman peaks are significantly enhanced. In addition, many new peaks aroused at 152, 188, 222, 284, 322, 453, 500, 548 $cm^{-1}$ of $MoS_2$, respectively[61]. The changes of these Raman peaks indicate that new chemical bonds are formed between our samples and the functional groups of –OH, $H^+$, and $H_2O$ molecules in the electrolyte, suggesting that it has a strong absorption capacity of ions and $H_2O$ molecules. In addition, it can be observed for two slight new peaks of 429 and 488 $cm^{-1}$ under the bias potential of −0.4 V (Supplementary Fig. 46d), corresponding to the $\nu_{Ni-OH}$ band of our samples. This result indicates the adsorbed $H_2O$ molecules during the cathodic polarization process are decomposed into $H_{ads}$ species and $OH^-$ ions[62]. More importantly, the intensity of these characteristic peaks of $1T_{0.72}$-$MoS_2@NiS_2$ are increased significantly as the potential varies from −0.4 to −1.5 V. It may be due to the $OH^-$ being driven to adsorb on Mo, Ni, S atoms in the alkaline medium, and then OOH* intermediates are formed[63]. As to $1T_{0.81}$-$MoS_2@Ni_2P$ sample, we also obtained similar results (Supplementary Fig. 46e, f). We used in situ growth of $NiS_2$ (or $Ni_2P$) nanoparticles on the entire surface of 1T-2H $MoS_2$ microspheres to construct multi-heterojunction interface, which may generate Ni–Mo metal bonds, thereby increasing the number of effective active sites of the catalyst. Due to the introduction of Ni atoms, there is a strong interaction between Ni and Mo atoms on the surface of the catalyst, thereby increasing the local electronic state of Mo atoms, reducing the hydrogen-adsorption energy of the $H^+$ on Mo atoms, and thus improving its intrinsic catalytic activity.

**X-ray absorption spectroscopy.** To investigate electronic states of catalysts, X-ray absorption near-edge structure (XANES) spectra were measured on the fresh catalysts and those after being used in the HER process at three representative potentials (−0.04, −0.1, and −0.2 V), near the onset potential and the overpotential at the current densities of 5 and 10 mA $cm^{-2}$ (for $1T_{0.81}$-$MoS_2@Ni_2P$ sample), respectively. Figure 5a presents Ni $K$-edge XANES spectra of $1T_{0.81}$-$MoS_2@Ni_2P$ catalyst recorded at different applied potentials and reference spectra of Ni foil, NiO, and $NiO_2$. From the fresh catalyst to that under the −0.04 V potential condition, the absorption edge is shifted to the lower energy side by ~0.5 eV, along with a broadening of the white-line peak, meaning a decrease of the Ni oxidation state. Moreover, when cathodic potentials of −0.04 and −0.1 V versus RHE were applied, a further shift of the absorption edge towards lower energy by ~0.2 eV occurs in relation to the case under the −0.04 V potential condition, implying a distinct decrease in the Ni valence state in $1T_{0.81}$-$MoS_2@Ni_2P$ during the HER. Notably, all catalyst spectra exhibit white line at 8350.2-8350.9 eV (Supplementary Fig. 47), corresponding to the $1s$ to $4p$ electronic transition, indicating Ni–O local coordination similar to NiOOH and Ni oxides[64]. Using the edge positions of NiO and $NiO_2$ as references, the Ni average valence state is determined as +3.3, +2.2, +1.8, and +2.0 for the fresh and those used at −0.04, −0.1, and −0.2 V, respectively (Supplementary Table 9). Therefore, Ni cations are reduced under working conditions, which is consistent with the Ni $2p$ XPS results (Supplementary Fig. 28d). In addition, $1T_{0.72}$-$MoS_2@NiS_2$ (Supplementary Fig. 48) and $1T_{0.41}$-$MoS_2@$-$Ni(OH)_2$ (Supplementary Fig. 49) show similar behavior as Ni valency is decreased from +3.6 (fresh) to +2.4 (−0.1 V) and from +2.7 (fresh) to 1.8 (−0.2 V), respectively. In addition, oxidation of Mo from +4 to +6 after catalysis is observed as shown in Fig. 5b. The fresh $1T_{0.81}$-$MoS_2@Ni_2P$ exhibit much broader Mo $L_3$ XANES absorption than those of the Mo standards ($MoS_2$, $MoO_2$, and $MoO_3$), and its lower edge position indicates reduced Mo admixture. In most cases, the broadening of XANES relates to a lack of crystallinity. Interestingly, after the HER reaction, the broad peak is shifted to higher energy and split, indicating $4dt_{2g}$ and $4de_g$ absorption bands of $MoO_3$[65] located at 2524 and 2526 eV, respectively. These results indicate the valence states of Mo cations are increased from approximately +4 to a higher oxidation state (+6) under working conditions. Moreover, sulfur does not take part in catalysis as shown in Fig. 5c. All S $K$-edge XANES spectra of catalysts before and after the reaction are similar and agree well with that of $MoS_2$ standard. P $K$-edge XANES spectra of $1T_{0.81}$-$MoS_2@Ni_2P$ and $FePO_4$ are shown in Fig. 5d. The white line at 2154 eV belongs to $PO_4^{2-}$ associated with hybridized O $2p$- P $3p$ absorption band[66] whereas the original $Ni_2P$ species appear only as a minor peak at 2146 eV. The peak position and absorption line shape of $Ni_2P$ are close to that of $Co_2P$ indicates the valence state $P^{3-}$[67]. The ratio of $Ni_2P$ to $PO_4^{2-}$ changes with the cathodic potential, and the highest ratio is 1:3 at −0.2 V. As to the fresh catalyst, the front peak is much weaker, which indicates that although there is a Ni–P bond, its signal is changed by the presence of some other elements. When cathodic potentials of −0.1 and −0.2 V versus RHE were applied, showing the highest intensity of the front peak, which indicates that both P and Ni are involved in the reaction during the HER process.

Overall, the XANES spectra studies provide clear evidence that the structures of as-prepared catalysts can drastically change under realistic catalytic conditions. The Ni site at the interface of heterojunction is most susceptible to low-valence induced by chemisorbed $OH^-$ under electrochemical conditions. While the valence state of the Mo site at the interface increases, suggesting that the charge transfer (electron transfer from the Mo site to the

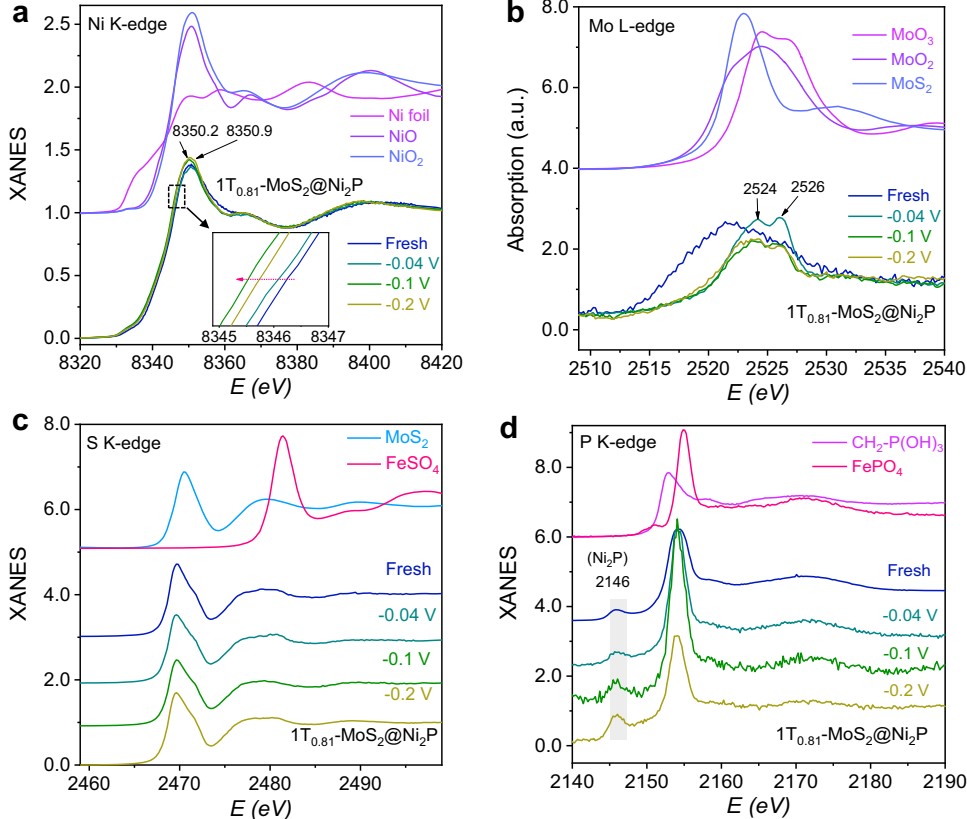

**Fig. 5 XANES spectra measurements of fresh $1T_{0.81}$-MoS$_2$@Ni$_2$P catalyst and after being used in the HER process at −0.04, −0.1, and −0.2 V vs RHE, respectively. a** Ni $K$-edge XANES spectra of $1T_{0.81}$-MoS$_2$@Ni$_2$P catalyst and standards Ni foil, NiO, and NiO$_2$. Inset, Magnified pre-edge XANES region. **b** Mo $L$-edge XANES spectra of $1T_{0.81}$-MoS$_2$@Ni$_2$P catalyst and standards MoS$_2$, MoO$_2$, and MoO$_3$. **c** S $K$-edge XANES spectra of $1T_{0.81}$-MoS$_2$@Ni$_2$P catalyst and standards MoS$_2$, and FeSO$_4$. **d** P $K$-edge XANES spectra of $1T_{0.81}$-MoS$_2$@Ni$_2$P catalyst and standards CH$_2$-P(OH)$_3$, and FeSO$_4$.

Ni site) on the surface of the heterojunction catalyst is accelerated during the HER reaction. Therefore, reduced nickel possesses empty $d$-orbitals, which is beneficial to additional H binding ability. Moreover, it can decrease Mo–H bond strength, and so greatly enhance the HER catalytic activity of as-prepared catalysts.

## Discussion

In summary, we have constructed multi-heterogeneous-interface catalysts ($1T_{0.81}$-MoS$_2$@Ni$_2$P and $1T_{0.72}$-MoS$_2$@NiS$_2$) by tuning its electronic structure of phase modulation synergistic with interfacial chemistry and defects to phosphorus or sulfur implantation strategies, which is an efficient approach to obtain abundant reactive sites of long-cycling and stable electrocatalysts for HER in alkaline and acidic surroundings. The as-achieved $1T_{0.81}$-MoS$_2$@Ni$_2$P and $1T_{0.72}$-MoS$_2$@NiS$_2$ electrodes only require small overpotentials of 38.9 (or 186) and 98.5 (or 128) mV to drive HER at 10 mA/cm$^2$ and have low Tafel slopes: 41 (or 79) and 42 (or 68) mV/dec in 0.5 M H$_2$SO$_4$ (or 1.0 M KOH). Accordingly, these results show varieties of multi-heterogeneous interfaces in $1T_{0.81}$-MoS$_2$@Ni$_2$P and $1T_{0.72}$-MoS$_2$@NiS$_2$ electrodes, which can be considered versatile electroactive sites and facilitate electron transfer because of their unique heterogeneous effects. DFT calculation results also display that the introduction of metallic-phase MoS$_2$ and intrinsic HER-active Ni-based materials can regulate MoS$_2$ electronic structure effectively for making the bandgap narrower. In situ electrochemical-Raman spectroscopy indicates that the OH$^−$ ions are driven to be adsorbed on Mo, Ni atoms in the alkaline medium, and then there form the OOH* intermediates. There is a strong interaction

between Ni and Mo on the surface of the catalyst, thereby increasing the local electronic state of Mo atoms, reducing the hydrogen-adsorption energy for protons on Mo atoms, and thus improving its intrinsic catalytic. Additionally, XANES spectroscopy results imply that reduced Ni supply empty $d$-orbitals to facilitate H atom capture, and decrease Mo–H strength of $1T_{0.81}$-MoS$_2$@Ni$_2$P (or $1T_{0.72}$-MoS$_2$@NiS$_2$) catalysts that account for the outstanding HER properties with lower Tafel slopes and overpotentials compared with $2H_{phase}$-MoS$_2$, $1T_{phase}$-MoS$_2$ counterparts and other Mo-based catalysts. Thus, our work provides a new horizon for rationally designing multi-heterogeneous interfaces of non-precious electrocatalysts to realize excellent HER activities.

## Methods

**Synthesis of $1T_{0.41}$-MoS$_2$.** MoS$_2$ microspheres were grown on carbon cloth (CC) hydrothermally. First, a CC (2 × 4 cm) was cleaned (for 15 min) using acetone and then sonicated in deionized water and ethanol for 10 min. Then, sodium molybdate (Na$_2$MoO$_4$·2H$_2$O, 411.9 mg) and thiourea (CS(NH$_2$)$_2$, 608.96 mg) were added to deionized water (40 mL) and citric acid (20 mL). The mixture was magnetically stirred to form a cleaning solution, then placed into a 100 mL Teflon-lined autoclave and held in it at 180 °C for 12 h. Finally, the CC substrates with $1T_{0.41}$-MoS$_2$ microspheres (denoted through the paper as $1T_{0.41}$-MoS$_2$) were rinsed using deionized water and ethanol and vacuum-dried for 6 h at 60 °C. For comparison, deionized water was used as the solvent, and $2H_{phase}$-MoS$_2$ microspheres were synthesized hydrothermally at 220 °C for 24 h from the same precursors.

**Synthesis of $1T_{phase}$-MoS$_2$.** We used Li-intercalated bulk MoS$_2$ to prepare $1T_{phase}$-MoS$_2$[68]. In an Ar-filled glove box, bulk MoS$_2$ (1.0 g) prepared by stripping were dispersed in 15 mL of 2 M $n$-BuLi/hexane solution and stirred at ambient conditions for 48 h. The resulting black materials were repeatedly rinsed with anhydrous $n$-hexane and then centrifuged to eliminate $n$-butyl lithium excess and other solution impurities. The $1T_{phase}$-MoS$_2$ powder was prepared and was then coated

on the CC substrate. In order to promote better contact between $1T_{phase}$-$MoS_2$ and CC substrate, we annealed (500 °C) the CC loaded with $1T_{phase}$-$MoS_2$ sample under the protection of Ar gas.

**Synthesis of $1T_{0.41}$-MoS2@Ni(OH)$_2$.** We use a standard three-electrode system to prepare $1T_{0.41}$-$MoS_2$@$Ni(OH)_2$. $1T_{0.41}$-$MoS_2$ acted as a working electrode, while Pt sheet and Ag/AgCl/3.5 M KCl acted as counter and reference electrodes. $Ni(OH)_2$ was electrodeposited on the $1T_{0.41}$-$MoS_2$ using 0.1 M $Ni(NO_3)_2$ at 5.0 mA/cm$^2$ cathode current density applied for 300 s. $1T_{0.41}$-$MoS_2$@$Ni(OH)_2$ samples were rinsed with deionized water and ethanol several times and vacuum-dried at 60 °C.

**Synthesis of $1T_{0.72}$-MoS2@NiS$_2$.** The $1T_{0.72}$-$MoS_2$@$NiS_2$ multi-heterogeneous interfaces were prepared by the solid-vapor reaction method. First, a piece of $1T_{0.41}$-$MoS_2$@$Ni(OH)_2$ grew on CC was put into the quartz tube with 32.0 mg S powder and was then sealed. Secondly, the quartz tube was positioned inside a tube furnace and was calcinated at 500 °C for 60 min to obtain a $1T_{0.72}$-$MoS_2$@$NiS_2$ electrode.

**Synthesis of $1T_{0.81}$-MoS2@Ni$_2$P.** Similarly, the $1T_{0.81}$-$MoS_2$@$Ni_2P$ multi-heterogeneous interfaces were also obtained by the solid-vapor reaction method. First, a piece of $1T_{0.41}$-$MoS_2$@$Ni(OH)_2$ grew on CC was put into the quartz tube with 31.0 mg red phosphorus and was then sealed. Secondly, the quartz tube was also calcinated at 580 °C for 1.0 h to prepare the $1T_{0.81}$-$MoS_2$@$Ni_2P$ electrode. Additionally, 20 wt% Pt/C was also coated on CC substrate (2.0 mg/cm$^2$) and was labeled as 20% Pt/C for comparison.

**Materials characterization**. All as-synthetized electrodes were characterized by XRD (performed by Bruker D8 Advance instrument) and Raman spectroscopy (performed using Horiba LabRAB HR800 instrument). The sample morphologies were studied using SEM performed by Hitachi SU8010 instrument and TEM (performed by FEI Tecnai F30 instrument). XPS spectra were collected by the ESCALAB 250Xi instrument manufactured by ThermoFisher using Al Kα radiation.

**Electrochemical measurements**. All electrochemical measurements were performed with a CHI 660E Electrochemical Workstation (CHI Instruments, Shanghai Chenhua Instrument Corp., China). The HER performance of different catalysts (1.0 cm$^2$) was characterized using a three-electrode electrochemical cell in $N_2$-saturated 1.0 M KOH and 0.5 M $H_2SO_4$ electrolyte, respectively. Before testing the polarization curve, we first perform cyclic voltammetry (CV) for more than 20 cycles to activate the as-prepared catalysts with a scan rate of 50 mV/s. The EIS tests were measured by AC impedance spectroscopy at the frequency ranges $10^6$ to 1.0 Hz at 300 mV. According to the Nernst equation ($E_{RHE} = E_{Hg/HgO} + 0.059$ pH $+ 0.098$), where $E_{RHE}$ was the potential vs. a reversible hydrogen potential, $E_{Hg/HgO}$ was the potential vs. Hg/HgO electrode, and pH was the pH value of electrolyte. The electrochemical stability was evaluated by chronoamperometry measurements at a static overpotential, during which the current variation with time was recorded. The ECSA values were measured through CV in the selected non-faradaic range. The current densities have a linear relationship against different scan rates (10–60 mV/s) and the values of the slop were considered as twice of $C_{dl}$. The Faraday efficiency of the as-fabricated electrodes was determined by the water drainage method, which can be found in Supplementary Note 2 in Supplementary Information for details. The OER tests were performed in $O_2$-saturated 1.0 M KOH solution, and the others are the same as HER test conditions. The overall-water splitting performance was characterized in 1.0 M KOH using a two-electrode configuration, and the polarization curve was recorded at a scan rate of 5 mV/s. In order to better compare, Pt/C and $IrO_2$ ink were also synthesized by placing 8 mg Pt/C and 8 mg $IrO_2$ powder in the mixture of 700 μL ethanol, 300 μL deionized water, and 50 μL Nafion followed by ultrasonication for 30 min, respectively. Then the as-obtained ink was coated onto the carbon cloth (CC) with the loading mass density of about 3.0 mg/cm$^2$ and was then dried at 60 °C. The long-term stability measurements were carried out using the chronoamperometry measurements. All polarization curves at 5 mV/s were corrected without iR-compensation.

**DFT theoretical calculation**

*Model building*. According to the HRTEM micrographs, $1T_{phase}$-$MoS_2$, $2H_{phase}$-$MoS_2$, $Ni_2P$, and $NiS_2$ formed a multiphase heterojunction. $1T_{phase}$-$MoS_2$@$Ni_2P$ interface, $1T_{phase}$-$MoS_2$@$NiS_2$ interface, $2H_{phase}$-$MoS_2$@$Ni_2P$ interface, $2H_{phase}$-$MoS_2$@$NiS_2$ interface, bulk $1T_{phase}$-$MoS_2$, $2H_{phase}$-$MoS_2$ were also constructed as comparisons. Considering the Van der Waals forces between the two phases, the unrelaxed heterojunction interface distance was set to 3.0 Å. These original structures were obtained from Materials Project Database[69].

*Computational parameters*. DFT calculation was applied to calculate electronic structures of two crystal structures by the partial augmented plane-wave method (PAW) implemented in the VASP[70] using VASPKIT code for post-processing. Considering the heterojunction structure, the long-range force correction was

considered by using the DFT-D3 correction method of Grimme[71]. The Perdew–Burke–Ernzerhof (PBE) generalized gradient approximation[72] was implemented for exchange-correlation energy calculations using 550 eV kinetic energy cut off for the plane-wave basis. Then structural optimizations using a conjugate gradient (CG) method based on the pre-optimized structure were repeated until the maximum force component on each atom remained below 0.01 eV/Å. Monkhorst-Pack k-point meshes in the first Brillouin zone of the primitive cell were used the VASPKIT code recommended accuracy levels of 0.04 for the optimization calculation and 0.02 for the static calculation, respectively. After fully relaxing the structures, one final (electronic scf) step with the tetrahedron method using Blöchl corrections and denser k-meshes was employed for DOS calculation. In addition to the H adsorbed energy calculations, the frequency calculation of free H and free-energy correction at 298.15 K (including the entropy and zero-point energy contributions) were also calculated. To avoid abnormal entropy contribution, frequencies < 50 cm$^{-1}$ are set to be 50 cm$^{-1}$.

**XANES spectra measurements**. The Mo $L_3$-edge, P, S, and Ni $K$-edge XANES spectra were measured at the BL8 beamline of Synchrotron Light Research Institute (SLRI), Thailand[73]. The SLRI storage ring was operated at 1.2 GeV with an electron current of 80–150 mA. The incident X-ray beam was monochromatized with a double-crystal monochromator equipped with InSb (111) and Ge (220) crystals. XANES measurements were carried out in air at Ni K-edge and under He atmosphere at the lower edges on as-prepared catalysts embedded on carbon cloth and those used as the working electrode in electrochemical reaction with 1 M KOH solution. Cathode voltages from −0.04 V to −0.2 V vs. RHE ($E_{RHE} = E_{Ag/AgCl} + 0.059$ pH $+ 0.197$) were applied for 160 s using an electrochemical workstation (Autolab PGSTAT204) before the XANES experiment. All XANES spectra were collected in fluorescence-yield mode using a 13-element Si drift detector. Ni and foils, and elemental S and P were used for photon energy calibration. The edge position was defined as the point corresponding to the maximum value in the derivative curves of the XANES spectra. Data normalization was carried out using the Athena software[74].

## Data availability

The authors declare that the main data supporting the findings of this study are available within the article and its Supplementary Information. Extra data are available from the corresponding author upon request.

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

## Acknowledgements

We thank Dr. Yuting Luo, Feng-Ning Yang, Zhiyuan Zhang, and Jun-Rong Zeng for the helpful discussions. This work was supported by the Science Foundation of the National Key Laboratory of Science and Technology on Advanced Composites in Special Environments (Grant No. 6142905192507), Shenzhen Science and Technology Plan Supported Project (Grant No. JCYJ20170413105844696), China Scholarship Council (Grant No. 201606125092), the National Key R&D Project from Minister of Science and Technology in China (No. 2016YFA0202701), the University of Chinese Academy of Sciences (Grant No. Y8540XX2D2), the National Natural Science Foundation of China (No. 52072041), External Cooperation Program of BIC, Chinese Academy of Sciences (No. 121411KYS820150028), the 2015 Annual Beijing Talents Fund (No. 2015000021223ZK32), and Qingdao National Laboratory for Marine Science and Technology (No. 2017ASKJ01). XANES experiment was supported by Synchrotron Light Research Institute (SLRI), Suranaree University of Technology (SUT), and Thailand Science Research and Innovation (TSRI). Dr. Wipada Senanon and BL8 staffs are acknowledged for their assistances.

## Author contributions

M.L. and J.W. contributed equally to this work. G.-G.W. and Y.Y. supervised the project. M.L. designed, performed, and analyzed the experiments and devised the heterogeneous-interface catalysts; J.W. conducted theoretical calculation section; W.K. and S.S. performed XANES spectra measurements; W.K. revised XANES interpretation; F.L., Y.C., F.Z., and J.Y. discussed the results and helped the preparation of figures, which were revised by Y.Y. The manuscript was written by M.L., and revised by J.W., G.-G.W., and Y.Y.

## Competing interests

The authors declare no competing interests.
