## [Peer Review File · Nature Communications]

REVIEWER COMMENTS

Reviewer #1 (Remarks to the Author):

The authors reported non noble metal based electrocatalysts for hydrogen evolution reaction in acidic and alkaline medium. They investigated MoS₂/Ni₂P and MoS₂/Ni₂S materials. The main point in the manuscript is the discussion on phase transition (2H to 1T), interface between heterostructures (between MoS₂ and Ni₂S; between MoS₂ and Ni₂P) and defects. While examples of these strategies have been already reported (Adv. Mater. 2018, 30, 1705509; Nature Commun., 2020 11:5462), the combination of the proposed materials is new and timely, and can raise the strong attention of the broad readership of Nature Commun.

The results are very interesting and the conclusions are solidly supported by the experimental findings. For all these reasons, I suggest to accept the manuscript for publication, pending minor revisions.

Specific comments are given below:

1. Some in-situ techniques that offer direct evidence on the nature of active sites are suggested. Where do the authors suggest the location of the active sites in this study? How do the electronic states of Ni or Mo at the interface behave during catalysis?
2. What is the catalytic efficiency of the materials in terms of Faradaic efficiency and turnover frequency?
3. The authors reported the current density normalized with respect to geometric areas of the working electrode. It is important to investigate the intrinsic activity by normalizing with Cdl, TOF... A comparison with other literature would be beneficial.
4. On page-10, line 212, the authors claim the presence of S vacancy (in fact they mentioned S defect) from HRTEM observation. It is difficult to unequivocally associate this defect to S vacancy unless supported by other evidence.
5. The discussion on XPS result should be polished. For example,
 - On page-12, line 250, the authors state: "Both peaks were slightly shifted (by 0.5 eV) towards higher binding energies". What is the reference for this shifting? Did the authors compare with Ni 2p spectrum of pure Ni₂S or Ni₂P?
 - I am a bit surprised to see the Ni-P type bonding correlated to the peak at around 135 in XPS spectra of P2P (Fig.S7). This peak is meant for surface oxidized phosphate. The metal -phosphide bond appears at BE around 129 or 130. May the authors comment on that?
6. I would suggest to include in Figure S4a the standard patterns of MoS₂, Ni₂P, Ni₂S, to offer ease of comparison on the presence of suggested crystalline phases. It is also important to include/discuss the XRD pattern and Raman spectra of pure 1T MoS₂.
7. It is well documented that the metallic 1T phase delivers fast electron transfer during HER, the authors also mentioned this. However, we see that the 2H phase is performing better than the 1T phase in Figure 3a and c. A comment from the authors on this would be beneficial.
8. Detailed procedure of the electrochemical measurements should be given: was there surface activation? At what voltage did the authors measure the EIS?
9. This study seems based purely on electrocatalysis not on photo-electrocatalysis. So, what is the relevance of light on/off during the EIS study?
10. How about the OER and overall water splitting activities of 1T0.72-MoS₂@Ni₂S and 1T0.81-MoS₂@Ni₂P?
11. I am wondering if there could be leaching of some components (phosphides, sulfides..) during catalysis especially in alkaline medium. May the authors conduct some post catalytic characterizations in this direction?
12. Figure S2 a and b are identical. Please cancel one of the two.
13. Figures 4,5 show that pure 1T phase performs less than mixed phase (1T0.41-MoS₂). However,

the DFT ignores this mixed phase simulations and only considered pure 1T or pure 2H phases. In fact, one of the main concepts elucidated in the experiment is "multi-heterogeneous-interface". Is there any special reason (maybe too complicate or too heavy computational time required) for the omission of mixed phase (1T and 2H) in the DFT study?

Reviewer #2 (Remarks to the Author):

This work proposes a synergistic strategy for the design of efficient Mo-based HER electrocatalyst. By combining phase transition, defect engineering as well as interface engineering together, a series of multi-heterojunction-interface electrocatalysts have been synthesized (e.g., 1T0.81-MoS₂@Ni₂P and 1T0.72-MoS₂@NiS₂), yielding highly improved hydrogen evolution reaction (HER) performance. In particular, the 1T0.81-MoS₂@Ni₂P or 1T0.72-MoS₂@NiS₂ catalysts with heterogeneous-phase interfaces possess a low onset overpotential and Tafel slope, comparative to many other recently-reported Mo-based catalysts. I think this work caters to the readership in electrocatalysis and electrochemistry fields and will arouse the interests in interface engineering induced electronic structure modulation for HER performance improvement. Thus, I recommend that this manuscript should be published in Nature Communications after addressing the following minor revisions:

1. The authors declared the sulfur or phosphorus vapor thermal treatment can improve the conversion rate of 2H to 1T phase of MoS₂ substantially from 41% to over 70%. Then, what's the possible mechanism governing such interesting phenomenon? Can the authors provide some personal insights about it from phase free-energy change or other theoretical perspectives?
2. In the theoretical calculation and mechanism analysis part, the authors only carried out hydrogen adsorption energy (ΔG_{H^*}) calculation which is mainly effective for HER property evaluation in acid media. However, as the water dissociation step usually takes a dominant role in alkaline HER process, it's necessary to carefully analyze the concrete water dissociation, subsequent intermediate evolution steps as well as the interface engineering effect on modulating these steps, including the water dissociation energy barrier, the explicit water dissociation related active sites and rate-limiting step for the whole alkaline HER process.
3. In the mechanism analysis part, the authors focused on the local electronic structure change at MoS₂-NiS₂ heterojunction interface. To go a little further, what about the concrete effects of local atomic sites or electronic structure at 1T-2H MoS₂ homojunction interface which also exists in the system? Can the authors provide some corresponding explanation by experiment or simulation?
4. As to the HER mechanism, the as-generated Hads could be adsorbed on 1T0.81-MoS₂@Ni₂P (and 1T0.72-MoS₂@NiS₂) and further be converted into H₂ readily through the Tafel step (Page 17, line 12-22). However, the free energy of NiS₂ and Ni₂P for hydrogen adsorption was not calculated in the diagram (Fig. 4f).
5. Was the computation for NiS₂ spin-polarized? If so, please provide another orbital for your computation of DOS/energy band.
6. As we seen, under low current density conditions, 1T0.81-MoS₂@Ni₂P and 1T0.72-MoS₂@NiS₂ electrodes exhibit excellent HER performance; however, under high current density conditions (such as more than 250 mA cm⁻²), whether they still have excellent HER performance? The authors should add this part of the data to further explain it.

7. In this manuscript, 1T0.81-MoS₂@Ni₂P and 1T0.72-MoS₂@NiS₂ electrodes exhibit the fastest HER processes and better reactivity which is attributed to the multi-heterogeneous interface effect, a large number of defects, and a higher proportion of 1T phase-MoS₂, and these also show excellent long-term stability after i-t curve measurement of 16 h. The authors should add some characterizations (such as XRD, Raman, XPS measurement) of the electrode materials after the HER test to further prove the stability of their multi-heterojunction interfaces catalysts, including the 1T phase percentage, chemical composition and their elemental valences of the electrode materials after the HER reaction. Whether these has changes?

8. In Introduction Part (Page 2, line 7-10), the authors only cited Ref.[10] and Ref.[11] to describe "Pt-based electrocatalysts are recognized as.....above-mentioned process". Here, I suggest that authors should add some references to further explain why Pt-based catalysts have fast reaction kinetics. In addition, in Introduction Part (Page 2, line 19-21), "the electrocatalytic activity of MoS₂ is closely associated with.....defect", the author should cite several references about MoS₂-based catalysts to elaborate on the influence of phase structure, interface and defects on the electrocatalytic performance of MoS₂.

Reviewer #3 (Remarks to the Author):

Comment on NCOMMS-21-09774

The authors work to engineer the interfacial electronic structure of molybdenum sulfide, which demonstrated enhanced HER performance in both alkaline and acid electrolytes. DFT calculation was also used to prove the electronic structure regulation induced by Ni-based materials. However, at this current state, the evidence for some of the statements is not so convincing. I am reluctant to offer my favorable recommendation for the publication of this work in Nature Communications, with the following reasons:

1. From the LSV curves in 1M KOH in Fig. 3a, the as-prepared electrocatalysts with the multi-heterojunction interfaces (1T0.81-MoS₂@Ni₂P and 110 1T0.72-MoS₂@NiS₂) show inferior HER performance to Pt/C. How could the authors declare that the as-prepared catalysts outperformed the commercial Pt/C in abstract? Besides, the authors declare "the incorporation of metallic-phase and intrinsic HER-active Ni-based materials ...reduced nickel possesses empty orbitals, which is helpful for additional H binding ability". However, I cannot find any characterizations and analysis to certify the "metallic-phase" and "reduced nickel" statement in the manuscript.
2. It's unnecessary to present two identical SEM images of T0.72-MoS₂@NiS₂ in Supplementary Fig. 2, maybe one should be deleted.
3. For better understanding the morphology of target catalysts, morphology characterizations of 1T0.41-MoS₂@Ni(OH)₂ are essential. Besides, the authors described that many random Ni₂P nanoparticles distributed on the MoS₂ microspheres from line 164-166 in manuscript, but it is difficult to distinguish Ni₂P nanoparticles from Supplementary Fig. 3.
4. From line 191-196 in manuscript, the statement, that NiS₂ nanoparticles are between the 1T0.41-MoS₂ layer on the basis of Raman results with slightly red-shift of E_{2g}1 and A_{1g} peaks, is not so convincing. Please cite some references to support this statement.
5. From TEM images of 1T0.72-MoS₂@ NiS₂ and 1T0.81-MoS₂@Ni₂P in Supplementary Fig. 5a, b and Supplementary Fig. 6a, b, the NiS₂ and Ni₂P seem to be in the form of nanosheets instead of nanoparticles, which needs to be further confirmed.
6. In the XPS analysis section, peak shift of Ni 2p_{1/2} and 2p_{3/2} in Ni 2p XPS spectrum for 1T0.72-MoS₂@NiS₂, 1T0.81-MoS₂@Ni₂P cannot be used as the evidence for interaction between NiS₂ (or

Ni₂P) and MoS₂ via as-formed hetero-structures. To demonstrate that, peak shift of Mo 3d may be more useful.

7. Usually, current density at overpotential of 0 V vs. RHE is very close to 0 mA cm⁻² for HER.

However, it deviates far from X axis for 1T0.72-MoS₂@NiS₂ and 1T0.81-MoS₂@Ni₂P in both alkaline and acid electrolytes in Fig. 3a, c. Please explain.

8. The reaction pathways on NiS₂ and Ni₂P need to be evaluated as a comparison.

9. English writing needs to be improved all over the manuscript!

Reviewer #1:

The first reviewer comment and response

List of changes (blue in the revised manuscript)

Comments: The authors reported non noble metal based electrocatalysts for hydrogen evolution reaction in acidic and alkaline medium. They investigated MoS₂/Ni₂P and MoS₂/Ni₂S materials. The main point in the manuscript is the discussion on phase transition (2H to 1T), interface between heterostructures (between MoS₂ and Ni₂S; between MoS₂ and Ni₂P) and defects. While examples of these strategies have been already reported (Adv. Mater. 2018, 30, 1705509; Nature Commun., 2020 11:5462), the combination of the proposed materials is new and timely, and can raise the strong attention of the broad readership of Nature Commun. The results are very interesting and the conclusions are solidly supported by the experimental findings. For all these reasons, I suggest to accept the manuscript for publication, pending minor revisions.

Our response: We thank the reviewer very much for positive recommendations and comments on the novelty and interesting of this work. Below we provide point-by-point answers to the comments made by the reviewer.

Comment 1: Some in-situ techniques that offer direct evidence on the nature of active sites are suggested. Where do the authors suggest the location of the active sites in this study? How do the electronic states of Ni or Mo at the interface behave during catalysis?

Response 1: Thanks for the comments and suggestions from the reviewer. (1) We agree with the reviewer that some in-situ techniques can provide direct evidence of electrocatalytically active sites. However, the current research on the active sites of the heterojunction interface has always been a huge challenge in the field of electrocatalysis. First of all, the real active sites and structural changes of the 1T_{0.81}-MoS₂@Ni₂P and 1T_{0.72}-MoS₂@NiS₂ electrodes during HER process was monitored by using in situ electrochemical-Raman spectroscopy (**Figure R1b**). The electrochemical-Raman spectra were conducted in 1.0 mol L⁻¹ KOH with the sample acting as working electrode, Ag/AgCl acting as reference electrode and Pt wire acting as count electrode, respectively (**Figure R1a**). As shown in **Figure R1c-d**, the Raman spectra of 1T_{0.72}-MoS₂@NiS₂ electrode collected under potentiostatic conditions at stepped potential values from 0 V to -1.5 V. The Raman peaks located at 147.3, 235.4 and 335.2 cm⁻¹ could be attributed to J₁, J₂ and J₃ vibrations of 1T-MoS₂ while the peaks at 382 and 407 cm⁻¹ are ascribed to the E_{2g}¹ and A_{1g} vibrational bands of 2H-MoS₂, and a vibrational peak (437.3 cm⁻¹) of Ni-S. However, when the 1T_{0.72}-MoS₂@NiS₂ sample is put in the electrolyte solution (1.0 M KOH solution), at different applied voltages from the -0.4 V to -1.5 V during electrocatalytic HER, these Raman peaks are significantly enhanced. In addition, many new peaks aroused at 152, 188, 222, 284, 322, 453, 500, 548 cm⁻¹ of MoS₂, respectively.¹ These peaks

changes suggested the formation of new chemical bonds among the $1T_{0.72}\text{-MoS}_2\text{@NiS}_2$, hydroxyl, protons and water molecules in 1.0 mol L^{-1} KOH solution, indicating the strong absorption ability for ions and molecules. As shown in **Figure R1d**, when applying bias potential of -0.4 V , the emerged two slight peaks of 429 and 488 cm^{-1} correspond to the $\nu_{\text{Ni-OH}}$ band of $1T_{0.72}\text{-MoS}_2\text{@NiS}_2$, suggesting the adsorbed water molecules dissociate into H_{ads} species and OH^- ions during cathodic polarization.² With the increased potentials in the range of $-0.4 \sim -1.5 \text{ V}$, the intensities of these bands of $1T_{0.72}\text{-MoS}_2\text{@NiS}_2$ are enhanced gradually, probably due to the fact that OH^- in alkaline media are driven to adsorb on the Mo, Ni, S atoms and the formation of the OOH^* intermediates³. As to $1T_{0.81}\text{-MoS}_2\text{@Ni}_2\text{P}$ sample, similar results were also obtained (Figure R1e, f). Designing the $1T_{0.72}\text{-MoS}_2\text{@NiS}_2$ (or $1T_{0.81}\text{-MoS}_2\text{@Ni}_2\text{P}$) multi-heterojunction interfaces through the in-situ interfacial engineering could generate the Ni–Mo metallic bond all over the surface of $1T\text{-}2H$ MoS_2 , leading to the doubled effective area of the active surface. The strong interaction between Ni and Mo increases the local electronic state of Mo atoms, decreasing the hydrogen adsorption energy for protons on Mo atoms and increasing the intrinsic activity of catalysts. We have added these data and evidences in the revised manuscript (**Supplementary Fig. 46**).

Figure R1. (a) In-situ Raman electrolysis cell. (b) In-situ electrochemical-Raman test system. (c, d) In-situ Raman spectra recorded of $1T_{0.72}\text{-MoS}_2\text{@NiS}_2$ and electrode, at different applied voltages from the 0 V to -1.5 V during electrocatalytic HER in 1.0 M KOH solution, scan rate of 50 mV s^{-1} . (e, f) In-situ Raman spectra recorded of $1T_{0.81}\text{-MoS}_2\text{@Ni}_2\text{P}$ electrode, at different applied voltages from the 0 V to -1.5 V during electrocatalytic HER in 1.0 M KOH solution, scan rate of 50 mV s^{-1} .

References:

- [1] Li, Z. et al. Effects of structural changes on the enhanced hydrogen evolution reaction for Pd NPs@2H-MoS₂ studied by In-Situ Raman spectroscopy[J]. Chem. Phys. Lett. **764**, 138267 (2020).
- [2] Zuleta, A. et al. Improvement of the erosion-corrosion resistance of magnesium by electroless Ni-P/Ni(OH)₂-ceramic nanoparticle composite coatings[J]. Surf. Coat. Technol. **304**, 167-178 (2016).
- [3] Zhang, S. et al. In situ interfacial engineering of nickel tungsten carbide janus

structures for highly efficient overall water splitting[J]. *Sci. Bull.* **65**, (8):640-650 (2020).

(2) In order to understand how do the electronic states of Ni or Mo at the interface behave during catalysis. We performed X-ray absorption near-edge structure (XANES) spectra measurements on the fresh catalysts and those after being used in the HER process at three representative potentials (-0.04, -0.1 and -0.2 V). Unfortunately, our beamline (the BL8 beamline of Thailand Synchrotron Light Research Institute (SLRI), Thailand) does not have enough photo flux to collect EXAFS data. We have added these data and evidences in the revised manuscript (**Fig. 5** and **Supplementary Fig. 47-49**, and **Supplementary Table 9**)

X-ray absorption spectroscopy. To investigate electronic states of catalysts, X-ray absorption near-edge structure (XANES) spectra were measured on the fresh catalysts and those after being used in the HER process at three representative potentials (-0.04, -0.1 and -0.2 V), near the onset potential and the overpotential at current densities of 5 and 10 mA cm⁻² (for 1T_{0.81}-MoS₂@Ni₂P sample), respectively. **Figure R2a** presents Ni K-edge XANES spectra of 1T_{0.81}-MoS₂@Ni₂P catalyst recorded at different applied potentials and reference spectra of Ni foil, NiO and NiO₂. From the fresh catalyst to that under the -0.04 V potential condition, the absorption edge is shifted to the lower energy side by ~0.5 eV, along with a broadening of the white line peak, meaning a decrease of the Ni oxidation state. Moreover, when cathodic potentials of -0.04 and -0.1 V versus RHE were applied, a further shift of the absorption edge towards lower energy by ~0.2 eV occurs in relation to the case under the -0.04 V potential condition, implying a distinct decrease in the Ni valence state in 1T_{0.81}-MoS₂@Ni₂P during the HER. Notably, all catalyst spectra exhibit white line at 8350.2~8350.9 eV (**Supplementary Fig. 47**), corresponding to the 1s to 4p electronic transition, indicating Ni-O local coordination similar to NiOOH and Ni oxides.⁶⁵ Edge position of the fresh catalyst is lower than that of NiO₂, indicating less ionic bonding character of Ni and P. Using the edge positions of NiO and NiO₂ as references, the Ni average valence state was determined as +3.3, +2.2, +1.8 and +2.0 for the fresh and those used at -0.04, -0.1 and -0.2V, respectively (**Table R1**). Therefore, Ni cations are reduced under working conditions, which is consistent with the Ni 2p XPS result (**Supplementary Fig. 28d**). In addition, 1T_{0.72}-MoS₂@NiS₂ (**Supplementary Fig. 48**) and 1T_{0.41}-MoS₂@Ni(OH)₂ (**Supplementary Fig. 49**) show similar behavior as Ni valency is decreased from +3.6 (fresh) to +2.4 (-0.2 V) and from +2.7 (fresh) to 1.8 (-0.2 V), respectively. In addition, oxidation of Mo from +4 to +6 after catalysis was observed as shown in **Figure R2b**. The fresh 1T_{0.81}-MoS₂@Ni₂P exhibit much broader Mo L₃ XANES absorption than those of the Mo standards (MoS₂, MoO₂ and MoO₃), and its lower edge position indicates reduced Mo admixture. In most cases, broadening of XANES relates to lack of crystallinity. Interestingly, after the HER reaction, the broad peak was shifted to higher energy and split, indicating *4d_{t_{2g}}* and *4d_{e_g}* absorption bands of MoO₃⁶⁶ at 2524 eV and 2526 eV, respectively. These results indicate the valence state of Mo cations is increased from approximately +4 to a

higher oxidation state (+6) under working conditions. Moreover, sulfur does not take part in catalysis as shown in **Figure R2c**. All S K-edge XANES spectra of catalysts before and after the reaction are similar and agree well with that of standard MoS₂. P K-edge XANES spectra of 1T_{0.81}-MoS₂@Ni₂P and FePO₄ are shown in **Figure R2d**. The white line at 2154 eV belongs to PO₄²⁻ associated with hybridized O 2p-P 3p absorption bands⁶⁷ whereas the original Ni₂P species appear only as a minor peak located at 2146 eV. The peak position and absorption line shape of Ni₂P are close to that of Co₂P indicates the valence state P³⁻.⁶⁸ The ratio of Ni₂P to PO₄²⁻ changes with the cathodic potential, and the highest ratio is 1:3 at -0.2 V. In the fresh catalyst, the front peak is much weaker, which indicates that although there is a Ni-P bond, its signal is changed by the presence of some other elements. When cathodic potentials of -0.1 and -0.2 V versus RHE were applied, showing the highest intensity of the front peak, which indicates that both P and Ni are involved in the reaction during the HER process.

Figure R2. XANES spectra of fresh 1T_{0.81}-MoS₂@Ni₂P catalyst and after being used in the HER process at -0.04, -0.1 and -0.2 V vs RHE, respectively. **a** Ni K-edge XANES spectra of 1T_{0.81}-MoS₂@Ni₂P catalyst and standards Ni foil, NiO, and NiO₂. Inset, Magnified pre-edge XANES region. **b** Mo L-edge XANES spectra of 1T_{0.81}-MoS₂@Ni₂P catalyst and standards MoS₂, MoO₂, and MoO₃. **c** S K-edge XANES spectra of 1T_{0.81}-MoS₂@Ni₂P catalyst and standards MoS₂, and FeSO₄. **d** P

K-edge XANES spectra of 1T_{0.81}-MoS₂@Ni₂P catalyst and standards CH₂-P(OH)₃, and FeSO₄.

Table R1. The fitted average oxidation states of Ni from XANES spectra on the fresh catalyst and after being used in the HER process at -0.04, -0.1 and -0.2 V vs RHE, respectively. ΔE : Ni K-edge position (eV), relative to Ni foil, Error = ± 0.2 eV; OS: Oxidation state (Linear fit: OS = 2.2222* ΔE -26.2222).

1T _{0.81} MoS ₂ @Ni ₂ P	White line peak	E ₀	ΔE	OS = 2.2222* ΔE -26.2222
Fresh	8350.5	8346.3	13.3	3.3
-0.04V	8350.9	8345.8	12.8	2.2
-0.1V	8350.2	8345.6	12.6	1.8
-0.2V	8350.5	8345.7	12.7	2.0
NiO ₂	8351.1	8346.6	13.6	4
NiO	8350.8	8345.7	12.7	2.0
Ni foil		8333	0	0

References:

- [65] Chung, Y-H. et al. Anomalous in situ activation of carbon-supported Ni₂P nanoparticles for oxygen evolving electrocatalysis in alkaline media, *Scientific Reports* **7**, 8236 (2017).
- [66] Tsai, H-M. et al. Anisotropic electronic structure in quasi-one-dimensional K_{0.3}MoO₃: An angle-dependent x-ray absorption study, *Appl. Phys. Lett.* **91**, 022109 (2007).
- [67] Prietzel, J. et al. Reference spectra of important adsorbed organic and inorganic phosphate binding forms for soil P speciation using synchrotron-based K-edge XANES spectroscopy, *J. Synchrotron Rad.* **23**, 532-544 (2016).
- [68] Zhang, Y. et al. Structural designs and in-situ X-ray characterizations of metal phosphides for electrocatalysis, *Chem. Cat. Chem.* **12**, 3621-3628 (2020).

Comment 2: What is the catalytic efficiency of the materials in terms of Faradaic efficiency and turnover frequency?

Response 2: We thank the reviewer for his (her) suggestions. 1) Calculation for Faradaic efficiency: a glass-sealed gas circulation system (Labsolar 6A, Perfect Light, China) equipped with a three electrode H-type electrochemical cell was used to collect the gas product. The product was subsequently detected by a thermal conductivity detector (TCD) in Agilent 7890B gas chromatography (GC). The as-prepared 1T_{0.72}-MoS₂@NiS₂ and 1T_{0.81}-MoS₂@Ni₂P electrodes were directly used as the working electrode (1.0 cm²). The electrochemical experiments were performed at a constant applied current density of 10 mA cm⁻². The electrolytes were 1.0 M KOH and 0.5 M H₂SO₄, respectively. GC data was collected for 120 min at the same intervals. The faradaic efficiency was calculated by the following equation¹: FE% = 2nF/Q.

Where F and n are the faraday constant and the amount of produced hydrogen,

respectively; Q is the total amount of charge flowed past the electrochemical cell.

In this case, the Faradaic efficiency of HER in 0.5 M H_2SO_4 solution has been presented, confirming the real electrocatalytic activities of $1\text{T}_{0.81}\text{-MoS}_2\text{@Ni}_2\text{P}$ catalyst. As for the theoretical value, we assumed that 100% current efficiency during the reaction, which means only the HER process was occurring at the working electrode. The expected-theoretically amount of hydrogen was then calculated by applying the Faraday law, which states that the passage of 96485.4 C, causes 1 equivalent of reaction. Afterwards, the amount of hydrogen evolution of $1\text{T}_{0.81}\text{-MoS}_2\text{@Ni}_2\text{P}$ catalyst was measured in comparison of theoretical quantity in **Figure R3**, presenting a promising Faraday efficiency of $98.7 \pm 0.5\%$ towards real water splitting into hydrogen. Typically, the amount of hydrogen evolution of $1\text{T}_{0.72}\text{-MoS}_2\text{@NiS}_2$ catalyst in 1.0 M KOH solution was also measured in comparison of theoretical quantity in **Figure R4**, presenting HER Faraday efficiency of $97.6 \pm 0.6\%$ owe to the synergistic effect of the phase, defect and heterostructure engineering. In the revised manuscript, we added these evidences as Supplementary Fig. 18 and Supplementary Fig. 22 in the Supplementary Information.

Figure R3. The yields of hydrogen theoretically calculated from HER current and tested from an online gas chromatography system by use of as-prepared $1\text{T}_{0.81}\text{-MoS}_2\text{@Ni}_2\text{P}$ electrode at a current density of 10 mA cm^{-2} in 0.5 M H_2SO_4 electrolyte.

Figure R4. The yields of hydrogen theoretically calculated from HER current and tested from an online gas chromatography system by use of as-prepared $1T_{0.72}\text{-MoS}_2\text{@NiS}_2$ electrode at a current density of 10 mA cm^{-2} in 1.0 M KOH electrolyte.

2) For rough estimation of per-site turnover frequency (TOF), we previously carried out according to the previously approach adopted by Jaramillo et al.² In this way, the geometric areas of the assembled electrodes were hypothesized. Then TOF values were also calculated. In order to ascertain the reasonable TOF values, cycle voltammetry (CV) method can be applied in this part. The TOF values (S^{-1}) were calculated with the following formula: $\text{TOF} = I/2NF$

I: current density extracted from the LSV curves;

F: Faraday constant;

N: the number of active sites.

Cycle voltammetry (CV) measurements were conducted between -0.2 V and 0.6 V vs. RHE in 1.0 M PBS at a scan rate of 50 mv.S^{-1} (**Figure R5**). The absolute components of the voltammetric charges tested during one CV cycle were calculated. Assuming a one-electron process for both reduction and oxidation, the absolute charges was divided by two and the Faraday constant to obtain the number of active sites of the catalysts. The upper limit of active sites (N) for $1T_{0.72}\text{-MoS}_2\text{@NiS}_2$ and $1T_{0.81}\text{-MoS}_2\text{@Ni}_2\text{P}$ catalysts could be calculated according to the equation: $N = Q/2F$, where F and Q are the Faraday constant and the whole charge of CV curve, respectively.

Figure R5. Cycle voltammetry curves for 2H_{phase}-MoS₂, 1T_{0.41}-MoS₂, 1T_{phase}-MoS₂, 1T_{0.41}-MoS₂@Ni(OH)₂, 1T_{0.72}-MoS₂@NiS₂ and 1T_{0.81}-MoS₂@Ni₂P in 1.0 M PBS at a scan rate of 50 mv.S⁻¹.

Table R2. The turnover frequency (TOF) values for various HER electrocatalysts at different overpotentials in alkaline condition.

Catalysts	TOF values in HER		
	$\eta = 100$ mV	$\eta = 200$ mV	$\eta = 300$ mV
1T _{0.81} -MoS ₂ @Ni ₂ P	0.97 S ⁻¹	3.56 S ⁻¹	12.78 S ⁻¹
1T _{0.72} -MoS ₂ @NiS ₂	0.31 S ⁻¹	2.26 S ⁻¹	10.93 S ⁻¹
1T _{0.41} -MoS ₂ @Ni(OH) ₂	0.26 S ⁻¹	0.89 S ⁻¹	2.36 S ⁻¹
1T _{phase} -MoS ₂	0.12 S ⁻¹	0.56 S ⁻¹	1.17 S ⁻¹
1T _{0.41} -MoS ₂	0.09 S ⁻¹	0.27 S ⁻¹	0.88 S ⁻¹
2H-MoS ₂	0.06 S ⁻¹	0.19 S ⁻¹	0.56 S ⁻¹

To explore the intrinsic electrocatalytic performance of each active sites, the turnover frequency (TOF) is calculated. Based on above-mentioned analysis, cycle voltammetry approach is regarded as the promising way to determine the reasonable results. The TOF values of 1T_{0.81}-MoS₂@Ni₂P (3.56 S⁻¹) and 1T_{0.72}-MoS₂@NiS₂ (2.26 S⁻¹) heterojunction catalyst at the overpotential of 200 mV are 18.7 and 11.9 times higher than of 2H_{phase}-MoS₂ catalyst (0.19 S⁻¹) for HER, respectively, as shown in **Table R2**. In the revised manuscript, we added these evidences as Supplementary Fig. 17, and **Supplementary Table 1** in the Supplementary Information.

References:

- [1] Zhai, P. et al. Engineering active sites on hierarchical transition bimetal oxides/sulfides heterostructure array enabling robust overall water splitting[J]. Nat. Commun. **11**, 5462 (2020).
- [2] Benck, J. D. et al. Amorphous molybdenum sulfide catalysts for electrochemical hydrogen production: insights into the origin of their catalytic activity. ACS Catal. **2**,

1916-1923 (2012).

Comment 3: The authors reported the current density normalized with respect to geometric areas of the working electrode. It is important to investigate the intrinsic activity by normalizing with C_{dl} , TOF... A comparison with other literature would be beneficial.

Response 3: We thank to the reviewer's helpful comment. According to your suggestion, we have added relevant references to compare the C_{dl} and TOF of our catalysts with other literature, as shown in **Table R3**. We have added these data in the revised Supplementary Information (**Supplementary Table 2**).

Table R3. Comparison of TOF values of HER catalysts in alkaline condition.

Catalysts	TOF (H_2 S^{-1} @mV)	C_{dl} (mF cm^{-2})	Reference
NiCo ₂ P _x	0.056 S^{-1} @100 mV	24.2 mF cm^{-2}	[1]
MoNi ₄ /MoO _{3-x}	1.13 S^{-1} @100 mV	128 mF cm^{-2}	[2]
N-NiCo ₂ S ₄	1.0 S^{-1} @125 mV	18 mF cm^{-2}	[3]
NiO@1T-MoS ₂	0.7 S^{-1} @130 mV	18.32 mF cm^{-2}	[4]
Mo ₁ N ₁ C ₂	1.46 S^{-1} @150 mV	21.56 mF cm^{-2}	[5]
Co-NiS ₂	4.1 S^{-1} @200 mV	18.9 mF cm^{-2}	[6]
MoO ₃ @MoS ₂	1.93 S^{-1} @250 mV	337.3 mF cm^{-2}	[7]
NiMoO _x @NiMoS	0.28 S^{-1} @50 mV	23.4 mF cm^{-2}	[8]
Ni-Co@1T-MoS ₂	0.98 S^{-1} @100 mV	108.1 mF cm^{-2}	[9]
1T-2H MoS ₂	13.14 S^{-1} @250 mV	2.41 mF cm^{-2}	[10]
(N, PO ₄ ³⁻)-MoS ₂ /VG	0.03 S^{-1} @100 mV	140 mF cm^{-2}	[11]
1T-MoS ₂ QS/Ni(OH) ₂	0.37 S^{-1} @100 mV	11.5 mF cm^{-2}	[12]
Ni ₂ P/MoS ₂ /N:RGO	12.27 S^{-1} @100 mV	61.7 mF cm^{-2}	[13]
2D-MoS ₂ /Co(OH) ₂	4.31 S^{-1} @200 mV	0.91 mF cm^{-2}	[14]
1T_{0.72}-MoS₂@NiS₂	12.78 S^{-1}@300 mV	316.2 mF cm^{-2}	This work
1T_{0.81}-MoS₂@Ni₂P	10.93 S^{-1}@300 mV	359.7 mF cm^{-2}	This work

References:

- [1] Zhang, R. et al. Ternary NiCo₂P_x nanowires as pH-universal electrocatalysts for highly efficient hydrogen evolution reaction[J]. Adv. Mater. **29**, 1605502 (2017).
- [2] Chen, Y. et al. Self-templated fabrication of MoNi₄/MoO_{3-x} nanorod arrays with dual active components for highly efficient hydrogen evolution[J]. Adv. Mater. **29**, 1703311 (2017).
- [3] Wu, Y. et al. Electron density modulation of NiCo₂S₄ nanowires by nitrogen incorporation for highly efficient hydrogen evolution catalysis[J]. Nat. Commun. **9**, 1425 (2018).
- [4] Huang, Y. et al. Atomically engineering activation sites onto metallic 1T-MoS₂ catalysts for enhanced electrochemical hydrogen evolution[J]. Nat. Commun. **10**, 982 (2019).
- [5] Chen, W X. et al. Rational design of single molybdenum atoms anchored on

N-doped carbon for effective hydrogen evolution reaction[J]. *Angew. Chem. Int. Ed.* **56**, 16086-16090 (2017).

[6] Yin, J. et al. Atomic arrangement in metal-doped NiS₂ boosts the hydrogen evolution reaction in alkaline media[J]. *Angew. Chem. Int. Ed.* **58**, 18676-18682 (2019).

[7] Huang, L. B. et al. Self-limited on-site conversion of MoO₃ nanodots into vertically aligned ultrasmall monolayer MoS₂ for efficient hydrogen evolution[J]. *Adv. Energy Mater.* **8**, 1800734 (2018).

[8] Zhai, P. et al. Engineering active sites on hierarchical transition bimetal oxides/sulfides heterostructure array enabling robust overall water splitting[J]. *Nat. Commun.* **11**, 5462 (2020).

[9] Li, H. et al. Amorphous nickel-cobalt complexes hybridized with 1T-phase molybdenum disulfide via hydrazine-induced phase transformation for water splitting. *Nat. Commun.* **8**, 15377 (2017).

[10] Wang, S. et al. Ultrastable in-plane 1T-2H MoS₂ heterostructures for enhanced hydrogen evolution reaction[J]. *Adv. Energy Mater.* **8**, 1801345 (2018).

[11] Deng, S. et al. Synergistic doping and intercalation: realizing deep phase modulation on MoS₂ arrays for high-efficiency hydrogen evolution reaction[J]. *Angew. Chem. Int. Ed.* **131**, 16435-16442 (2019).

[12] Chen, W. et al. Achieving rich and active alkaline hydrogen evolution heterostructures via interface engineering on 2D 1T-MoS₂ quantum sheets[J]. *Adv. Funct. Mater.* **30**, 2000551 (2020).

[13] Kim, M. et al. Activating MoS₂ basal plane with Ni₂P nanoparticles for Pt-Like hydrogen evolution reaction in acidic media[J]. *Adv. Funct. Mater.* **29**, 1809151 (2019).

[14] Zhu, Z. et al. Ultrathin transition metal dichalcogenide/3D metal hydroxide hybridized nanosheets to enhance hydrogen evolution activity[J]. *Adv. Mater.* **30**, 1801171 (2018).

Comment 4: On page-10, line 212, the authors claim the presence of S vacancy (in fact they mentioned S defect) from HRTEM observation. It is difficult to unequivocally associate this defect to S vacancy unless supported by other evidence.

Response 4: We thank for the reviewer's helpful comment. Indeed, it maybe that we are not rigorous enough. We have added some new HRTEM data (**Figure R6**), where we can only observe S defects or disorder. We will carefully revise it in our revised manuscript.

Figure R6. Typical HRTEM image of $1T_{0.81}\text{-MoS}_2@\text{Ni}_2\text{P}$.

Comment 5: The discussion on XPS result should be polished. For example,

- On page-12, line 250, the authors state: “Both peaks were slightly shifted (by 0.5 eV) towards higher binding energies”. What is the reference for this shifting? Did the authors compare with Ni 2p spectrum of pure Ni_2S or Ni_2P ?
- I am a bit surprised to see the Ni-P type bonding correlated to the peak at around 135 in XPS spectra of P 2P (Fig.S7). This peak is meant for surface oxidized phosphate. The metal-phosphide bond appears at BE around 129 or 130. May the authors comment on that?

Response 5: Thanks for the comments and suggestions from the reviewer. (1) According to your suggestion, we have added the Ni 2p spectrum of pure Ni_2P and NiS_2 as references in our revised manuscript. As shown in **Figure R7a**, the Ni 2p spectrum of $1T_{0.81}\text{-MoS}_2@\text{Ni}_2\text{P}$ shows two spin-orbit doublets at binding energies of 856.6 and 874.9 eV, which are assigned to $\text{Ni}^{2+} 2p_{3/2}$ and $\text{Ni}^{2+} 2p_{1/2}$ oxidation states in Ni_2P , respectively, along with two satellite peaks (identified as “Satellite.”).¹ Notably, compared to the values (857.4 eV for $\text{Ni} 2p_{3/2}$ and 875.4 eV for $\text{Ni} 2p_{1/2}$) of pure Ni_2P , the two binding energies have a noticeable negative-shift of approximately 0.8 and 0.6 eV in $\text{Ni} 2p_{3/2}$ and $\text{Ni} 2p_{1/2}$ for $1T_{0.81}\text{-MoS}_2@\text{Ni}_2\text{P}$ sample (**Figure R2a**), respectively. The negative-shift in the Ni 2p binding energies peaks manifest the lower valence state and electron-rich structure of Ni^{2+} sites caused by the electron transfer from Mo^{4+} to Ni^{2+} sites in the $1T_{0.81}\text{-MoS}_2@\text{Ni}_2\text{P}$ sample². For pure NiS_2 sample, the peaks of $\text{Ni} 2p_{3/2}$ and $\text{Ni} 2p_{1/2}$ at 854.7 and 872.2 eV, and the corresponding satellite appear at 858.7 and 878.6 eV, respectively. However, the binding energies of $\text{Ni} 2p_{3/2}$, $\text{Ni} 2p_{1/2}$ and satellite in $1T_{0.72}\text{-MoS}_2@\text{NiS}_2$ sample (**Figure R2b**) are positive-shifted to 856.6 (by 1.9 eV), 875.3 (by 3.1 eV), 861.8 (by 3.1 eV) and 862.2 eV (by 1.9 eV),

respectively. The positive-shift in the Ni 2p binding energies peaks manifest the higher valence state which are attributed to Ni bonded to S and O atoms in the form of sulfides or surface oxides/hydroxides³.

Figure R7. (a) XPS spectra for $1T_{0.81}\text{-MoS}_2\text{@Ni}_2\text{P}$ catalyst and pure Ni_2P . (b) XPS spectra for $1T_{0.72}\text{-MoS}_2\text{@NiS}_2$ catalyst and pure NiS_2 .

References:

- [1] Zeng, L. et al. Three-dimensional-networked $\text{Ni}_2\text{P}/\text{Ni}_3\text{S}_2$ hetero-nanoflake arrays for highly enhanced electrochemical overall-water-splitting activity[J]. *Nano Energy*, **51**, 26-36 (2018).
- [2] Zeng, L. et al. Multiple modulation of hierarchical NiS_2 nanosheets by Mn heteroatom doping engineering for boosting alkaline and neutral hydrogen evolution[J]. *J. Mater. Chem. A*, **7**, 25628 (2019).
- [3] Feng, J. X. et al. Efficient hydrogen evolution on Cu nanodots-decorated Ni_3S_2 nanotubes by optimizing atomic hydrogen adsorption and desorption[J]. *J. Am. Chem. Soc.* **140**, 610-617 (2017).

(2) We are very grateful for the reviewers' suggestions. We agree with the reviewer that 135 eV peak is meant for surface oxidized phosphate. As shown in **Figure R8b**, two peaks of 130.4 and 129.5 eV appear in the P 2p region, which can be attributed to the binding energy of P 2p_{1/2} and P 2p_{3/2} in Ni_2P .^{1,2} And another peak located at 134.7 eV can also be observed, which indicates the existence of oxidized phosphate (P-O) species (**Figure R8a**), which is attributed to the contact between air and the surface of Ni_2P .³ In our revised manuscript, we have carefully revised it.

Figure R8. (a, b) XPS spectra of P 2p spectrum of 1T_{0.81}-MoS₂@Ni₂P catalyst.

References:

- [1] Lu, Y. et al. Ni₂P/Graphene Sheets as Anode Materials with Enhanced Electrochemical Properties versus Lithium[J]. *J. Phys. Chem. C*, **116**, 22217–22225 (2012).
- [2] Cai, Z. et al. Electrodeposition-Assisted Synthesis of Ni₂P Nanosheets on 3D Graphene/Ni Foam Electrode and Its Performance for Electrocatalytic Hydrogen Production. *Chem. Electro. Chem.* **2**, 1665–1671 (2015).
- [3] Pan, Y. et al. Monodispersed nickel phosphide nanocrystals with different phases: synthesis, characterization and electrocatalytic properties for hydrogen evolution[J]. *J. Mater. Chem. A*, **3**, 1656-1665 (2015).

Comment 6: I would suggest to include in Figure S4a the standard patterns of MoS₂, Ni₂P, Ni₂S, to offer ease of comparison on the presence of suggested crystalline phases. It is also important to include/discuss the XRD pattern and Raman spectra of pure 1T MoS₂.

Response 6: Thanks for the comments and suggestions from the reviewer. According to your suggestion, we have added the standard patterns of MoS₂, Ni₂P and NiS₂, as shown in **Figure R9**. This figure was replaced as **Supplementary Fig. 4** in the revised Supplementary Information.

Figure R9. XRD patterns of as-synthesized materials on carbon cloth. a. XRD patterns of $1T_{0.41}\text{-MoS}_2$ and $1T_{0.41}\text{-MoS}_2\text{@Ni(OH)}_2$. b. XRD patterns of $1T_{0.41}\text{-MoS}_2$ and $1T_{0.72}\text{-MoS}_2\text{@NiS}_2$ catalysts. c. XRD patterns of $1T_{0.41}\text{-MoS}_2$ and $1T_{0.81}\text{-MoS}_2\text{@Ni}_2\text{P}$ catalysts. d. Raman spectra of $2H_{\text{phase}}\text{-MoS}_2$, $1T_{0.41}\text{-MoS}_2\text{@Ni(OH)}_2$, $1T_{0.72}\text{-MoS}_2\text{@NiS}_2$, $1T_{0.81}\text{-MoS}_2\text{@Ni}_2\text{P}$ catalysts.

Comment 7: It is well documented that the metallic 1T phase delivers fast electron transfer during HER, the authors also mentioned this. However, we see that the 2H phase is performing better than the 1T phase in Figure 3a and c. A comment from the authors on this would be beneficial.

Response 7: We appreciate the reviewer's kind comment. Indeed, for MoS_2 , the metallic 1T phase has better HER catalytic activity than the 2H phase, due to the metallic 1T phase delivers fast electron transfer during the HER process. However, in our manuscript, we used Li-intercalated bulk MoS_2 to obtain $1T_{\text{phase}}\text{-MoS}_2$ sample, and then loaded it on a carbon cloth (CC). It is because it is difficult to obtain pure 1T- MoS_2 with our organic acid induced hydrothermal method. Therefore, we believe that the reason that the HER performance of 1T phase is worse than that of 2H is that the poor contact between MoS_2 and CC leads to poor conductivity. In order to solve this problem, we annealed (500 °C) the CC loaded with $1T_{\text{phase}}\text{-MoS}_2$ sample under the protection of Ar gas. As shown in **Figure R10**, there are LSV curves for $1T_{\text{phase}}\text{-MoS}_2$ (before annealing), $2H_{\text{phase}}\text{-MoS}_2$, $1T_{\text{phase}}\text{-MoS}_2$ (after annealing) in 1.0 M KOH solution. We found that the HER performance of $1T_{\text{phase}}\text{-MoS}_2$ after annealing treatment is significantly better than that of $2H_{\text{phase}}\text{-MoS}_2$. In our revised manuscript, we used the HER performance data of $1T_{\text{phase}}\text{-MoS}_2$ after annealing treatment, and a detailed description has been added in the "Methods" section.

Figure R10. LSV curves for $1T_{\text{phase}}\text{-MoS}_2$ (before annealing), $2H_{\text{phase}}\text{-MoS}_2$, $1T_{\text{phase}}\text{-MoS}_2$ (after annealing) in 1.0 M KOH solution.

Comment 8: Detailed procedure of the electrochemical measurements should be

given: was there surface activation? At what voltage did the authors measure the EIS?

Response 8: Thanks for the suggestions from the reviewer. Before the polarization curve test, we first used cyclic voltammetry to activate our catalyst, and the general voltage range is -0.6 V~ -1.2 V. Until the CV curve is completely closed and coincides with the previous path, indicating that the surface of our sample has stabilized, other electrochemical measurements can be carried out. The EIS were measured by AC impedance spectroscopy at the frequency ranges 10^6 to 1.0 Hz at 300 mV. The detailed procedure of the electrochemical measurements is as follows:

“All electrochemical measurements were performed with a CHI 660E Electrochemical Workstation (CHI Instruments, Shanghai Chenhua Instrument Corp., China). The electrocatalytic HER performance of different electrocatalysts (1.0 cm^2) were evaluated using a typical three-electrode system in N_2 -saturated 1.0 M KOH and 0.5 M H_2SO_4 electrolyte, respectively. All polarization curves at 5 mV/s were corrected without iR compensation. Before testing the polarization curve, we first performed cyclic voltammetry (CV) for more than 20 cycles to activate the as-prepared catalysts with a scan rate of 50 mV/s. The EIS tests were measured by AC impedance spectroscopy at the frequency ranges 10^6 to 1.0 Hz at 300 mV. According to the Nernst equation ($E_{\text{RHE}} = E_{\text{Hg/HgO}} + 0.059 \text{ pH} + 0.098$), where E_{RHE} was the potential vs. a reversible hydrogen potential, $E_{\text{Hg/HgO}}$ was the potential vs. Hg/HgO electrode, and pH was the pH value of electrolyte. The electrochemical stability was evaluated by chronoamperometry measurements at a static overpotential, during which the current variation with time was recorded. The ECSA values were measured through CV in the selected non-faradaic range. The current densities have linear relationship against different scan rates ($10 \sim 60 \text{ mV s}^{-1}$) and the values of the slope were considered as twice of C_{dl} . To determination of Faradaic efficiency, the Faradaic efficiency of HER catalyst is defined as the ratio of the amount of experimentally determined hydrogen to that of the theoretically expected hydrogen from the HER in 1.0 M KOH (or 0.5 M H_2SO_4) aqueous solution by use of an online gas chromatography system (GC, Techcomp GC 7890 T, Ar carrier gas, Thermo Conductivity Detector). The OER tests were performed in O_2 -saturated 1.0 M KOH solution, the others are the same as HER test conditions. The overall water splitting performance was evaluated in 1.0 M KOH using a two-electrode configuration, and the polarization curve was recorded at a scan rate of 5 mV s^{-1} . For the comparison experiment, Pt/C and IrO_2 ink were prepared by dissolving 8 mg Pt/C and 8 mg IrO_2 powder in the mixture of 700 μL ethanol, 300 μL deionized water and 50 μL Nafion with 30 min of ultrasonication, respectively. Then the as-prepared ink was coated onto the carbon cloth (CC) with the loading mass density of about 3.0 mg/cm^2 and was dried at $60 \text{ }^\circ\text{C}$. The long-term stability measurements were performed using the chronoamperometry measurements. All polarization curves at 5 mV/s were corrected without iR compensation.”

Comment 9: This study seems based purely on electrocatalysis not on

photo-electrocatalysis. So, what is the relevance of light on/off during the EIS study?

Response 9: We thank for the reviewer's kind comment. Indeed, we are reporting pure electrocatalysis rather than photo-electrocatalysis. We are very grateful to the reviewers for pointing out our mistakes, which may be our previous clerical errors. We have deleted "with light on or off" in our revised manuscript.

Comment 10: How about the OER and overall water splitting activities of $1T_{0.72}\text{-MoS}_2\text{@NiS}_2$ and $1T_{0.81}\text{-MoS}_2\text{@Ni}_2\text{P}$?

Response 10: We are grateful for the reviewers' suggestions. According to your suggestion, we have added OER and overall water splitting activities of $1T_{0.72}\text{-MoS}_2\text{@NiS}_2$ and $1T_{0.81}\text{-MoS}_2\text{@Ni}_2\text{P}$ in our revised manuscript.

(1) **Electrocatalytic OER performance.** In general, the efficiency is always limited by OER as major barrier for overall water splitting. The OER performance of our catalysts was investigated in 1.0 M KOH solution in a three-electrode system. The polarization curves of different samples are shown in **Figure R11a**, $1T_{0.72}\text{-MoS}_2\text{@NiS}_2$ sample exhibits the best OER performance among all samples. Interestingly, OER activities of $1T_{0.72}\text{-MoS}_2\text{@NiS}_2$ and $1T_{0.81}\text{-MoS}_2\text{@Ni}_2\text{P}$ are better than commercial IrO_2 catalyst. Especially, the as-prepared $1T_{0.72}\text{-MoS}_2\text{@NiS}_2$ (or $1T_{0.81}\text{-MoS}_2\text{@Ni}_2\text{P}$) sample presents the low overpotentials of 320 (370), 420 (400), and 510 (610) mV at current densities of 50, 100, and 150 mA cm^{-2} towards OER, which are better than others and commercial IrO_2 catalyst (**Figure R11c**), compared with $1T_{\text{phase}}\text{-MoS}_2$ (420, 550, 680 mV), $1T_{0.41}\text{-MoS}_2$ (570, 750, 910 mV), $2H_{\text{phase}}\text{-MoS}_2$ (790 mV @50 mA cm^{-2}) and IrO_2 (380, 510, 790 mV). To in-depth understand the OER kinetic mechanism, we calculated the Tafel slopes of these electrodes using the Tafel equation^{1,2} and obtained the smallest slopes equal to 56 and 57 mV/dec for the electrodes containing $1T_{0.72}\text{-MoS}_2\text{@NiS}_2$ and $1T_{0.81}\text{-MoS}_2\text{@Ni}_2\text{P}$, respectively (**Figure R11b**). These values are even closer to the corresponding slope of the IrO_2 electrode (54 mV/dec), demonstrating the fast OER kinetics of $1T_{0.72}\text{-MoS}_2\text{@NiS}_2$ and $1T_{0.81}\text{-MoS}_2\text{@Ni}_2\text{P}$ electrodes. It can be also seen that the present $1T_{0.72}\text{-MoS}_2\text{@NiS}_2$ possesses excellent catalytic activities towards OER with smaller overpotential and Tafel slope than most of the previous reports (**Figure R11d** and **Table R4**). In addition, the $1T_{0.72}\text{-MoS}_2\text{@NiS}_2$ and $1T_{0.81}\text{-MoS}_2\text{@Ni}_2\text{P}$ have excellent stabilities. As shown in **Figure R12 (a, c)**, there is only a slight decrease of the current density after testing for 16 h at 20 mA cm^{-2} , and the LSV curves measured before and after the long-term tests are also a slight drop (**Figure R12 (b, d)**). This may be due to the slight shedding of phosphide or sulfide in our catalyst during the electrochemical reaction of the alkaline medium.

Figure R11. OER performance of different samples tested in 1.0 M KOH. **a** OER polarization curves, **b** Tafel slopes, **c** overpotentials at typical current densities of $2H_{\text{phase}}\text{-MoS}_2$, $1T_{0.41}\text{-MoS}_2$, $1T_{\text{phase}}\text{-MoS}_2$, $1T_{0.72}\text{-MoS}_2@NiS_2$, $1T_{0.81}\text{-MoS}_2@Ni_2P$ and IrO_2 . **d** η_{100} and Tafel slopes for various transition metal-based HER electrocatalysts in 1.0 M KOH. (All LSV curves were corrected without iR-compensation).

Table R4. Comparison of OER activities for various electrocatalysts.

Catalysts	Overpotential @100 mA/cm ² (mV)	Tafel slope (mV/dec)	Reference
NiMoO _x /NiMoS	220	34	Nat. Commun. 11 , 5462 (2020).
MoS ₂ /Co ₉ S ₈ /Ni ₃ S ₂ /Ni	420	58	J. Am. Chem. Soc. 141 , 10417-10430 (2019)
Ni ₂ P-Ni ₃ S ₂ HNAs/NF	340	62	Nano Energy 51 , 26-36 (2019)
Ni ₂ P/NF	316	75	Nano Res. 13 , 2098-2105 (2020)
CoMoO ₄ -Ni(OH) ₂	349	68	ACS Sustainable Chem. Eng. 6 , 16086 (2018)
Co ₃ O ₄ -NP/N-rGO	380	62	Adv. Energy Mater. 8 , 1702222 (2018)
CoFe ₂ O ₄ @N-CNFs	349	80	Adv. Sci. 4 , 1700226 (2017)
CF/VGSs/MoS ₂	450	113.1	Nat. Commun. 12 , 1380 (2021)
CF/VMFO	200	29.9	Nat. Commun. 12 , 1380 (2021)
(MoS ₂) _{0.125} Mo ₂ C	~	209	Adv. Mater. Interface 6 , 1900948 (2019)
Fe-MoS ₂ /Ni ₃ S ₂ /NF	360	78.9	Dalton Trans. 48 , 12186-12192 (2019)
CoS ₂ -C@MoS ₂	500	46	ACS Sustainable Chem. Eng. 7 , 2899 (2019)
Co-MoS ₂ /BCCF-21	370	85	Adv. Mater. 30 , 1801450 (2018)

MoS ₂ /NiS ₂ /CC	384	75	Electrochim. Acta 385 , 138438 (2021)
1T _{0.81} -MoS ₂ @Ni ₂ P	400	56	This work
1T _{0.72} -MoS ₂ @NiS ₂	420	57	This work

Figure R12. **a** Plot of current density versus time at constant overpotential for 1T_{0.81}-MoS₂@Ni₂P in 1.0 M KOH electrolyte for 16 h, showing its excellent stability of the performance during continuous tests. **b** LSV curves of durability of 1T_{0.81}-MoS₂@Ni₂P. **c** Plot of current density versus time at constant overpotential for 1T_{0.72}-MoS₂@NiS₂ in 1.0 M KOH electrolyte for 16 h, showing its excellent stability of the performance during continuous tests. **d** LSV curves of 1T_{0.72}-MoS₂@NiS₂ for the durability test. (All LSV curves were corrected without iR-compensation).

References:

- [1] Chang, B. et al. Bimetallic NiMoN nanowires with a preferential reactive facet: an ultraefficient bifunctional electrocatalyst for overall water splitting. *Chem. Sus. Chem.* **11**, 3198–3207 (2018).
- [2] Zhang, J. et al. Synergistic interlayer and defect engineering in VS₂ nanosheets toward efficient electrocatalytic hydrogen evolution reaction. *Small* **14**, 1703098 (2018).

(2) **Electrocatalytic performance for overall water splitting.** Based on the above results, the overall water splitting measurement was carried out in a standard two-electrode system by using 1T_{0.81}-MoS₂@Ni₂P (or 1T_{0.72}-MoS₂@NiS₂) heterostructure catalyst as cathode and 1T_{0.81}-MoS₂@Ni₂P (or 1T_{0.72}-MoS₂@NiS₂) catalyst as anode in alkaline medium (1.0 M KOH solution), respectively. For

comparison purposes, $\text{CC@IrO}_2(+)//\text{CC@Pt/C}(-)$ was also prepared and tested in 1.0 M KOH solution. **Figure R13a** shows the polarization curves of the above two water-splitting electrolyzers. It was found that the $1\text{T}_{0.72}\text{-MoS}_2@\text{NiS}_2(+)//1\text{T}_{0.72}\text{-MoS}_2@\text{NiS}_2(-)$ and $1\text{T}_{0.81}\text{-MoS}_2@\text{Ni}_2\text{P}(+)//1\text{T}_{0.81}\text{-MoS}_2@\text{Ni}_2\text{P}(-)$ cells could require a cell voltage of 1.62 V and 1.86V at a current density of $20 \text{ mA}\cdot\text{cm}^{-2}$, respectively, while the voltages of 1.92 V is required for the $\text{IrO}_2(+)//\text{Pt/C}(-)$ cell. Clearly, the overall water splitting performance of as-prepared catalysts is better than that of $\text{IrO}_2(+)//\text{Pt/C}(-)$, especially at high current density. The water splitting potential of 1.62 V at the current density of $20 \text{ mA}\cdot\text{cm}^{-2}$ is smaller than those of most reported electrocatalysts, such as $\text{MoS}_2/\text{NiS}_2(+)//\text{MoS}_2/\text{NiS}_2(-)$ (1.63 V)¹, $\text{MoS}_2/\text{NiS}\text{ NCs}(+)//\text{MoS}_2/\text{NiS}\text{ NCs}(-)$ (1.71 V)², $\text{MoS}_2/\text{Co}_9\text{S}_8/\text{Ni}_3\text{S}_2(+)//\text{MoS}_2/\text{Co}_9\text{S}_8/\text{Ni}_3\text{S}_2(+)$ (1.70 V)³, $\text{Fe-Ni@NC-CNFs}(+)//\text{Fe-Ni@NC-CNFs}(-)$ (1.88 V)⁴, $\text{NC/NiMo/NiMoO}_x(+)//\text{NC/NiMo/NiMoO}_x(-)$ (1.64 V)⁵, and $\text{N-NiMoO}_4/\text{NiS}_2(+)//\text{N-NiMoO}_4/\text{NiS}_2(-)$ (1.70 V)⁶ (**Table R5**). Currently, most of the water splitting electrocatalysts reported require voltages higher than 1.62 V to reach $20 \text{ mA}\cdot\text{cm}^{-2}$ (with *iR*-compensation). It should be noticed that our OER electrocatalyst $1\text{T}_{0.72}\text{-MoS}_2@\text{NiS}_2$ is much better than the HER electrocatalyst $1\text{T}_{0.81}\text{-MoS}_2@\text{Ni}_2\text{P}$. The working stability of our electrocatalysts during overall water splitting was tested (**Figure R13b**), which indicates that the voltage keeps almost unchanged after 16 h at the current densities of $10 \text{ mA}\cdot\text{cm}^{-2}$, indicating excellent stability.

Figure R13. Overall water splitting activities of the samples in 1.0 M KOH solution. **a** Polarization curves of $1\text{T}_{0.81}\text{-MoS}_2@\text{Ni}_2\text{P}(+)//1\text{T}_{0.81}\text{-MoS}_2@\text{Ni}_2\text{P}(-)$, $1\text{T}_{0.72}\text{-MoS}_2@\text{NiS}_2(+)//1\text{T}_{0.72}\text{-MoS}_2@\text{NiS}_2(-)$ and $\text{IrO}_2//\text{Pt/C}$ at a scan rate of $5 \text{ mV}\cdot\text{s}^{-1}$. **b** Catalytic stability of $1\text{T}_{0.81}\text{-MoS}_2@\text{Ni}_2\text{P}(+)//1\text{T}_{0.81}\text{-MoS}_2@\text{Ni}_2\text{P}(-)$ at $10 \text{ mA}\cdot\text{cm}^{-2}$ tested in a two-electrode configuration. (All LSV curves were corrected without *iR*-compensation).

Table R5. Comparison of water-splitting activities of $1\text{T}_{0.81}\text{-MoS}_2@\text{Ni}_2\text{P}(+)//1\text{T}_{0.81}\text{-MoS}_2@\text{Ni}_2\text{P}(-)$, $1\text{T}_{0.72}\text{-MoS}_2@\text{NiS}_2(+)//1\text{T}_{0.72}\text{-MoS}_2@\text{NiS}_2(-)$ cell in this work with the other reported electrocatalysts in 1.0 M KOH solution (V_{20} -cell voltage at $20 \text{ mA}\cdot\text{cm}^{-2}$). *The data were calculated according to the curves (LSV curves were corrected with *iR*-compensation) given in the literature.

Catalysts	Support	V ₂₀ (V)	Reference
CF/VMFO(+) CF/VMFO(-)	Carbon fibers	1.48*	Nat. Commun. 12 , 1380 (2021)
Cu@NiFe LDH(+) Cu@NiFe LDH(-)	Cu foam	1.58*	Energy Environ. Sci. 10 , 1820-1827 (2017)
MoS ₂ /NiS ₂ (+) MoS ₂ /NiS ₂ (-)	Carbon cloth	1.63*	Adv. Sci. 6 , 1900246 (2019)
MoS ₂ /NiS NCs(+) MoS ₂ /NiS NCs(-)	Ni foam	1.71*	J. Mater. Chem. A 6 , 9833-9838 (2018)
MoS ₂ /Co ₉ S ₈ /Ni ₃ S ₂ (+) MoS ₂ /Co ₉ S ₈ /Ni ₃ S ₂ (+)	Ni foam	1.70*	J. Am. Chem. Soc. 141 , 10417-10430 (2019)
NiMoO _x /NiMoS(+) NiMoO _x /NiMoS(-)	~	1.54*	Nat. Commun. 11 , 5462 (2020)
Ni ₂ P-Ni ₃ S ₂ HNAs(+) Ni ₂ P-Ni ₃ S ₂ HNAs(-)	Ni foam	1.54*	Nano Energy 51 , 26-36 (2018)
Fe-Ni@NC-CNFs(+) Fe-Ni@NC-CNFs(-)	Carbon fibers	1.88*	Angew. Chem. Int. Ed. 57 , 8921-8926 (2018)
Ni@N-C(+) Ni@N-C(-)	~	1.78*	Inorg. Chem. 60 , 6764-6771 (2021)
NC/NiMo/NiMoO _x (+) NC/NiMo/NiMoO _x (-)	Ni foam	1.64*	Small 13 , 1702018 (2017)
CoNi(OH) _x (+) NiN _x (-)	Ni foam	1.73*	Adv. Energy Mater. 6 , 1501661 (2016)
N-NiMoO ₄ /NiS ₂ (+) N-NiMoO ₄ /NiS ₂ (-)	Ni foam	1.70*	Adv. Funt. Mater. 29 , 1805298 (2019)
Co _x PO ₄ /CoP(+) Co _x PO ₄ /CoP(-)	~	1.91*	Adv. Mater. 27 , 3175-3180 (2015)
IT_{0.81}-MoS₂@Ni₂P(+)//IT_{0.81}-MoS₂@Ni₂P(-)	Carbon cloth	1.86	This work
IT_{0.72}-MoS₂@NiS₂(+)//IT_{0.72}-MoS₂@NiS₂(-)	Carbon cloth	1.62	This work

Reference:

- [1] Lin, J. et al. Defect Rich Heterogeneous MoS₂/NiS₂ nanosheets electrocatalysts for efficient overall water splitting[J]. Adv. Sci. **6**, 1900246 (2019).
- [2] Zhai, Z. et al. Dimensional construction and morphological tuning of heterogeneous MoS₂/NiS electrocatalysts for efficient overall water splitting[J]. J. Mater. Chem. A, **6**, 9833-9838 (2018).
- [3] Yang, Y. et al. Hierarchical nano-assembly of MoS₂/Co₉S₈/Ni₃S₂/Ni as a highly efficient electrocatalyst for overall water splitting in a wide pH range[J]. J. Am. Chem. Soc. **141**, 10417-10430 (2019).
- [4] Zhao, X. et al. Bifunctional electrocatalysts for overall water splitting from an iron/nickel-based bimetallic metal-organic framework/dicyandiamide composite[J]. Angew. Chem. Int. Ed. **57**, 8921-8926 (2018).
- [5] Hou, J. et al. Active sites intercalated ultrathin carbon sheath on nanowire arrays as integrated core-shell architecture: highly efficient and durable electrocatalysts for overall water splitting[J]. Small **13**, 1702018 (2017).
- [6] An, L. et al. Epitaxial heterogeneous interfaces on N□NiMoO₄/NiS₂ nanowires/nanosheets to boost hydrogen and oxygen production for overall water splitting[J]. Adv. Funt. Mater. **29**, 1805298 (2019).

Comment 11: I am wondering if there could be leaching of some components (phosphides, sulfides.) during catalysis especially in alkaline medium. May the

authors conduct some post catalytic characterizations in this direction?

Response 11: We thank for the reviewer's helpful suggestions. Indeed, after 16 hours of HER stability test on our samples, some phosphides or sulfides fell off the catalyst in alkaline medium (**Figure R14**). Therefore, we performed XPS analysis on the samples after 16 hours of HER stability test. Elemental analysis of the as-prepared catalyst samples before and after the HER stability (16 hours) test, as shown in **Table R6**. Obviously, the contents of Mo, Ni, S, P in $1T_{0.81}\text{-MoS}_2\text{@Ni}_2\text{P}$ and $1T_{0.72}\text{-MoS}_2\text{@NiS}_2$ samples have all decreased after 16 hours of HER stability test (**Table R6**). In contrast, there are negative-shift of about 0.8 eV for two distinctive peaks of Mo $3d_{5/2}$ (228.44 eV) and Mo $3d_{3/2}$ (232.37 eV) for $1T_{0.81}\text{-MoS}_2\text{@Ni}_2\text{P}$ sample after HER stability test (**Figure R15a**), compared with that in the pristine $1T_{0.81}\text{-MoS}_2\text{@Ni}_2\text{P}$ sample. With regard to S 2p regions, two peaks S $2p_{3/2}$ (161.79 eV) and S $2p_{1/2}$ (163.28 eV) are also negative-shifted of about 0.3 eV (**Figure R15b**). **Figure R15c** shows the P 2p core level XPS spectrum, in which the peak located at 129.06 eV and 130.60 eV is due to the Ni-P bonding, whereas the another one at 133.20 eV is associated with the P-O bonding of phosphate. With regard to Ni 2p regions, the peaks of Ni $2p_{3/2}$ and the corresponding satellite appear at 855.51 and 861.09 eV, respectively. The peaks of Ni $2p_{1/2}$ and the corresponding satellite appear at 873.47 and 880.50 eV, respectively. These are also negative-shifted of about 0.7 eV and a new peak Ni^0 (851.90 eV) appears for $1T_{0.81}\text{-MoS}_2\text{@Ni}_2\text{P}$ sample after HER stability test, indicating that Ni was reduced during the HER process. For the $1T_{0.72}\text{-MoS}_2\text{@NiS}_2$ sample after HER stability test, we also obtained the similar results. All these results indicate that electron transfer occurs between Mo and Ni to promote HER activity in the catalyst. For other related characterizations after acidic medium of HER, such as Raman, XRD, and XPS, please see the **Response 7 of Reviewer #2** for details. We added this evidence and the corresponding figure in the revised manuscript.

Figure R14. Electrolyte photos of our sample after 16 hours of HER stability test. **a** $1T_{0.72}\text{-MoS}_2\text{@NiS}_2$, **b** $1T_{0.81}\text{-MoS}_2\text{@Ni}_2\text{P}$.

Table R6. Elemental analyses (atomic %) of the as-prepared catalysts before and after

the HER stability (16 hours) test.

HER	$1T_{0.81}\text{-MoS}_2\text{@Ni}_2\text{P}$				$1T_{0.72}\text{-MoS}_2\text{@NiS}_2$		
	Mo 3d	Ni 2p	S 2p	P 2p	Mo 3d	Ni 2p	S 2p
Before	19.29 %	1.78 %	32.70 %	8.32 %	13.96 %	4.39 %	36.96 %
After	1.21 %	0.96 %	11.71 %	2.36 %	2.78 %	1.08 %	15.46 %

Figure R15. XPS spectra for $1T_{0.81}\text{-MoS}_2\text{@Ni}_2\text{P}$ heterostructure catalyst after 16 h stability measurement at 10 mA cm^{-2} in 1.0 M KOH solution. (a) Mo 3d, (b) S 2p, (c) P 2p, (d) Ni 2p.

Figure R16. XPS spectra for $1T_{0.72}\text{-MoS}_2\text{@NiS}_2$ heterostructure catalyst after 16 h stability measurement at 10 mA cm^{-2} in 1.0 M KOH solution. (a) Mo 3d, (b) S 2p, (c) Ni 2p.

Comment 12: Figure S2 a and b are identical. Please cancel one of the two.

Response 12: We thank the reviewer for his (her) suggestions. It is indeed due to our negligence, we have replaced one of them, please see **Supplementary Fig. 2** in revised Supplementary Information for details.

Comment 13: Figures 4, 5 show that pure 1T phase performs less than mixed phase ($1T_{0.41}\text{-MoS}_2$). However, the DFT ignores this mixed phase simulations and only considered pure 1T or pure 2H phases. In fact, one of the main concepts elucidated in the experiment is “multi-heterogeneous-interface”. Is there any special reason (maybe too complicate or too heavy computational time required) for the omission of mixed phase (1T and 2H) in the DFT study?

Response 13: Thanks for the comments and suggestions from the reviewer. In fact, **Fig. 4** show that pure 1T phase performs more than mixed phase ($1T_{0.41}\text{-MoS}_2$). It is because our methods of preparing these are different, and there may be some differences in results. Please see your **Comment 7** and **Response 7** for details. We agree with the reviewer's point of view. Indeed, the local atomic sites or electronic structure of the 1T-2H MoS_2 homojunction interface that also exists in our system has an impact on the HER reaction. If 1T-2H homogeneous interface is considered, the our as-prepared catalyst system is indeed very complicated, and the theoretical

calculation is very computationally intensive and time-consuming. We have also tried to calculate the complex system. Unfortunately, we have not obtained the desired result so far. In addition, our topic is focused on the interface. So, we only concentrate on the interface based on the experimental observation, it is found that the interface is composed by two pure phases at the most situations, so we selected the pure phase as our model for the clearly investigating the interface effect. For example, it is indeed the performance of $1T_{0.81}\text{-MoS}_2\text{@Ni}_2\text{P}$ is outstanding and it is surely the three-phase containing in, but the interface is mostly composed by only two phases, so we did the two-phase connection regardless of three-phase calculation because the three-phase-site is negligibly existed in our catalysts.

1T-2H interface is very important for HER performance and we added the calculation for 1T-2H interface in this work. Also, we did the computation in alkaline situation. Actually, we missed 1T-2H interface calculation, so here the authors thanks for your suggestion in terms of work integration. Moreover, the concrete effects of local atomic sites or electronic structure at 1T-2H MoS_2 homojunction interface also exists in the system. Please see **Response 3** of **Reviewer#2** for details.

Reviewer #2: (Remarks to the Author):

The second reviewer comment and response

List of changes (purple in the revised manuscript)

Comment: This work proposes a synergistic strategy for the design of efficient Mo-based HER electrocatalyst. By combining phase transition, defect engineering as well as interface engineering together, a series of multi-heterojunction-interface electrocatalysts have been synthesized (e.g., $1T_{0.81}\text{-MoS}_2\text{@Ni}_2\text{P}$ and $1T_{0.72}\text{-MoS}_2\text{@NiS}_2$), yielding highly improved hydrogen evolution reaction (HER) performance. In particular, the $1T_{0.81}\text{-MoS}_2\text{@Ni}_2\text{P}$ or $1T_{0.72}\text{-MoS}_2\text{@NiS}_2$ catalysts with heterogeneous-phase interfaces possess a low onset overpotential and Tafel slope, comparative to many other recently-reported Mo-based catalysts. I think this work caters to the readership in electrocatalysis and electrochemistry fields and will arouse the interests in interface engineering induced electronic structure modulation for HER performance improvement. Thus, I recommend that this manuscript should be published in Nature Communications after addressing the following minor revisions:

Our response: We are grateful to the reviewer's overall comments and positive recommendations. We have made careful revisions according to each comment, as summarized below.

Comment 1: The authors declared the sulfur or phosphorus vapor thermal treatment can improve the conversion rate of 2H to 1T phase of MoS₂ substantially from 41% to over 70%. Then, what's the possible mechanism governing such interesting phenomenon? Can the authors provide some personal insights about it from phase free-energy change or other theoretical perspectives?

Response 1: Thanks for the comments and suggestions from the reviewer. Currently, various strategies have been proposed to realize 2H→1T phase transformation, including electron beam scanning,¹ ion intercalation (such as Li⁺, tert-butyllithium),² mechanical strain,³ and element doping.⁴ In recent years, single N doping or single PO₄³⁻ intercalation methods have been verified as effective ways to induce phase transformation in MoSe₂ or MoS₂ systems (2H→1T).^{4,5} As theoretically calculated by Ye et al., red Phosphorus is a light-weight nonmetal element with one unpaired electron, possessing the ability to occupy the interspace between S–Mo–S layers of 2H MoS₂.⁶ As experimentally demonstrated by Wang et al., red phosphorus vapor can be facily inserted into the interlayer of 2H MoS₂ bulk to amplify the interlayer spacing to 0.63 nm and even can exfoliate the bulk MoS₂ to thin planes. More importantly, phosphorus can simultaneously embed into the S–Mo–S atomic planes, inducing the glide of S atomic planes around the P doping regions, affording in-plane heterostructures with the built form of 1T and 2H MoS₂ domains.⁷ Here, to exemplify the phase-change tendency, DFT calculation was employed for comparing the phase-changing free energy of 2H→1T and 2H(P) →1T(P) to simulate the process. The results reveal that if our 2H-MoS₂ contains P, the phase transformation energy became relatively low, which means if we use P vapor to treat 2H-MoS₂, the process for 2H to 1T will become easier than normal situation.

Figure R17. DFT calculation for comparing the phase changing free energy of 2H→1T and 2H(P) →1T(P) to simulate the process.

References:

[1] Lin, Y. C. et al. Atomic mechanism of the semiconducting-to-metallic phase

- transition in single-layered MoS₂[J]. Nat. Nanotechnol. **9**, 391–396 (2014).
- [2] Zeng, Z. Y. et al. Single-layer semiconducting nanosheets: high-yield preparation and device fabrication[J]. Angew. Chem. Int. Ed. **50**, 11093–11097 (2011).
- [3] N. Duerloo, K. -A. et al. Structural phase transitions in two-dimensional Mo- and W-dichalcogenide monolayers[J]. Nat. Commun. **5**, 4214 (2014).
- [4] Deng, S. J. et al. Directional construction of vertical nitrogen-doped 1T-2H MoSe₂/Graphene shell/core nanoflake arrays for efficient hydrogen evolution reaction[J]. Adv. Mater. **29**, 1700748 (2017).
- [5] Deng, S. J. et al. Synergistic doping and intercalation: a new way to realize deep phase modulation on MoS₂ arrays for high-efficiency hydrogen evolution reaction[J]. Angew. Chem. Int. Ed. **58**, 16289–16296 (2019).
- [6] Ye, L. J. et al. Tuning the electrical transport properties of multilayered molybdenum disulfide nanosheets by intercalating phosphorus[J]. J. Phys. Chem. C **119**, 9560–9567 (2015).
- [7] Wang, S. et al. Ultrastable in-plane 1T–2H MoS₂ heterostructures for enhanced hydrogen evolution reaction[J]. Adv. Energy Mater. **8**, 1801345 (2018).

Comment 2: In the theoretical calculation and mechanism analysis part, the authors only carried out hydrogen adsorption energy (ΔG_{H^*}) calculation which is mainly effective for HER property evaluation in acid media. However, as the water dissociation step usually takes a dominant role in alkaline HER process, it's necessary to carefully analyze the concrete water dissociation, subsequent intermediate evolution steps as well as the interface engineering effect on modulating these steps, including the water dissociation energy barrier, the explicit water dissociation related active sites and rate-limiting step for the whole alkaline HER process.

Response 2: Thanks for the comments and suggestions from the reviewer. According to your suggestion, we have added relevant information in our revised manuscript (**Fig. 4f** and **Supplementary Fig. 42**).

Figure R18. The optimized structures and Free-energy diagrams for HER on the pure NiS_2 , pure Ni_2P , $2\text{H}_{\text{phase}}\text{-MoS}_2$, $1\text{T}/2\text{H}_{\text{mix}}\text{-MoS}_2$, $1\text{T}_{\text{phase}}\text{-MoS}_2$, $2\text{H}_{\text{phase}}\text{-MoS}_2@ \text{Ni}_2\text{P}$, $2\text{H}_{\text{phase}}\text{-MoS}_2@ \text{NiS}_2$, $1\text{T}_{\text{phase}}\text{-MoS}_2@ \text{NiS}_2$ and $1\text{T}_{\text{phase}}\text{-MoS}_2@ \text{Ni}_2\text{P}$ interface edge.

Figure R19. Free-energy diagrams for HER on the $2\text{H}_{\text{phase}}\text{-MoS}_2$, $1\text{T}_{\text{phase}}\text{-MoS}_2$, pure Ni_2P , pure NiS_2 , $1\text{T}/2\text{H}_{\text{mix}}\text{-MoS}_2$, $2\text{H}_{\text{phase}}\text{-MoS}_2@ \text{Ni}_2\text{P}$, $2\text{H}_{\text{phase}}\text{-MoS}_2@ \text{NiS}_2$, $1\text{T}_{\text{phase}}\text{-MoS}_2@ \text{NiS}_2$ and $1\text{T}_{\text{phase}}\text{-MoS}_2@ \text{Ni}_2\text{P}$ interface edge.

Modification to the manuscript:

“To reveal further the relationship of HER activity of catalysts with phase structure and heterojunction-interface, we used DFT to calculate the optimized structures and free-energy diagrams for HER on $2\text{H}_{\text{phase}}\text{-MoS}_2$, $1\text{T}_{\text{phase}}\text{-MoS}_2$, $1\text{T}/2\text{H}_{\text{mix}}\text{-MoS}_2$, pure

Ni₂P, pure NiS₂, 2H_{phase}-MoS₂@NiS₂, 2H_{phase}-MoS₂@Ni₂P, 1T_{phase}-MoS₂@NiS₂, and 1T_{phase}-MoS₂@Ni₂P catalysts with partially multi-heterojunction interface modification. As shown in **Fig. 4f** and **Supplementary Fig. 42**, the reaction pathway for alkaline HER, including prior water dissociation to form H* intermediates (Volmer step) and hydrogen generation (Tafel step or Heyrovsky step), is constructed.^{60,61} However, the energy of the intermediate state H*(ΔG_{H^*}) is a critical indicator of the ability of hydrogen evolution (Tafel step or Heyrovsky step).^{35,60} **Fig. 4f** displays the calculated free-energy diagram on the most stable energy of the 2H_{phase}-MoS₂, 1T_{phase}-MoS₂, Ni₂P, NiS₂, 2H_{phase}-MoS₂@Ni₂P, 2H_{phase}-MoS₂@NiS₂, 1T_{phase}-MoS₂@NiS₂ and 1T_{phase}-MoS₂@Ni₂P catalysts (**Supplementary Fig. 41-42**). For 2H_{phase}-MoS₂, the ΔG_{H^*} is very positive (1.49 eV), indicating that there is a strong interaction between H* and 2H_{phase}-MoS₂, showing poor HER reaction kinetics. More importantly, MoS₂ exhibits unfavorable catalyst-OH_{ad} energetics, exhibiting relatively high activated water-adsorption energy ($\Delta G_{H_2O} = 0.82$ eV). The high ΔG_{H_2O} significantly hinders the decomposition of water into H* intermediates and results in slow HER kinetics. The introduction of the 1T/2H_{mix}-phase into MoS₂ can obviously decrease the value of ΔG_{H^*} to 0.97 eV and ΔG_{H_2O} to 0.16 eV, implying the promoted HER activity compared with 2H_{phase}-MoS₂. Notably, constructing multi-heterointerface interface edges active sites with NiS₂ can provide the active sites for hydroxyl adsorption, and the followed ΔG_{H_2O} and ΔG_{H^*} are reduced to 0.10 eV and -0.12 eV on the 1T_{phase}-MoS₂@NiS₂ interface, indicating the 1T/2H_{mix}-phase and NiS₂ nanoparticles are effective for cleaving HO-H bonds and weaker interaction between H*. Moreover, the charge transfer from Ni₂P to the MoS₂ is confirmed by the DFT calculations, and leads to a more optimal ΔG_{H^*} value about -0.09 eV. Hence, the NiS₂ (or Ni₂P) can act a water dissociation promoter and product hydrogen intermediates that then adsorb on nearby MoS₂ catalyst sites. In this way, the multi-heterointerface interface can also accelerate subsequent H₂ generation. The reaction pathways on the single side (such as Ni₂P, NiS₂ and MoS₂) of the interface have also been shown in **Fig. 4f** and **Supplementary Fig. 42**. These both show much unfavorable energetics than that of synergetic pathway on MoS₂@NiS₂ or MoS₂@Ni₂P interface.”

References:

- [35] Zhang, B. et al. Interface engineering: the Ni(OH)₂/MoS₂ heterostructure for highly efficient alkaline hydrogen evolution. *Nano Energy* **37**, 74-80 (2017).
- [60] Zhang, J. et al. Engineering water dissociation sites in MoS₂ nanosheets for accelerated electrocatalytic hydrogen production. *Energy Environ. Sci.* **9**, 2789-2793 (2016).
- [61] Wu, Y. et al. Electron density modulation of NiCo₂S₄ nanowires by nitrogen incorporation for highly efficient hydrogen evolution catalysis. *Nat. Commun.* **9**, 1425 (2018).

Comment 3: In the mechanism analysis part, the authors focused on the local electronic structure change at MoS₂-NiS₂ heterojunction interface. To go a little

further, what about the concrete effects of local atomic sites or electronic structure at 1T-2H MoS₂ homojunction interface which also exists in the system? Can the authors provide some corresponding explanation by experiment or simulation?

Response 3: Thanks for the comments and suggestions from the reviewer. We agree with the issue raised by the reviewer. Indeed, in our system, the existence of 1T-2H MoS₂ homojunction intrinsic atomic sites or electronic structure may promote the HER reaction process. Metallic 1T-MoS₂ is highly desirable for catalyzing electrochemical hydrogen production from water owing to its high electrical conductivity. However, stable pure 1T-MoS₂ is difficult to be obtained in large-scale by either common chemical or physical approaches. Therefore, what we obtain through various methods is often a stable in-plane 1T-2H MoS₂ heterojunction catalyst. Benefiting from its significantly improved electrical conductivity, and highly exposed active sites, in-plane 1T-2H MoS₂ heterostructures exhibit largely-improved electrocatalytic properties for hydrogen evolution reaction (HER) in alkaline electrolytes. Here, we also believe that the 1T-2H MoS₂ heterojunction promotes the electrocatalytic hydrogen evolution reaction mainly caused by increasing the electrical conductivity and increasing its active sites. For the number of active sites of our catalyst, we have calculated them by cyclic voltammetry (CV), please see **Response 2** of **Reviewer#1** for details. Above-mentioned phase conversion of MoS₂ can be further demonstrated by the electrical measurement with a ST2263 double electric measurement digital four-probe tester as shown in **Figure R20**. Clearly, the in-plane 1T-2H MoS₂ heterostructures (1T_{0.41}-MoS₂) exhibit a high electrical conductivity of 8718 S m⁻¹, which is more than 8.3 times higher than those of 2H_{phase}-MoS₂ and carbon cloth (956 and 1045 S m⁻¹). In addition, we calculated their bandgap energy and DOS through theoretical calculations (**Supplementary Fig. 34.**), and it is found that the bandgap of the 1T-2H homojunction interface is significantly narrowed, implying that its conductivity is improved.

Figure R20. Electrical conductivity of $1T_{0.41}\text{-MoS}_2$, $1T_{0.72}\text{-MoS}_2\text{@NiS}_2$, $1T_{0.81}\text{-MoS}_2\text{@Ni}_2\text{P}$ heterostructures compared with $2H_{\text{phase}}\text{-MoS}_2$ and carbon cloth.

Comment 4: As to the HER mechanism, the as-generated H_{ads} could be adsorbed on $1T_{0.81}\text{-MoS}_2\text{@Ni}_2\text{P}$ (and $1T_{0.72}\text{-MoS}_2\text{@NiS}_2$) and further be converted into H_2 readily through the Tafel step (Page 17, line 12-22). However, the free energy of NiS_2 and Ni_2P for hydrogen adsorption was not calculated in the diagram (Fig. 4f).

Response 4: Thanks for the comments and suggestions from the reviewer. According to your suggestion, we have added relevant information in our revised manuscript (Fig. 4f). Please see your **comment 2** and corresponding **response 2** for details.

Comment 5: Was the computation for NiS_2 spin-polarized? If so, please provide another orbital for your computation of DOS/energy band.

Response 5: First of all, we thank the reviewer for the question. Yes, the calculation was under spin-polarized background. The spin-polarized DOS & Band results have been added in our revised Supplementary Information.

Figure R21. Band structure and spin-polarized density of states (DOS) for NiS_2 .

Comment 6: As we seen, under low current density conditions, $1T_{0.81}\text{-MoS}_2\text{@Ni}_2\text{P}$ and $1T_{0.72}\text{-MoS}_2\text{@NiS}_2$ electrodes exhibit excellent HER performance; however, under high current density conditions (such as more than 250 mA cm^{-2}), whether they still have excellent HER performance? The authors should add this part of the data to further explain it.

Response 6: We thank for the reviewer's helpful suggestions. According to your suggestion, we have added the HER performance of as-prepared catalysts under high

current density in our revised manuscript. As shown in **Figure R22**, obviously, the as-prepared catalysts still exhibit excellent HER performance under high current density, and are better than commercial 20% Pt/C. Unfortunately, due to the poor conductivity of carbon cloth, the current on the surface of the catalyst is unstable at high current densities.

Figure R22. a The HER polarization curves of 1T_{0.81}-MoS₂@Ni₂P, 1T_{0.72}-MoS₂@NiS₂ and 20% Pt/C electrodes at 5.0 mV s⁻¹ in 1.0 M KOH solution. b The HER polarization curves of 1T_{0.81}-MoS₂@Ni₂P, 1T_{0.72}-MoS₂@NiS₂ and 20% Pt/C electrodes at 5.0 mV s⁻¹ in 0.5 M H₂SO₄ solution. (All polarization curves were corrected without iR-compensation).

Comment 7: In this manuscript, 1T_{0.81}-MoS₂@Ni₂P and 1T_{0.72}-MoS₂@NiS₂ electrodes exhibit the fastest HER processes and better reactivity which is attributed to the multi-heterogeneous interface effect, a large number of defects, and a higher proportion of 1T phase-MoS₂, and these also show excellent long-term stability after i-t curve measurement of 16 h. The authors should add some characterizations (such as XRD, Raman, XPS measurement) of the electrode materials after the HER test to further prove the stability of their multi-heterojunction interfaces catalysts, including the 1T phase percentage, chemical composition and their elemental valences of the electrode materials after the HER reaction. Whether these has changes?

Response 7: Thanks for the comments and suggestions from the reviewer. According to your suggestion, we have added some SEM, XRD, Raman and XPS characterization for 1T_{0.81}-MoS₂@Ni₂P and 1T_{0.72}-MoS₂@NiS₂ after 16 h stability measurement in the revised Supplementary Information. Construction of a stable nanostructured electrocatalyst is of great importance for optimizing the electrochemical performance of HER. The morphologies of the 1T_{0.81}-MoS₂@Ni₂P and 1T_{0.72}-MoS₂@NiS₂ catalysts still remain flower-shaped MoS₂ microspheres (shown in **Figure R23**, SEM image) after the durability measurements, which reveals that the MoS₂ microspheres nanosheet structure can reduce the disintegration tendency during the alternate processes of bubble accumulation and bubble release. In addition, the XRD patterns of the 1T_{0.81}-MoS₂@Ni₂P and 1T_{0.72}-MoS₂@NiS₂ microspheres remain the original crystallographic structure after the long-term test

(shown in **Figure R24**). To better understand the activation mechanism, we examined the Raman, and XPS of $1T_{0.81}\text{-MoS}_2\text{@Ni}_2\text{P}$ and $1T_{0.72}\text{-MoS}_2\text{@NiS}_2$ sample before and after the durability measurements. The Raman spectroscopy of the $1T_{0.81}\text{-MoS}_2\text{@Ni}_2\text{P}$ shows obviously stronger and blueshifts after the long-term test (**Figure R24a**), due to strong susceptibility to the influence of electron–phonon coupling.¹ Raman spectra of the $1T_{0.72}\text{-MoS}_2\text{@NiS}_2$ also do not show any new peaks after HER long-term testing (**Figure R24b**). All of these confirm the good electrochemical stability of the as-prepared catalysts in the HER process under alkaline and acidic media. In addition, chemical stabilities during the durability measurements were also detected by XPS. For the $1T_{0.81}\text{-MoS}_2\text{@Ni}_2\text{P}$ heterostructure catalyst, two peaks located at 229.54 eV (Mo 3d_{5/2}) and 233.0 eV (Mo 3d_{3/2}) characteristic of MoS₂ are observed (**Figure R26a**). After 16 h stability measurement at 10 mA cm⁻², the Mo 3d spectra change and are shifted obviously (**Figure R27a**). The $1T_{0.81}\text{-MoS}_2\text{@Ni}_2\text{P}$ is also supported by the XPS measurement, which shows a shift of around + 0.3 eV in S 2p peaks after 16 h stability measurement (**Figure R27b**). This shift in binding energy is consistent with what was previously observed for p-doped MoS₂.^{2,3} With regard to Ni 2p regions, the peaks of Ni 2p_{3/2} and the corresponding satellite appear at 857.4 and 862.8 eV, respectively. The peaks of Ni 2p_{1/2} and the corresponding satellite appear at 875.6 and 880.2 eV, respectively. These are also negative-shifted of about 0.8 eV and a new peak of Ni⁰ (852.5 eV) is appeared for $1T_{0.81}\text{-MoS}_2\text{@Ni}_2\text{P}$ sample after HER stability test (**Figure R27**), indicating that Ni was reduced during the HER process. For the $1T_{0.72}\text{-MoS}_2\text{@NiS}_2$ sample after HER stability test, we also obtained similar results (**Figure R28** and **Figure R29**). All these results indicate that electron transfer occurs between Mo and Ni to promote HER activity in the catalyst.

Figure R23. a-b SEM images of $1T_{0.81}\text{-MoS}_2\text{@Ni}_2\text{P}$ after 16 h stability measurement at 10 mA cm^{-2} . c-d SEM images of $1T_{0.72}\text{-MoS}_2\text{@NiS}_2$ after 16 h stability measurement at 45 mA cm^{-2} . EDS mapping of Mo, Ni, and S elements of $1T_{0.72}\text{-MoS}_2\text{@NiS}_2$ (f-h) and overlap mapping of elements (e).

Figure R24. a XRD patterns for $1T_{0.72}\text{-MoS}_2\text{@NiS}_2$ catalyst before and after 16 h stability measurement at 45 mA cm^{-2} in $0.5\text{ M H}_2\text{SO}_4$ solution. b XRD patterns for $1T_{0.81}\text{-MoS}_2\text{@Ni}_2\text{P}$ catalyst before and after 16 h stability measurement at 10 mA cm^{-2} in $0.5\text{ M H}_2\text{SO}_4$ solution.

Figure R25. a Raman spectra of $1T_{0.81}\text{-MoS}_2@Ni_2P$ catalyst before and after 16 h stability measurement at 10 mA cm^{-2} in $0.5\text{ M H}_2\text{SO}_4$ solution. b Raman spectra of $1T_{0.72}\text{-MoS}_2@NiS_2$ catalyst before and after 16 h stability measurement at 45 mA cm^{-2} in $0.5\text{ M H}_2\text{SO}_4$ solution.

Figure R26. XPS spectra of $1T_{0.81}\text{-MoS}_2@Ni_2P$ heterostructure catalyst after 16 h stability measurement at 10 mA cm^{-2} in $0.5\text{ M H}_2\text{SO}_4$ solution. (a) Mo 3d, (b) S 2p, (c) P 2p, and (d) Ni 2p.

Figure R27. (a) S 2p, (b) Mo 3d, (c)P 2p and (d) Ni 2p XPS spectra of 1T_{0.81}-MoS₂@Ni₂P samples before and after HER cycling in 0.5 M H₂SO₄ solution. This figure was added as Figure S19 in the revised Supplementary Information.

Figure R28. XPS spectra of 1T_{0.72}-MoS₂@NiS₂ heterostructure catalyst after 16 h stability measurement at 45 mA cm⁻² in 0.5 M H₂SO₄ solution. (a) Mo 3d, (b) S 2p, (c) Ni 2p.

Figure R29. (a) S 2p, (b) Mo 3d, and (c) Ni 2p XPS spectra of 1T_{0.72}-MoS₂@NiS₂ samples before and after HER cycling in 0.5 M H₂SO₄ solution.

References:

- [1] Chakraborty, B. et al. Symmetry-dependent phonon renormalization in monolayer MoS₂ transistor[J]. Phys. Rev. B: Condens. Matter Mater. Phys. **85**, 161403 (2012).
- [2] Li, G. Q. et al. Activating MoS₂ for pH-Universal hydrogen evolution catalysis[J]. J. Am. Chem. Soc. **139**, 16194-16200 (2017).
- [3] Nipane, A. et al. Few-layer MoS₂ p-type devices enabled by selective doping using low energy phosphorus implantation[J]. ACS Nano, **10**, 2128-2137 (2016).

Comment 8: In Introduction Part (Page 2, line 7-10), the authors only cited Ref.[10] and Ref.[11] to describe “Pt-based electrocatalysts are recognized as.....above-mentioned process”. Here, I suggest that authors should add some references to further explain why Pt-based catalysts have fast reaction kinetics. In addition, in Introduction Part (Page 2, line 19-21), “the electrocatalytic activity of MoS₂ is closely associated with.....defect”, the author should cite several references about MoS₂-based catalysts to elaborate on the influence of phase structure, interface

and defects on the electrocatalytic performance of MoS₂.

Response 8: We thank to the reviewer's helpful comment. According to your suggestion, we have added relevant references in our revised manuscript.

Modification to the manuscript:

“Pt-based electrocatalysts are recognized as highly efficient electrocatalysts due to good electrical conductivity,¹⁰ fast kinetics^{11,12}, and the preference to overcome the large kinetic energy barrier involved in the above-mentioned process.¹³”

“However, the electrocatalytic activity of MoS₂ is closely associated with its surface electric structure,²⁶⁻³⁶ many researchers have focused on adjusting the electronic structure of the MoS₂ surface to promote electrocatalytic activity, such as surface engineering,²⁶ doping,²⁷ single-atom anchoring,²⁸ phase structure,²⁹⁻³³ interface active site,^{34,35} and defect.³⁶”

References:

- [11] Seh, Z. W. et al. Combining theory and experiment in electrocatalysis: insights into materials design. *Science* **355**, 1–12 (2017).
- [12] Morales-Guio, C. G. et al. Nanostructured hydrotreating catalysts for electrochemical hydrogen evolution[J]. *Chem. Soc. Rev.* **43**, 6555-6569 (2014).
- [26] Kibsgaard, K. et al. Engineering the surface structure of MoS₂ to preferentially expose active edge sites for electrocatalysis[J]. *Nat. Mater.* **11**, 963-969 (2012).
- [27] Sun, T. et al. Engineering the electronic structure of MoS₂ nanorods by N and Mn dopants for ultra-efficient hydrogen production[J]. *ACS Catal.* **8**, 7585–7592 (2018).
- [28] Zhang, H. et al. Surface modulation of hierarchical MoS₂ nanosheets by Ni single atoms for enhanced electrocatalytic hydrogen evolution[J]. *Adv. Funct. Mater.* **28**, 1807086 (2018).
- [29] Liu, Z. et al. Vertical nanosheet array of 1T phase MoS₂ for efficient and stable hydrogen evolution. *Appl. Catal. B: Environ.* **246**, 296-302 (2019).
- [30] Lei, C. et al. Efficient alkaline hydrogen evolution on atomically dispersed Ni-N_x Species anchored porous carbon with embedded Ni nanoparticles by accelerating water dissociation kinetics. *Energy Environ. Sci.* **12**, 149-156 (2019).
- [31] Wang, S. et al. Ultrastable In-Plane 1T–2H MoS₂ heterostructures for enhanced hydrogen evolution reaction. *Adv. Energy Mater.* **8**, 1801345 (2018).
- [32] Deng, S. et al. Synergistic doping and intercalation: realizing deep phase modulation on MoS₂ arrays for high-efficiency hydrogen evolution reaction. *Angew. Chem.* **58**, 16289-16296 (2019).
- [33] Chen, W. et al. Achieving rich and active alkaline hydrogen evolution heterostructures via interface engineering on 2D 1T-MoS₂ quantum sheets. *Adv. Funct. Mater.* **30**, 2000551 (2020).
- [34] Luo, Y. et al. Morphology and surface chemistry engineering toward pH-universal catalysts for hydrogen evolution at high current density. *Nat. Commun.* **10**, 269 (2019).

[35] Zhang, B. et al. Interface engineering: the Ni(OH)₂/MoS₂ heterostructure for highly efficient alkaline hydrogen evolution. *Nano Energy* **37**, 74-80 (2017).

[36] Cheng, Y. et al. Defects enhance the electrocatalytic hydrogen evolution properties of MoS₂-based materials[J]. *Chem Asian J.* **15**, 3123–3134 (2020).

Reviewer #3 (Remarks to the Author):

The third reviewer comment and response

List of changes (green in the revised manuscript)

Comment: The authors work to engineer the interfacial electronic structure of molybdenum sulfide, which demonstrated enhanced HER performance in both alkaline and acid electrolytes. DFT calculation was also used to prove the electronic structure regulation induced by Ni-based materials. However, at this current state, the evidence for some of the statements is not so convincing. I am reluctant to offer my favorable recommendation for the publication of this work in Nature Communications, with the following reasons:

Our response: We thank for the reviewer's overall comments for our manuscript. Based on the reviewer's comments above, in order to make our manuscript data more convincing, we first tried to supplement the in-situ electrochemical-Raman spectroscopy to further illustrate the real active sites of our samples. Secondly, we also added the X-ray absorption near-edge structure (XANES) spectra measurements, which can further prove the change of the oxidation state of Mo or Ni at the interface during the electrochemical reaction. Please see **Response 1** of **Reviewer#1** for details. At the same time, we also added the subsequent characterization after the electrocatalytic reaction, such as XRD, Raman, XPS data, etc. These results fully demonstrate the good stability of our catalyst, reduced nickel possesses empty orbitals, which is helpful for additional H binding ability. Please see **Response 11** of **Reviewer#1** and **Response 7** of **Reviewer#2** for details. In addition, we also added some theoretical calculation data, especially water-dissociation energy barrier, the explicit water dissociation related active sites and rate-limiting steps for the whole alkaline HER process, please see **Response 2** of **Reviewer#2** for details. By adding the new data mentioned above, we believe that our revised manuscript is more convincing. Our responses to the reviewer's comments are listed below one by one.

Comment 1: From the LSV curves in 1M KOH in Fig. 3a, the as-prepared electrocatalysts with the multi-heterojunction interfaces ($1T_{0.81}\text{-MoS}_2\text{@Ni}_2\text{P}$ and $1T_{0.72}\text{-MoS}_2\text{@NiS}_2$) show inferior HER performance to Pt/C. How could the authors declare that the as-prepared catalysts outperformed the commercial Pt/C in abstract? Besides, the authors declare "the incorporation of metallic-phase and intrinsic HER-active Ni-based materials ...reduced nickel possesses empty orbitals, which is helpful for additional H binding ability". However, I cannot find any characterizations and analysis to certify the "metallic-phase" and "reduced nickel" statement in the manuscript.

Response 1: (1) Thanks for the reviewer for pointing out this question. Indeed, the

LSV curve only shows a current density in the range of 0 ~ 40 mA cm⁻² in our original manuscript, and the overpotential of the commercial Pt/C catalyst is smaller in this range, indicating that Pt/C has a lower onset potential. However, when the current density increases to 200 mA cm⁻², it can be seen that the HER activity of as-prepared catalyst is significantly better than that of Pt/C (**Figure R30, this part of the data is based on your comment 7, reducing the electrochemical scan rate (5 mV/s) to regain the results**). And the Tafel slope is smaller than commercial Pt/C. We now have replaced these, please see **Fig. 3a, c** in our revised manuscript.

Figure R30. **a** The HER polarization curves of 1T_{0.81}-MoS₂@Ni₂P, 1T_{0.72}-MoS₂@NiS₂, 1T_{phase}-MoS₂, 1T_{0.41}-MoS₂, 2H_{phase}-MoS₂ and 20% Pt/C electrodes at 5.0 mV s⁻¹ in 1.0 M KOH solution. **b** The HER polarization curves of 1T_{0.81}-MoS₂@Ni₂P, 1T_{0.72}-MoS₂@NiS₂, 1T_{phase}-MoS₂, 1T_{0.41}-MoS₂, 2H_{phase}-MoS₂ and 20% Pt/C electrodes at 5.0 mV s⁻¹ in 0.5 M H₂SO₄ solution. (All polarization curves were corrected without iR-compensation).

(2) Thanks for the comments and suggestions from the reviewer. (a) For “metallic-phase”, MoS₂ is a typical polyphase material which includes trigonal (1T), hexagonal (2H), rhombohedral (3R). Among these, the unique structure of 1T phase MoS₂ endows it with a metallic-like property,¹ showing higher electrical conductivity and catalytic activity.²⁻⁵ In our manuscript, by phase-modulation synergistic with interfacial chemistry and defects of phosphorus or sulfur implantation, and we have successfully prepared a series of heterojunction-phase-interface electrocatalysts (1T_{0.81}-MoS₂@Ni₂P and 1T_{0.72}-MoS₂@NiS₂) with the 1T phase of 81% and 72%, respectively, which is among higher 1T ratios compared with other methods. Please see XPS result for details (Mo 3d, S 2p, **Fig. 2**) in our revised manuscript. The XRD and Raman results also proved this (**Supplementary Fig. 6**). So, we use the “metallic-phase” statement in the manuscript.

(b) In order to prove that Ni in the catalyst was reduced during the electrocatalytic reaction, we examined the XPS experiments of 1T_{0.81}-MoS₂@Ni₂P sample after the durability measurements in acidic and alkaline solutions. With regard to Ni 2p regions, the peaks of Ni 2p_{3/2}, Ni 2p_{1/2} and the corresponding satellite appear. These are also negative-shifted of about 0.8 eV and a new peak Ni⁰ (851.9 eV) appears for

$1T_{0.81}$ -MoS₂@Ni₂P sample after HER stability test in 1 M KOH solution (**Figure R31a**). In addition, after the HER reaction in an acidic solution, these also showed similar results. There is a new peak of Ni⁰ (852.5 eV) appeared for $1T_{0.81}$ -MoS₂@Ni₂P sample after HER stability test (**Figure R32b**), indicating that Ni is reduced during the HER process.

Figure R31. Ni 2p spectra for $1T_{0.72}$ -MoS₂@Ni₂P heterostructure catalyst after 16 h stability measurement at 45 mA cm⁻². (a) 1.0 M KOH solution, and (b) 0.5 M H₂SO₄ solution.

In addition, in order to study electronic states of Ni cations of catalysts, X-ray absorption near-edge structure (XANES) spectra were measured on the fresh catalysts and those after being used in the HER process at three representative potentials (-0.04, -0.1 and -0.2 V vs. RHE). **Figure R32a** presents Ni K-edge XANES spectra of $1T_{0.81}$ -MoS₂@Ni₂P catalyst. Using the edge positions of NiO and NiO₂ as references, the Ni average valence state (**Figure R32b**) is determined as +3.3, +2.2, +1.8 and +2.0 for the fresh and those used at -0.04, -0.1 and -0.2V vs. RHE, respectively. Therefore, Ni cations are reduced under working conditions. In addition, $1T_{0.72}$ -MoS₂@NiS₂ (**Figure R32c, d**) and $1T_{0.41}$ -MoS₂@Ni(OH)₂ (**Figure R32e, f**) show similar behavior as Ni valency is decreased from +3.6 (fresh) to +2.7 (-0.2 V) and from +2.7 (fresh) to 1.8 (-0.2 V), respectively. Please see the **Response 11 of Reviewer#1** for a detailed explanation. We have added these data and evidences in the revised manuscript (**Fig. 5a, Supplementary Fig. 47-49 and Supplementary Table 9**).

Figure R32. XANES spectra of fresh catalyst and after being used in the HER process at -0.04, -0.1 and -0.2 V vs RHE, respectively. **a** Ni K-edge XANES spectra of $1T_{0.81}\text{-MoS}_2@Ni_2P$ catalyst and standards Ni foil, NiO, and NiO₂. Inset, Magnified pre-edge XANES region. **b** the fitted average oxidation states of Ni from XANES spectra on the fresh catalyst and after being used in the HER process at -0.04, -0.1 and -0.2 V vs RHE, respectively. ΔE : Ni K-edge position (eV), relative to Ni foil, Error = ± 0.2 eV; OS: Oxidation state (Linear fit: OS = $2.2222 \cdot \Delta E - 26.2222$). **c** Ni K-edge XANES spectra of $1T_{0.72}\text{-MoS}_2@NiS_2$ catalyst and standard Ni foil, NiO, and NiO₂. Inset, Magnified pre-edge XANES region. **d** the fitted average oxidation states of Ni from XANES spectra on the fresh catalyst and after being used in the HER process at -0.04, and -0.1 V vs RHE, respectively. **e** XANES spectra recorded at the Ni K-edge of fresh $1T_{0.41}\text{-MoS}_2@Ni(OH)_2$, and at different applied voltages from the -0.04 to -0.2 V after electrocatalytic HER, and the XANES data of the reference standards of Ni foil, NiO, and NiO₂. **f** the fitted average oxidation states of Ni from XANES spectra on the fresh catalyst and after being used in the HER process at -0.04 V vs

RHE, respectively.

Reference:

- [1] Acerce, M. et al. Metallic 1T phase MoS₂ nanosheets as supercapacitor electrode materials[J]. Nat. Nanotechnol. **10**, 313 (2015).
- [2] Kang, Y. M. et al. Plasmonic hot electron induced structural phase transition in a MoS₂ monolayer[J]. Adv. Mater. **26**, 6467–6471 (2014).
- [3] Kang, K. et al. Targeted synthesis of 2H- and 1T-phase MoS₂ monolayers for catalytic hydrogen evolution[J]. Adv. Mater. **28**, 10033–10041 (2016).
- [4] Wei, S. T. et al. Iridium-triggered phase transition of MoS₂ nanosheets boosts overall water splitting in alkaline media[J]. ACS Energy Lett. **4**, 368–374 (2019).
- [5] Shang, B. et al. Lattice-mismatch-induced ultrastable 1T-phase MoS₂-Pd/Au for plasmon-enhanced hydrogen evolution[J]. Nano Lett. **19**, 2758–2764 (2019).

Comment 2: It's unnecessary to present two identical SEM images of 1T_{0.72}-MoS₂@NiS₂ in Supplementary Fig. 2, maybe one should be deleted.

Response 2: We thank the reviewer for his (her) suggestions. It is indeed due to our negligence, we have replaced one of them, please see **Supplementary Fig. 2** in revised Supplementary Information for details.

Comment 3: For better understanding the morphology of target catalysts, morphology characterizations of 1T_{0.41}-MoS₂@Ni(OH)₂ are essential. Besides, the authors described that many random Ni₂P nanoparticles distributed on the MoS₂ microspheres from line 164-166 in manuscript, but it is difficult to distinguish Ni₂P nanoparticles from Supplementary Fig. 3.

Response 3: We thank for the reviewer's helpful suggestions. (1) Based on your suggestion, we added morphology characterization of 1T_{0.41}-MoS₂@Ni(OH)₂ in our revised manuscript. When being electrodeposited for 100 s, a small amount of Ni(OH)₂ nanoparticles can be observed anchored on the surface of MoS₂ nanospheres from the SEM photos (**Figure R33**). When the electrodeposition time is increased to 300 s, a large number of Ni(OH)₂ nanoparticles can be observed to adhere to the MoS₂ surface (**Figure R34**).

(2) For the 1T_{0.81}-MoS₂@Ni₂P sample, due to the low magnification of the SEM photo in the original manuscript, it is difficult to distinguish the Ni₂P nanoparticles randomly distributed on the surface of the MoS₂ nanospheres. Now we zoomed in it for 50,000 times, it can clearly be seen for a large number of Ni₂P nanoparticles anchored on the MoS₂ surface (**Figure R35**). We have also added this part of the SEM data in the revised Supplementary Information.

Figure R33. (a-a₃) SEM images of 1T_{0.41}-MoS₂@Ni(OH)₂. Ni(OH)₂ nanoparticles were electrodeposited on the 1T_{0.41}-MoS₂ using 0.1 M Ni(NO₃)₂ at 5.0 mA/cm² cathode current density applied for 100 s.

Figure R34. (b-b₃) SEM images of 1T_{0.41}-MoS₂@Ni(OH)₂. Ni(OH)₂ nanoparticles were electrodeposited on the 1T_{0.41}-MoS₂ using 0.1 M Ni(NO₃)₂ at 5.0 mA/cm² cathode current density applied for 300 s.

Figure R35. (a-d) SEM images of $1T_{0.81}\text{-MoS}_2\text{@Ni}_2\text{P}$.

Comment 4: From line 191-196 in manuscript, the statement, that NiS_2 nanoparticles are between the $1T_{0.41}\text{-MoS}_2$ layer on the basis of Raman results with slightly red-shift of E_{2g}^1 and A_{1g} peaks, is not so convincing. Please cite some references to support this statement.

Response 4: We thank for the reviewer's helpful suggestions. According to your suggestion, we have added some references to support this statement in our revised manuscript. Zheng et al.¹ reported a 1T-2H MoS_2/Au heterostructure for surface enhanced Raman scattering, upon decoration of MoS_2 and 1T-2H MoS_2 with Au nanoparticles, the E_{2g}^1 and A_{1g} peaks are upshifted by $\sim 2\text{ cm}^{-1}$, indicating strong interaction between the Au and the MoS_2 substrate. Sun et al.² reported a $\text{Cu}_2\text{S}/\text{MoS}_2$ hetero-nanostructure towards an advanced catalyst for ullmann couplings. After immobilization of Cu_2S onto MoS_2 , the in-plane vibration mode E_{2g}^1 and the out-of-plane vibration mode A_{1g} are blue-shifted compared to pristine MoS_2 (from 379 to 381 cm^{-1} for E_{2g}^1 and from 404 to 410 cm^{-1} for A_{1g}). This could be attributed to the in-situ growth of Cu_2S , which exploits the S layer of MoS_2 as an external S source, hence changing the primitive vibration mode of the Mo-S bonds, of which the out-of-plane vibration mode is altered more significantly. Guo et al.³ developed a $\text{Ni}_3\text{S}_2/\text{MoS}_2$ heterojunction photocatalyst for enhancing hydrogen evolution. After the formation of $\text{Ni}_3\text{S}_2/\text{MoS}_2$ heterojunction, Ni_3S_2 nanostructures were attached with MoS_2 nanosheets, changing the vibration mode (E_{2g}^1 and A_{1g}) of MoS_2 because of interfacial stress between Ni_3S_2 and MoS_2 , indicating that the formation of $\text{Ni}_3\text{S}_2/\text{MoS}_2$ heterojunction leads to the Raman shift of MoS_2 .

References:

- [1] Zheng, X. L. et al. Building a lateral/vertical 1T-2H MoS₂/Au heterostructure for enhanced photo-electrocatalysis and surface enhanced Raman scattering [J]. *J. Mater. Chem. A*, **7**, 19922 (2019).
- [2] Sun, X. et al. Interface engineering in two-dimensional heterostructures: towards an advanced catalyst for ullmann couplings[J]. *Angew. Chem. Int. Ed.* **55**, 1704–1709 (2016).
- [3] Guo, S. H. et al. Enhanced hydrogen evolution via interlaced Ni₃S₂/MoS₂ heterojunction photocatalysts with efficient interfacial contact and broadband absorption[J]. *J. Alloys Compd.* **749**, 473e480 (2018).

Comment 5: From TEM images of 1T_{0.72}-MoS₂@NiS₂ and 1T_{0.81}-MoS₂@Ni₂P in Supplementary Fig. 5a, b and Supplementary Fig. 6a, b, the NiS₂ and Ni₂P seem to be in the form of nanosheets instead of nanoparticles, which needs to be further confirmed.

Response 5: We thank for the reviewer's helpful suggestions. According to your suggestion, we have added some TEM images of 1T_{0.72}-MoS₂@NiS₂ and 1T_{0.81}-MoS₂@Ni₂P samples. As shown in **Figure R36**, it is clearly observed that a large number of NiS₂ and Ni₂P nanoparticles are anchored on the MoS₂ nanosheets. In addition, the SEM images can also be clearly observed that a large number of Ni₂P nanoparticles are anchored on the surface of the MoS₂ nanospheres (**Figure R35**). We have also added this part of the TEM information in the revised Supplementary Information.

Figure R36. (a, b) Typical TEM images of $1T_{0.72}\text{-MoS}_2\text{@NiS}_2$ samples. (c, d) Typical TEM images of $1T_{0.81}\text{-MoS}_2\text{@Ni}_2\text{P}$ samples.

Comment 6: In the XPS analysis section, peak shift of Ni $2p_{1/2}$ and $2p_{3/2}$ in Ni 2p XPS spectrum for $1T_{0.72}\text{-MoS}_2\text{@NiS}_2$, $1T_{0.81}\text{-MoS}_2\text{@Ni}_2\text{P}$ cannot be used as the evidence for interaction between NiS_2 (or Ni_2P) and MoS_2 via as-formed hetero-structures. To demonstrate that, peak shift of Mo 3d may be more useful.

Response 6: We are very grateful for the reviewers' suggestions. For $1T_{0.41}\text{-MoS}_2$ sample, the high-resolution Mo 3d XPS core level (**Figure R37a**) shows that the two main peaks of Mo 3d located at 232.68 and 229.43 eV are identified as $\text{Mo}^{4+} 3d_{3/2}$ and $\text{Mo}^{4+} 3d_{5/2}$, respectively, confirming the existence of Mo^{4+} in the $1T_{0.41}\text{-MoS}_2$ sample. As for the $1T_{0.72}\text{-MoS}_2\text{@NiS}_2$, or $1T_{0.81}\text{-MoS}_2\text{@Ni}_2\text{P}$ heterostructures catalyst, the high-resolution Mo 3d XPS spectrum indicates that both $\text{Mo}^{4+} 3d_{3/2}$ and $\text{Mo}^{4+} 3d_{5/2}$ peaks for mixed phase MoS_2 exhibit a shift of 0.23 eV and 0.15 eV to lower binding energy compared to $1T_{0.41}\text{-MoS}_2$, which is attributed to the existence of $1T\text{-MoS}_2$.¹ In addition, two peaks located at 163.41 and 162.22 eV are observed in the $1T_{0.41}\text{-MoS}_2$ sample, corresponding to $\text{S}^{2-} 2p_{1/2}$ and $\text{S}^{2-} 2p_{3/2}$, respectively. However, the binding energies of $\text{S}^{2-} 2p_{1/2}$ and $\text{S}^{2-} 2p_{3/2}$ in $1T_{0.72}\text{-MoS}_2\text{@NiS}_2$, or $1T_{0.81}\text{-MoS}_2\text{@Ni}_2\text{P}$ heterostructures catalyst shift to 163.28 and 162.10 eV, respectively (**Figure R37b**).

This negative shift (0.13 eV) suggests little electron transfer between NiS₂ (or Ni₂P) and MoS₂, also suggesting the occurrence of electronic structure reconfiguration in electron transfer from Mo⁴⁺ to the surrounding Ni sites.²

Figure R37. (a) The high-resolution Mo 3d core-level XPS spectra of 1T_{0.41}-MoS₂ and 1T_{0.41}-MoS₂@Ni(OH)₂, 1T_{0.72}-MoS₂@NiS₂, and 1T_{0.81}-MoS₂@Ni₂P. (b) The high-resolution S 2p core-level XPS spectra of 1T_{0.41}-MoS₂ and 1T_{0.41}-MoS₂@Ni(OH)₂, 1T_{0.72}-MoS₂@NiS₂, and 1T_{0.81}-MoS₂@Ni₂P.

References:

- [1] Yu, Y. F. et al. High phase-purity 1T'-MoS₂- and 1T'-MoSe₂-layered crystals[J]. Nat. Chem. **10**, 638-643 (2018)
- [2] Zeng, L. Y. et al. Multiple modulations of pyrite nickel sulfides via metal heteroatom doping engineering for boosting alkaline and neutral hydrogen evolution[J]. J. Mater. Chem. A, **7**, 25628 (2019).

Comment 7: Usually, current density at overpotential of 0 V vs. RHE is very close to 0 mA cm⁻² for HER. However, it deviates far from X axis for 1T_{0.72}-MoS₂@NiS₂ and 1T_{0.81}-MoS₂@Ni₂P in both alkaline and acid electrolytes in Fig. 3a, c. Please explain.

Response 7: Thanks for the comments and suggestions from the reviewer. Indeed, you are right. According to the definition of the polarization curve, if the applied overpotential is 0 V and the current density deviates greatly from 0 mA cm⁻², it is because the sweep rate (50 mV/s) is too large during the electrochemical test, and the electrode surface is not in a steady state at this time. Now we reduce the scanning rate (1.0 ~ 5.0 mV/s) to stabilize the electrode surface, and re-measure the polarization curve as shown in **Figure R30**. This error has been corrected in our revised manuscript.

Comment 8: The reaction pathways on NiS₂ and Ni₂P need to be evaluated as a comparison.

Response 8: Thanks for the suggestions from the reviewer. According to your

suggestion, we have added some relevant information in our revised manuscript. We used DFT to calculate the optimized structures and free-energy diagrams for HER on NiS₂ and Ni₂P interface edges in alkaline medium, as shown in **Figure R38**. For example, NiS₂ (**Figure R38a**), the ΔG_{H^*} is very positive (0.81 eV), indicating that there is a strong interaction between H* and NiS₂, showing poor HER reaction kinetics. More importantly, NiS₂ exhibits unfavorable catalyst-OH_{ad} energetics, exhibiting relatively high activated water-adsorption energy ($\Delta G_{H_2O} = 0.59$ eV). This high ΔG_{H_2O} significantly hinders the decomposition of water into H* intermediates and leads to slow HER kinetics. Similarly, we also get similar results with Ni₂P (**Figure R38b**). Please see **Response 2** of **Reviewer#2** for details.

Figure R38. The optimized structures and Free-energy diagrams for HER on the NiS₂, and Ni₂P.

Comment 9: English writing needs to be improved all over the manuscript!

Response 9: We are very grateful for the reviewers' suggestions, and now we have revised our manuscript for several times after careful examination.

REVIEWERS' COMMENTS

Reviewer #1 (Remarks to the Author):

the authors have replied very satisfactorily to all reviewers' criticisms
I suggest to accept the paper for publication in its present form

Reviewer #2 (Remarks to the Author):

I have carefully reviewed the revised version of this manuscript. The authors have well answered my raised questions by providing some more detailed experimental results and new theoretical analysis. Besides, I have also checked the authors' replies to the other two reviewers' questions, and recognized these explanations to be reasonable. Therefore, after comprehensive consideration, the revised manuscript should have reached the high standard of this journal, and I recommend it to be published in Nature Communications.

Reviewer #3 (Remarks to the Author):

The revision has answered my questions, and I think it can be accepted for publication.

Response letter

We sincerely thank the referees for their careful review and valuable comments, which certainly help improve our manuscript.

Reviewer #1 (Remarks to the Author):

Comment: the authors have replied very satisfactorily to all reviewers' criticisms, I suggest to accept the paper for publication in its present form

Response: We warmly thank the reviewer for recommending our work for publication in *Nature Communications*.

Reviewer #2 (Remarks to the Author):

Comment: I have carefully reviewed the revised version of this manuscript. The authors have well answered my raised questions by providing some more detailed experimental results and new theoretical analysis. Besides, I have also checked the authors' replies to the other two reviewers' questions, and recognized these explanations to be reasonable. Therefore, after comprehensive consideration, the revised manuscript should have reached the high standard of this journal, and I recommend it to be published in Nature Communications.

Response: We thank the reviewer very much for positive recommendations and comments on the version of our work for publication in *Nature Communications*.

Reviewer #3 (Remarks to the Author):

Comment: the revision has answered my questions, and I think it can be accepted for publication.

Response: We warmly thank the reviewer for recommending our work for publication in *Nature Communications*.